



# Debris cover and the thinning of Kennicott Glacier, Alaska: in situ measurements, automated ice cliff delineation and distributed melt estimates

**Leif S. Anderson**[1,2], **William H. Armstrong**[1,3], **Robert S. Anderson**[1], **and Pascal Buri**[4]

[1]Department of Geological Sciences and Institute of Arctic and Alpine Research,
University of Colorado Campus Box 450, Boulder, CO 80309-0450, USA
[2]GFZ German Research Centre for Geosciences, Telegrafenberg, 14473 Potsdam, Germany
[3]Department of Geological and Environmental Sciences, Appalachian State University,
033 Rankin Science West, ASU Box 32067, Boone, NC 28608-2067, USA
[4]Geophysical Institute, University of Alaska-Fairbanks, 2156 Koyukuk Drive, Fairbanks, AK 99775, USA

**Correspondence:** Leif S. Anderson (leif.anderson@unil.ch)

**Abstract.** Many glaciers are thinning rapidly beneath melt-reducing debris cover, including Kennicott Glacier in Alaska where glacier-wide maximum thinning also occurs under debris. This contradiction has been explained by melt hotspots, such as ice cliffs, scattered within the debris cover. However, melt hotspots alone cannot account for the rapid thinning at Kennicott Glacier. We consider the significance of ice cliffs, debris, and ice dynamics in addressing this outstanding problem.

We collected abundant in situ measurements of debris thickness, sub-debris melt, and ice cliff backwasting, allowing for extrapolation across the debris-covered tongue (the study area and the lower $24.2\,\mathrm{km}^2$ of the $387\,\mathrm{km}^2$ glacier). A newly developed automatic ice cliff delineation method is the first to use only optical satellite imagery. The adaptive binary threshold method accurately estimates ice cliff coverage even where ice cliffs are small and debris color varies.

Kennicott Glacier exhibits the highest fractional area of ice cliffs (11.7 %) documented to date. Ice cliffs contribute 26 % of total melt across the glacier tongue. Although the *relative* importance of ice cliffs to area-average melt is significant, the *absolute* area-averaged melt is dominated by debris.

At Kennicott Glacier, glacier-wide melt rates are not maximized in the zone of maximum thinning. Declining ice discharge through time therefore explains the rapid thinning. There is more debris-covered ice in Alaska than in any other region on Earth. Through this study, Kennicott Glacier is the first glacier in Alaska, and the largest glacier globally, where melt across its debris-covered tongue has been rigorously quantified.

## 1 Introduction

Loose rock (debris) is common on glacier surfaces globally and is especially abundant on glaciers in Alaska (Scherler et al., 2018; Herreid and Pellicciotti, 2020). Where debris is thicker than a few centimeters it insulates the underlying ice, leading to the reduction of melt rates (Østrem, 1959; we refer to "thick debris" as any debris that reduces melt rates relative to bare-ice melt rates). Adding to this insulating effect, debris covers are expanding on many glaciers even as glaciers contract in response to rising temperatures (e.g., Tielidze et al., 2020). Expanding and thickening debris cover (Banerjee, 2017; Gibson et al., 2017) should reduce glacier thinning relative to glaciers without debris, but the melt-reducing effect of debris is not always apparent in the observed thinning patterns of glaciers (Kääb et al., 2012; Gardelle et al., 2013). In high-mountain Asia CE1 many debris-covered and debris-free glaciers are thinning at similar rates (Nuimura et al., 2012; Agarwal et al., 2017; Lamsal et al., 2017; Brun et al., 2018; Wu et al., 2018). This apparent paradox, in which rapid thinning is occurring under thick debris cover is known as the

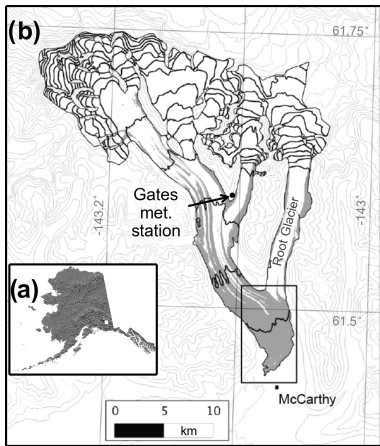

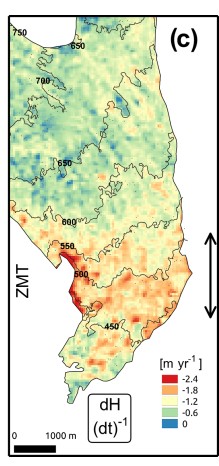

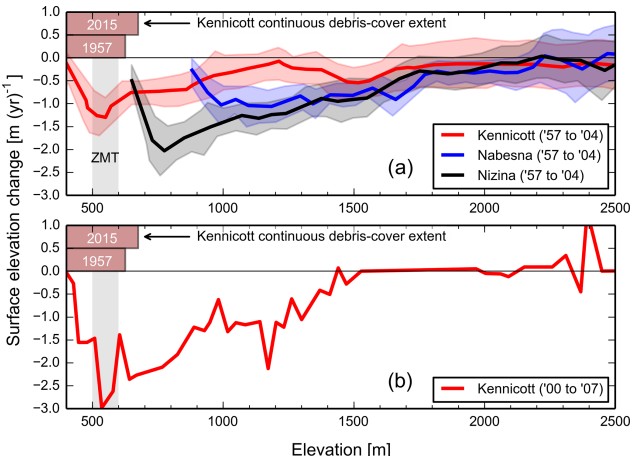

**Figure 1.** Map of Kennicott Glacier and the study area. **(a)** Map of Alaska showing the location of the Wrangell Mountains. **(b)** The Kennicott Glacier with the location of the Gates Glacier meteorological station (1240 m a.s.l.). May Creek meteorological station is located 15 km to the southwest of the terminus at 490 m a.s.l. Contour intervals are 250 m based on the ASTER GDEM V2 (2009). **(c)** Map of the general study area with $\mathrm{d}H\,(\mathrm{d}t)^{-1}$ from 1957 to 2004 (Das et al., 2014). ZMT refers to the zone of maximum thinning, the extent of which is shown with the double-headed arrow. This map of the study area includes the bare-ice parts of Root and Kennicott glaciers, where some ablation measurements were made. Elevation contours are from 2013. The units for the legend are above the labeled colors.

**Figure 2.** Surface elevation changes from three glaciers in the Wrangell Mountains. Surface elevation change data from Das et al. (2014). Elevations on the $x$ axis are derived from the 1957 digital elevation model (DEM). Take care in comparing these data to those presented in other figures which are referenced to the 2013 glacier surface. **(a)** Surface elevation change derived from DEM differencing. The shaded areas reflect the standard deviation of DEM differencing (see Das et al., 2014). Kennicott Glacier is the only glacier in the figure with a continuous debris cover spanning its entire width. The Nabesna and Nizina glaciers have individual medial moraines but the majority of the glaciers' termini are debris-free. The vertical grey bar is the zone of maximum thinning corrected for elevation differences. The greatest change in glacier surface elevation occurs within the portion of the glacier where debris spans the glacier width continuously between 1957 and 2015 (shown as brown bars; see Fig. S21). **(b)** Surface elevation change derived from laser altimetry profiles differenced from a DEM from 2000 to 2007. See Das et al. (2014) for the laser altimetry path and a discussion of uncertainties.

"debris-cover anomaly" (Pelliccciotti et al., 2015) and has also been documented in the European Alps (Mölg et al., 2019).

The debris-cover anomaly is occurring in Alaska, but to date, research into the effect of debris on glaciers in Alaska has been limited. A close look at previously published glacier thinning patterns from southeast Alaska reveals that maximum thinning rates within single glaciers are similar whether debris is present or absent (Figs. 1 and 2; Berthier et al., 2010; Das et al., 2014). Kennicott Glacier in the Wrangell Mountains is an example where rapid thinning is occurring under debris cover (Figs. 1 and 2). Greater thinning is documented within the Kennicott debris-covered tongue than from any portion of the largely debris-free Nabesna Glacier, north of Kennicott Glacier (Fig. 2).

This brings us to our overarching question: why does the maximum thinning of Kennicott Glacier occur under debris at rates similar to nearby debris-free glaciers? To guide our analysis, we define a zone of maximum thinning or ZMT where Kennicott Glacier thinned at an average rate greater than $1.2\,\mathrm{m\,yr^{-1}}$ between 1957 and 2004 (Figs. 1 and 2; Das et al., 2014). For Kennicott Glacier, thinning rates this high only occur within 4 km of the terminus and under debris. The ZMT occupies a 2 km downglacier by 3.5 km across-glacier portion of the debris-covered tongue. The ZMT, as defined, is consistent with maximum thinning rates between 2000 and 2007 (Fig. 2; Das et al., 2014).

The continuity equation for ice is fundamental for understanding how glaciers thin, with or without debris. It can be formulated as

$$\frac{\mathrm{d}H}{\mathrm{d}t} = \dot{b} - \nabla \cdot Q, \tag{1}$$

where $H$ is the ice thickness, $t$ is time, $\dot{b}$ is the annual specific mass balance (or loosely ice melt in the ablation zone), and $Q$ is the column-integrated ice discharge (Fig. 3). Constraining $\dot{b}$ on debris-covered glaciers is particularly difficult due to the presence of ice cliffs, ponds, and streams within debris covers. The annual specific balance in the ablation zone can be subdivided,

$$\dot{b} = \dot{b}_{\mathrm{s}} + \dot{b}_{\mathrm{e}} + \dot{b}_{\mathrm{b}}, \tag{2}$$

where $\dot{b}_{\mathrm{s}}$ is the annual surface ablation, $\dot{b}_{\mathrm{e}}$ is the annual englacial ablation, and $\dot{b}_{\mathrm{b}}$ is the annual basal ablation rate. Surface ablation typically dominates $\dot{b}$ in most non-polar settings. We neglect the effects of $\dot{b}_{\mathrm{e}}$ and $\dot{b}_{\mathrm{b}}$ because their contribution to rapid thinning is likely small and it is not yet possible to quantify them within and under debris-covered

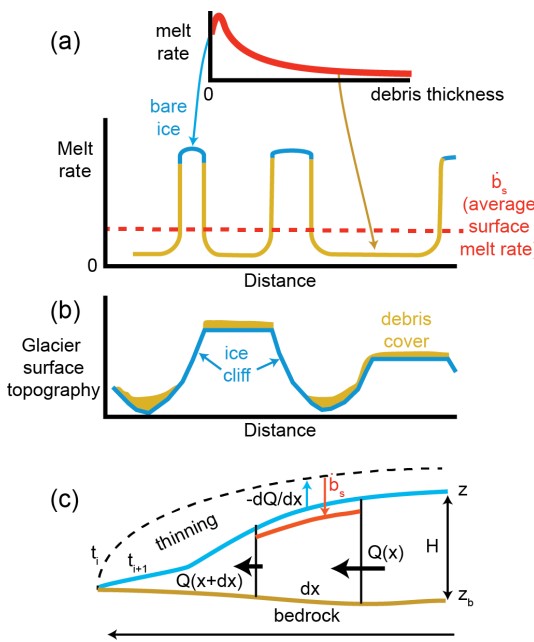

**Figure 3.** Schematic comparing the relative roles of ice cliff backwasting, sub-debris melt, and ice emergence to the lowering of an idealized glacier terminus. **(a)** Idealized relationship between ice cliff backwasting and sub-debris melt. Note that the inclination and low albedo of ice cliffs can lead to melt rates that exceed bare-ice melt rates on a flat surface. **(b)** Glacier surface topography with debris cover and ice cliffs compared to melt rates in **(a)**. **(c)** Schematic showing the relationship between surface melt, ice dynamics, and the thinning of the glacier through time.

tongues (see Benn et al., 2017). Building from Eq. (1), $\dot{b}_s$ is negative in the ablation zone and therefore shifts $\frac{dH}{dt}$ towards negative values, thinning the glacier. In the ablation zone, ice emergence velocity, more formally referred to as $-\nabla \cdot Q = -\partial Q_x / \partial x - \partial Q_x \boxed{} / \partial y$ ($x$ is the along-flow direction, and $y$ is the across-flow direction), tends to be positive due to the slowing of ice downglacier. This ice emergence velocity counters surface lowering due to melt every year.

Two common explanations for the debris-cover anomaly follow from Eq. (1), which are not mutually exclusive (Immerzeel et al., 2014; Vincent et al., 2016; Brun et al., 2018). First, it is possible that surface melt $\dot{b}$ is higher than we expect from the melt-reducing debris alone, therefore leading to rapid thinning. Ponds and ice cliffs can locally increase melt rates by an order of magnitude compared to adjacent melt rates measured under debris (Immerzeel et al., 2014). *Melt hotspots* such as ice cliffs, ponds, streams, and thermokarst counter the insulating effects of debris by raising area-averaged melt rates (Kirkbride, 1993; Sakai et al., 2002; Reid and Brock, 2014; Miles et al., 2018). Conceptually, melt hotspots perturb the area-averaged melt rate from a melt rate solely defined by the melt-reducing effects of debris towards a melt rate solely defined by the melt of bare ice. The degree

to which these hotspots increase area-averaged melt rates is an area of active debate. Second, less-positive surface mass balance upglacier from the zone of maximum thinning leads to reduced ice flow into the ZMT. Reduced ice flow leads to declining ice emergence rates and locally amplified thinning (Nye, 1960; Vincent et al., 2016). We revisit the continuity equation for ice in the discussion.

Kennicott Glacier provides an opportunity to test the importance of melt hotspots in controlling debris-covered glacier thinning: more than 15 000 TS2 ice cliffs are scattered within otherwise continuous debris (Anderson, 2014). If melt hotspots are the only control on the location of the ZMT for Kennicott Glacier then we should expect melt rates (averaged across the glacier width) to be maximized there. Here, we address two questions. (1) What is the surface mass balance across the debris-covered tongue and zone of maximum thinning of Kennicott Glacier? (2) Do ice cliffs maximize glacier-wide melt in the zone of maximum thinning? To address these questions, we quantify the role of ice cliffs and debris in setting the melt pattern across the debris-covered tongue of Kennicott Glacier.

Partly because of the significant effort required, in situ measurements from debris-covered glaciers are abundant on only a few keystone glaciers in the Himalayas (e.g., Lirung, Ngozumpa, and Khumbu glaciers; Benn et al., 2012; Immerzeel et al., 2014) and European Alps (e.g., Miage and Zmutt glaciers; Brock et al., 2010; Mölg et al., 2019). The lack of in situ observations from a range of debris-covered glaciers hinders the inclusion of debris cover in global projections of glacier change. Measurements from debris-covered areas in overlooked regions like Alaska are therefore a pressing need.

Using abundant in situ measurements, we estimate the distributed melt rate across the debris-covered tongue of Kennicott Glacier for the summer of 2011. We measured debris thickness, debris conductivity, air temperature, sub-debris melt rates, and ice cliff backwasting rates. We focus on the effects of ice cliffs, which are abundant at Kennicott Glacier, leaving a detailed examination of other melt hotspots for a later contribution. Despite this, we consider the general role of melt hotspots in sensitivity tests in the discussion.

In order to generate distributed melt estimates on debris-covered glaciers, we must delineate ice cliff extent. Quantifying their extent efficiently and accurately is difficult. Previous efforts to delineate ice cliffs have largely relied on the manual digitization of remotely sensed data (Sakai et al., 1998; Han et al., 2010; Thompson et al., 2016; Watson et al., 2017). Automatic methods include object-based image analysis using images derived from unmanned aerial vehicles (Kraaijenbrink et al., 2016) and principal component analysis using near-infrared and infrared satellite bands (Racoviteanu and Williams, 2012). Herreid and Pellicciotti (2018) most recently developed an automatic method to delineate ice cliffs using digital elevation models (DEMs). Despite the efforts of projects like the ArcticDEM (Porter et al., 2018), glacier

coverage with high-resolution DEMs (or high-resolution hyperspectral imagery) is still rarer than coverage with high-resolution optical-satellite imagery.

Here we develop a novel, automatic method to delineate ice cliffs using only 0.5 m resolution WorldView-1 satellite imagery. We use this method to delineate the abundant ice cliffs on the surface of Kennicott Glacier. We combine our in situ measurements and remotely delineated ice cliffs to quantify surface melt rates in a distributed fashion across the zone of maximum thinning.

## Study glacier

Kennicott Glacier is a large ($387 \, \mathrm{km^2}$) broadly south-southeast facing glacier on the south side of the Wrangell Mountains at $61.5°$ N. The glacier exists across a 4600 m elevation range between 5000 and 400 m a.s.l. (Fig. 1). For comparison, Khumbu Glacier, in Nepal, has an area of $26.5 \, \mathrm{km^2}$ and spans an elevation range of 3950 m from 8850 to about 4900 m a.s.l. (Pfeffer et al., 2014). Kennicott Glacier is almost 15 times larger by area than Khumbu Glacier. The main trunk of Kennicott Glacier is 42 km long and is joined by two tributaries, the Root and Gates glaciers. Kennicott Glacier has only retreated 600 m since its maximum Little Ice Age extent in 1860 (Rickman and Rosenkrans, 1997).

As of 2015, 20 % of Kennicott Glacier was debris-covered (based on manual digitization of a Landsat image). At elevations below the equilibrium-line altitude at about 1500 m a.s.l. (Armstrong et al., 2017), nine medial moraines are identifiable within the debris-covered tongue. These medial moraines form primarily from the erosion of hillslopes above the glacier and express themselves as stripes on the glacier surface (Anderson, 2000). Above 700 m a.s.l., debris is typically about one clast thick (Anderson, 2014). Below this elevation debris thickness tends to increase downglacier through the debris-covered tongue (Anderson and Anderson, 2018). The medial moraines coalesce in the last 7 km of the glacier where ice cliffs, surface ponds, and streams are scattered within otherwise continuous debris cover.

## 2 Methods

Our methods fit into three categories: (1) in situ measurements, (2) automatic ice cliff delineation, and (3) distributed melt rate estimates. In situ measurements were made within the broad study area shown in Fig. 1c. Distributed melt estimates on the other hand are made across the delineated medial moraines shown in Fig. 4a. In total this $24.2 \, \mathrm{km^2}$ study area is referred to as the debris-covered tongue and is similar in size to the entirety of Khumbu Glacier. In situ measurements were all made within the study period from 18 June to 16 August 2011. All melt rate measurements are in ice equivalent units. We used WorldView stereoimagery from 2013 to produce glacier surface DEMs at 5 m spatial resolution using

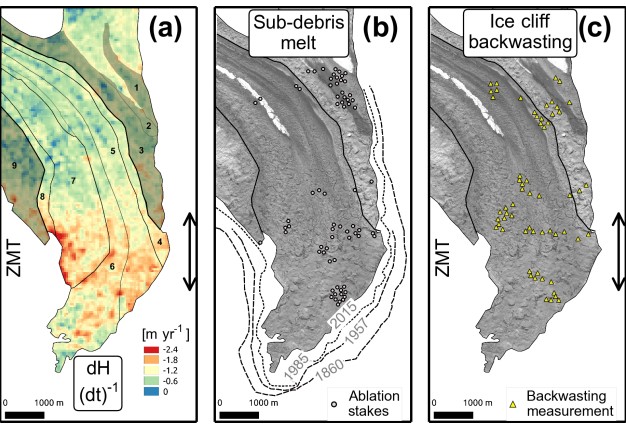

**Figure 4.** The study area with defined medial moraines and in situ measurement locations. This map of the study area includes the bare ice parts of Root Glacier, which are excluded and masked when making distributed melt estimates. The area defined by the nine medial moraines in panel **(a)** is used for distributed melt estimates. **(a)** Glacier thinning data from 1957 to 2004 (Das et al., 2014). This panel uses the same data as in Fig. 1c but the medial moraines are defined. The shaded medial moraines are treated differently for distributed debris thickness estimates (see Sects. 2.3 and S3.1). Note that medial moraines 4 through 8 contain the majority of the zone of maximum thinning. Medial moraines 3 and 9 show much thicker debris at the same elevation than the others (Fig. S14). The zone of maximum thinning (ZMT) is shown by the double-headed arrow. **(b)** Sub-debris melt rate measurement locations. Debris was measured at all locations in **(b)** and **(c)**; in some cases ice cliffs and sub-debris measurements were proximal and only one debris thickness measurement was made between them. The five central medial moraines are within the two black lines, within which 69 % of debris thickness measurements were made. **(c)** Locations where ice cliff backwasting rate was measured.

the Ames Stereo Pipeline (Shean et al., 2016), which we use to represent the glacier surface during the study period.

## 2.1 In situ measurements

Determining average melt rates across debris-covered areas is challenging due to the number and diversity of processes involved. Our solution is simple: to make abundant in situ measurements across the study area. For debris to be incorporated into large-scale models, debris thermal properties and on-glacier meteorology must also be documented as they vary across glacier surfaces. We also provide debris thermal conductivities (10 sites) and on-glacier air temperatures (three sites) (Sect. S1.4–S1.5 in the Supplement).

We measured debris thicknesses at 109 sites by digging through the debris to the ice surface (Figs. 5 and S1; after Zhang et al., 2011). Debris measurement locations coincide with the sites where we also measured ice cliff backwasting and sub-debris melt (Figs. 4 and S2–S3). Where debris was thinner than $\sim 10 \, \mathrm{cm}$, we dug several pits and recorded the average debris thickness. Uncertainty estimates were based

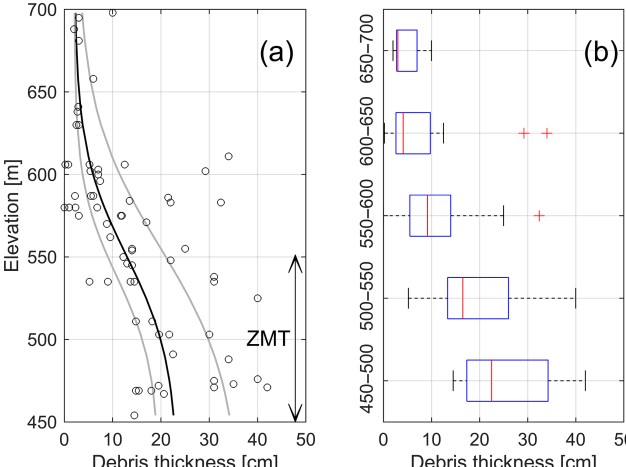

**Figure 5.** Debris thickness measurements for the five central medial moraines. **(a)** Debris thickness measurements as they vary with elevation (also see Fig. S14). The points plotted are the mean-measured debris thicknesses with symmetrical uncertainties around them. Curve fits through the median debris thickness (bold line) and the 25 % and 75 % quartiles (grey lines) from 50 m elevation bins are shown in **(b)**. The double-headed arrow represents the zone of maximum thinning. **(b)** Box plots of debris thickness binned in 50 m elevation bands. The red bars are the median and the vertical blue bars are the 25 % and 75 % quartiles, respectively. Note the sigmoidal shape of debris thickness with elevation. See the Supplement for curve fits applied to the other medial moraines as well as an exploration of linear curve fits through the data (Figs. S14 and S18).

on the repeated measurement of debris thickness at 52 ablation stakes.

We measured sub-debris melt at 74 locations (Figs. 4 and S4–S6). At each site we removed debris, installed ablation stakes, and then replaced the debris. We placed stakes in debris up to 40 cm thick. Sub-debris melt ($\dot{b}_{debris}$) was measured by removing the debris and measuring ice surface lowering (Fig. 6). We estimated uncertainty using data from all ablation stakes based on the uncertainty in marking and measurement as well as the tilt of the stake. We assume a $\pm 2$ cm error in the distance measurement along ablation stakes. The average-measured tilt of the ablation stakes was 5° from vertical. Bare-ice melt rates were also measured at several locations in the northeastern portion of the study area on the Root Glacier.

We measured in situ backwasting rates from 60 ice cliffs (Figs. 7 and S7–S8). We made repeat horizontal distance measurements between the upper ice cliff edge and a stationary marker (in a moving reference frame; after Han et al., 2010). Using all 60 measured ice cliffs, backwasting rate error was estimated based on an assumed uncertainty of $\pm 20$ cm applied to the initial and final distance measurements.

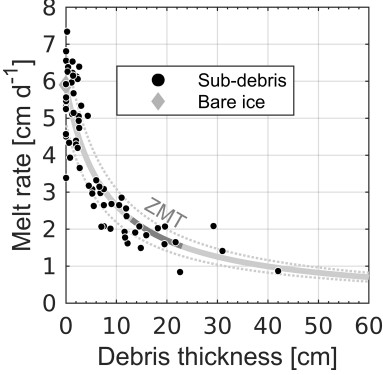

**Figure 6.** Sub-debris melt rate measurements. Melt rate as it varies with debris thickness. Sub-debris melt rates are corrected for the different measurement periods (see Sect. S1.1). The solid line is the curve fit using the hyper-fit CE2 model for the best debris thickness–melt CE3 relationship (RMSE of the data is 0.8 cm d$^{-1}$). The portion of the best curve fit in the zone of maximum thinning (ZMT) is shaded darker than the rest of the line. The dotted lines represent the $\pm 1\sigma$ error bounds used in the uncertainty estimates of distributed melt.

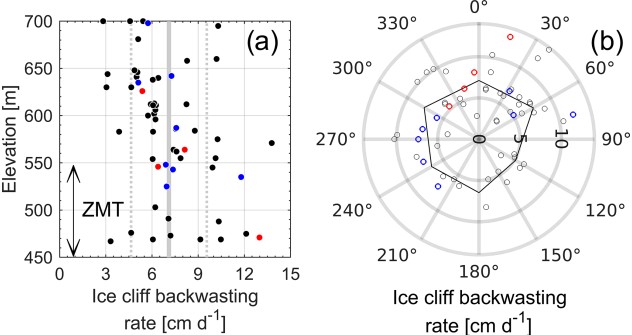

**Figure 7.** Ice cliff backwasting rate measurements. Ice cliff backwasting rates are corrected for the different measurement periods (Sect. S1.2). Cliffs with streams at their base are blue. Cliffs with ponds at their base are red. **(a)** Ice cliff backwasting rate as it varies with elevation. The solid grey line is the mean of all data 7.1 cm d$^{-1}$. The dashed lines are $\pm 1\sigma$ bounds used in the distributed melt calculations. The double-headed arrow represents the zone of maximum thinning (ZMT). **(b)** Ice cliff backwasting rate as it varies with aspect. The corners in the solid black line represent the mean backwasting rate from 60° bins. During the field survey, ice cliffs with ponds at their base were only found facing northward (between 300 and 30°).

Degree-day factors for each melt rate and backwasting rate measurement were calculated using air temperature data from off-ice meteorological stations (see Sect. S1.1–S1.2, S1.6 for the full explanation; Hock, 2003). We used hourly 2 m air temperature data from the Gates Glacier and May Creek meteorological stations to estimate the air temperature at each measurement location. Gates Glacier station is located just off the glacier margin at 1240 m a.s.l., and May

Creek station is located at 490 m a.s.l., 15 km to the southwest of the glacier terminus (Fig. 1). Each sub-debris melt and backwasting rate measurement was adjusted to represent the full study period using these degree-day factors. These corrections have a negligible effect on the distributed melt estimates. To represent the hypothetical case that no debris was present on the glacier, we also extrapolate bare-ice melt rates across the study area (Sect. S1.6).

## 2.2 Automatic ice cliff delineation methods

We develop an automated algorithm to delineate ice cliffs from optical satellite imagery. We use 0.5 m resolution WorldView satellite imagery acquired on 13 July 2009 (catalog ID: 1020010008B20800) to delineate ice cliffs across the study area. We use the panchromatic band, which integrates radiance across the visible spectrum and provides the highest spatial resolution. The 2009 WorldView image was the closest high-resolution image available in time to the 2011 summer field campaign. Our method for detecting ice cliffs relies on the observation that ice cliffs are generally darker than the debris around them. Ice cliffs, when actively melting, are typically coated with a thin, wet CE4 debris film that appears darker than the adjacent, dry debris in panchromatic-optical imagery (Fig. 8). In addition, steep ice cliffs are often more shaded than nearby lower-sloped debris-covered surfaces.

The workflow we outline relies on open-source Python packages, which facilitates the method's replication and improvement by other researchers. Our workflow consists of three general steps: (1) processing, stretching the image brightness histogram to a suitable range for our ice cliff detection methods; (2) detection, applying an ice cliff detection method; and (3) post-processing, morphologically filtering the detected ice cliffs (Fig. 8). We apply a linear histogram stretch uniformly across the image, including both the glacier and surrounding off-ice areas. These steps introduce several processing parameters, which we select using a Monte Carlo optimization method. Below, we first present the processing steps, followed by our parameter optimization procedure.

We test two methods to detect ice cliffs: (i) the adaptive binary threshold method (ABT; skimage.filters.adpative_threshold tool; e.g., Sauvola and Pietikäinen, 2000), and (ii) the Sobel edge delineation method (SED; skimage.filters.sobel tool; Richards, 2013). In pre-processing, we use separate saturation stretches (Fig. 8) for each method by applying the exposure function in the scikit-image package (skimage). The different methods perform best with different exposure levels, so we create two separate, stretched orthoimages in pre-processing.

The ABT approach runs a moving window over the image, calculates the mean-brightness value within that window, and then uses a threshold to binarize the image. Because the brightness threshold varies across the image, the ABT approach is less sensitive to changes in illumination and debris color than a global threshold.

The SED approach estimates spatial gradients in image brightness. The Sobel operator detects high contrasts between light-colored debris and dark-colored ice cliffs. The saturation stretch applied on the orthoimage causes dark ice cliffs to appear as featureless black regions, which the Sobel operator returns as low gradient values. We apply a brightness-gradient threshold to isolate ice cliffs.

The last step in our processing process is morphological filtering to remove spurious data. Both delineation methods (ABT and SED) produce false positives from shaded, overexposed, or textureless debris cover (SED only). The SED approach produces many false positives that which generally have a characteristic speckled appearance and often occur in small, isolated groups. We apply morphological opening (Dougherty, 1992) to remove these isolated false positives in both the ABT and SED approaches (skimage.morphology.opening; Fig. 8). In addition, the SED approach creates false positives in regions that have been overexposed by the saturation stretch and therefore lack texture. For the SED method only, we remove these false positives by masking pixels with the maximum brightness.

To maximize correct ice cliff identification and minimize false positives, we compare our ice cliff estimates to hand-digitized ice cliffs from twelve 90 000 m$^2$ regions. The cumulative area in the validation dataset was 1.8 km$^2$, approximately 7.4 % of the 24.2 km$^2$ study area (Fig. 9). There is some operator subjectivity in delineating ice cliffs from satellite imagery, especially for smaller ice cliffs (Steiner et al., 2019). To minimize this issue, two different human operators independently delineated ice cliffs. As these independent delineations agree within 3 % in their total ice cliff area, we consider operator misidentification to be a negligible source of error.

Seven parameters determine the success of these ice cliff delineation methods: (i–ii) the low- and high-end brightness values used CE5 for the saturation stretch; (iii–iv) the window size and offset from mean brightness in the ABT method, (v) the high-end value for thresholding in the SED method, and (vi–vii) the kernel sizes for morphological filtering of the SED and ABT results. To find the best parameter set, we use a Monte Carlo approach for multi-objective optimization (Yapo et al., 1998). We ran the ice cliff detection algorithm 2500 times with differing parameter choices. In each iteration, every parameter is randomly selected using uniform-probability distributions over that respective parameter's range of possible values (Duan et al., 1992). This method allows us to efficiently test performance across a wide range of parameter values and is sensitive to interaction between selected parameters across their ranges. We evaluate algorithm performance by comparing ice cliff area from the automated routine against the hand-digitized validation dataset. Our optimization simultaneously seeks to maximize true-positive ice cliff delineation, while minimizing

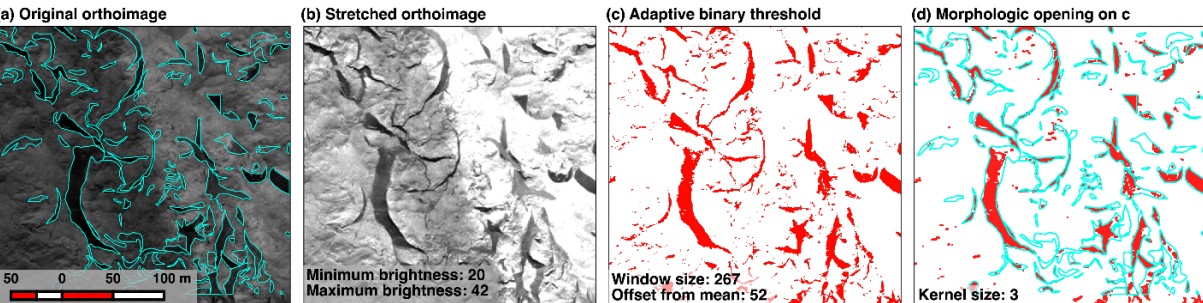

**Figure 8.** Ice cliff delineation workflow for the adaptive binary threshold (ABT) method. The extent of this area is shown by the third cyan box from the right in Fig. 9. **(a)** Original orthoimage with manually digitized ice cliffs shown in cyan. **(b)** Orthoimage after histogram stretch using a set of well-performing brightness values from the parameter optimization. **(c)** ABT on stretched orthoimage. **(d)** Morphologic opening on adaptive binary threshold to remove small isolated false positive ice cliff delineations. Manually digitized ice cliffs used as the validation dataset are again shown in cyan.

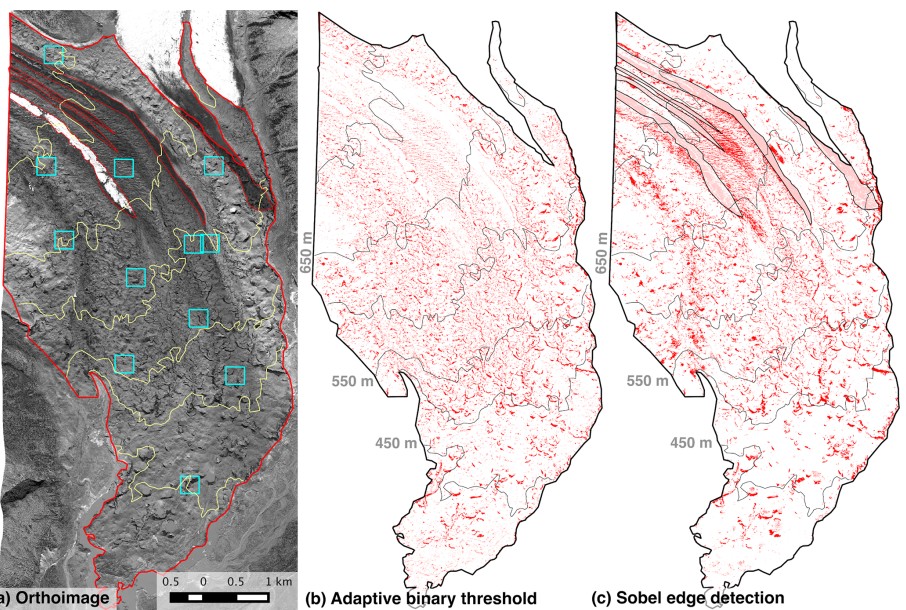

**Figure 9.** Results from the two ice cliff delineation methods. **(a)** Orthoimage of the terminus of Kennicott Glacier with the debris-covered area used for distributed melt estimates outlined by the thicker red line. The thinner red lines show regions of dark and light bare ice that required special treatment for the SED method. Thin yellow lines are elevation contours with a 50 m contour interval from 2013. Blue boxes show the locations of manually digitized ice cliff area, used for error analysis and parameter optimization. **(b)** Ice cliff spatial distribution as estimated by the adaptive binary threshold (ABT) method. The outlines in panels **(a)** and **(b)** show the area used for distributed melt calculations. **(c)** Ice cliff spatial distribution as estimated by the Sobel edge delineation (SED) method, with overlaid elevation contours from 2013.

false positives and false negatives. We manually inspect the top-performing parameter sets, ranked by Euclidean distance from the origin (Fig. S13), which defines perfect-algorithm performance (Sect. S2; Reed et al., 2013). We chose image processing parameters slightly off the set with the smallest Euclidean distance to reduce false positives (Table S3). We reduce false positives at the expense of true positives because this led to a higher ratio of true positives to false positives, so we are more certain that a given detection is likely to be a real ice cliff.

## 2.3 Distributed melt estimates

In order to extrapolate our in situ measurements across the study area, we divide the summer specific mass balance $\dot{b}_s$ into contributions from sub-debris and ice cliff melt: $\dot{b}_{debris}$ and $\dot{b}_{icecliff}$. Each 0.5 m pixel is designated as debris or ice cliff using the ABT ice cliff delineation method. We use the ABT method because it consistently performs better than the SED method (see Results section). For our best-case distributed-melt estimates, we apply a bias correction by adding 20 % to the ice cliff area in each elevation band based

on the consistent underprediction of ice cliffs. Extreme ice cliff areas are represented with $\pm 20\%$ areas from the best case.

We extrapolate debris thickness across the study area by applying the elevation-dependent curve fits to debris-designated pixels. For the five medial moraines in the center of the glacier (labeled 4–8 in Fig. 4a) in which 69 % of debris thickness measurements were made, we apply a sigmoidal curve fit (Fig. 5). Within these five medial moraines, debris thickness $h_{\mathrm{debris}}$ varies with elevation $z$ according to

$$h_{\mathrm{debris}} = \frac{a}{\left[1 + 10^{b(z-c)}\right]} + d, \qquad (3)$$

where $a$, $b$, $c$, and $d$ are fitted parameters derived using MATLAB's polyfit function (Table 1). We apply this sigmoidal curve fit because it best matches the pattern of debris thicknesses within these five medial moraines when they are binned in 50 m elevation bands. For other medial moraines with fewer debris thickness measurements, we apply linear curve fits (Fig. S14). For the westernmost medial moraine (no. 9 in Fig. 4a), which was difficult to access, we apply uniform debris thicknesses based on a few measurements. We test the importance of the debris thickness applied to medial moraine no. 9 in the Supplement (Sect. S3.1.2), the importance of this assumed debris thickness is minor, and viable debris thicknesses are well within the uncertainty scenarios explored.

We apply sub-debris melt rates to all debris-designated pixels based on the estimated debris thickness in each pixel. We use the hyper-fit CE6 model to relate debris thickness to sub-debris melt (Anderson and Anderson, 2016; Crump et al., 2017; Anderson et al., 2018). In the model, the relationship between specific-sub-debris melt $\dot{b}_{\mathrm{debris}}$ and debris thickness is

$$\dot{b}_{\mathrm{debris}} = \dot{b}_{\mathrm{ice}} \frac{h_*}{(h_{\mathrm{debris}} + h_*)}, \qquad (4)$$

where $\dot{b}_{\mathrm{ice}}$ the bare-ice melt rate measured near the top of the study area, and $h_*$ the characteristic debris thickness have values of 5.87 cm d$^{-1}$ and 8.17 cm respectively (Fig. 6). Sub-debris melt rates under debris $h_*$ thick will be half the value of the bare-ice melt rate. If ice is assumed to be at 0 °C, $h_*$ can be estimated from physical inputs and parameters following

$$h_* = \frac{kR}{(1 - \phi)}, \qquad (5)$$

where $k$ and $\varphi$ are the thermal conductivity and porosity of the debris cover and $R$ is the thermal resistance of the debris layer. Here we define $R$ as

$$R = \frac{\overline{T}_{\mathrm{s}}}{L \rho_{\mathrm{ice}} \dot{b}_{\mathrm{ice}}}, \qquad (6)$$

where $L$ and $\rho_{\mathrm{ice}}$ are the latent heat of fusion and density of ice, $\overline{T}_{\mathrm{s}}$ is the average debris surface temperature over the period used to estimate $h_*$, and $\dot{b}_{\mathrm{ice}}$ in this case is the bare-ice melt rate over the period used to estimate $h_*$. The hyperbolic fit between debris thickness and sub-debris melt assumes that energy is transferred through the debris by conduction. While these debris parameters can be measured, in practice they are difficult to measure across debris-covered glaciers so we use an empirical fit to debris thickness–melt data to constrain $h_*$.

We apply a uniform ice cliff backwasting rate to all ice-cliff-designated pixels. We ignore ice cliff backwasting variation with orientation, as there is no clear relationship between backwasting rate and orientation in our measurements (Fig. 7). We did not find a consistent difference between backwasting for ice cliffs with and without ponds at their base (Fig. 7) and no clear relationship between backwasting rate and medial moraine is apparent either (Fig. S8). We apply the mean specific horizontal ice cliff retreat across the study area:

$$\dot{b}_{\mathrm{backwasting}} = f, \qquad (7)$$

where $f$ is the mean backwasting rate 7.1 cm d$^{-1}$ (an elevation-dependent pattern is explored in Sect. S3.1.3). Because backwasting rates are measured horizontally, we apply an average dip relative to the horizontal plane ($\theta$) to estimate the melt perpendicular to ice cliff surfaces:

$$\dot{b}_{\mathrm{icecliff}} = \dot{b}_{\mathrm{backwasting}} \cos\left(90° - \theta\right). \qquad (8)$$

In the best case we assume a uniform ice cliff slope ($\theta$) for all ice cliffs of 48° based on the mean of slope measurements made at the top of each of the 60 ice cliffs where backwasting rates were measured (following Han et al., 2010). The mean of average ice cliff slope from six other glaciers is 49° (Sect. S1.3). Including the average slope estimate from this study, the standard deviation of mean ice cliff slopes is 5°, which we use for our uncertainty estimates.

In order to estimate melt rates with elevation, we integrate the contributions of ice cliff and sub-debris ablation across 20 m elevation bands:

$$\overline{b}^i = \frac{\displaystyle\iint\limits^{A^i_{\mathrm{debris}}} \dot{b}_{\mathrm{debris}} \mathrm{d}x\mathrm{d}y + \iint\limits^{A^i_{\mathrm{icecliff}}} \dot{b}_{\mathrm{icecliff}} \mathrm{d}x\mathrm{d}y}{A^i}, \qquad (9)$$

where $\overline{b}^i$ is the mean ablation rate within the elevation band $i$ in units of centimeter per day, $A^i_{\mathrm{debris}}$ is the total debris-covered area, corrected for the surface slope of each debris-covered pixel using the 2013 DEM, within the elevation band; $A^i_{\mathrm{icecliff}}$ is the total ice cliff area, correcting for the slope of each ice cliff pixel based on the assumed ice cliff slope, within the elevation band; $A^i$ is the total plan view area within the elevation band; and $\mathrm{d}x$ and $\mathrm{d}y$ are both 0.5 m.

**Table 1.** Parameters used for the best distributed melt and uncertainty estimates.

| Parameter name | Parameter symbol | Lower bound | Best | Upper bound | |
|---|---|---|---|---|---|
| Debris thickness [cm] | $a$ | 17.6 | 21.6 | 34.3 | Interquartile range |
| | $b$ | 0.016 | 0.13 | 0.010 | |
| | $c$ | 538 | 551 | 556 | |
| | $d$ | 2.1 | 2.1 | 2.6 | |
| Sub-debris melt rate [cm d$^{-1}$] | $\dot{b}_{\mathrm{ice}}$ | 4.87 | 5.87 | 6.87 | $\pm 1$ SD |
| | $h_*$ | 8.17 | 8.17 | 8.17 | |
| Ice cliff backwasting [cm d$^{-1}$] | $f$ | 4.6 | 7.1 | 9.6 | $\pm 1$ SD |
| Ice cliff slope [degree] | $\theta$ | 43 | 48 | 53 | $\pm 1$ SD |

### Uncertainty of distributed melt rates

We present one best-distributed melt rate estimate that we bound with two extreme cases. These bounds are based on the compounding uncertainty of parameter choices meant to tilt the estimates in the direction of reduced or increased melt; this allows us to test the plausibility of ice cliffs leading to maximum melt within the zone of maximum thinning. In the extreme cases for the debris thickness, curve fits were made through the 25 % and 75 % data points in each elevation bin. We use the interquartile range because the debris thickness within each elevation band is skewed towards values closer to 0, such that a normal distribution is not applicable (Fig. 5; Sect. S3.1). We also apply a $\pm 1$ standard deviation range for sub-debris melt and ice cliff backwasting rates and a $\pm 1$ standard deviation range for ice cliff slopes. Extreme ice cliff coverage was defined by $\pm 20$ % of the bias-corrected coverage within each elevation band. See Table 1 for the extreme parameters used for the distributed melt estimates. With these parameter choices 98.4 % of all simulations lie inside the uncertainty range for combined sub-debris and ice cliff melt.

We also explore five additional uncertainty cases in Sect. S3 of the Supplement. There we extrapolate debris thickness down each medial moraine using linear curve fits, using a single sigmoidal debris thickness–elevation CE7 relationship, using a linear relationship between backwasting and elevation, with even more uncertainties for each curve fit (in which the error envelope includes greater than 99.996 % of possibilities), and with different debris thicknesses for the westernmost medial moraine. All explorations produce similar melt–elevation relationships.

## 3 Results

### 3.1 In situ measurements

Figure 5 shows debris thickness as it varies with elevation. Debris thickness tends to increase downglacier and varies from less than a few millimeters above 700 m a.s.l. to as high as 1 m above an ice cliff at 475 m a.s.l. (Table 2). Debris tends to be thicker in the medial moraines near the glacier margin, especially where ice margin retreat has been small (Figs. 4 and S14). On the east side of the study area, in medial moraine 3, debris greater than 40 cm thick was measured. Debris consistently 1 m thick was observed at 730 m a.s.l. just to the west of the study area in moraine 9. Toward the glacier interior and between 650 and 700 m a.s.l. debris thickness did not exceed 15 cm. While we did not measure debris thickness below 450 m a.s.l., visual inspection from across the proglacial lake suggests that debris exceeded 1 m above some ice cliffs. The mean uncertainty of our debris thickness measurements is $\pm 1.3$ cm, and the standard deviation is $\pm 1.4$ cm (Fig. S4). These errors are negligible compared to the changes in measured debris thickness across the study area (Fig. 5).

Figure 6 shows the relationship between sub-debris melt rate and debris thickness (or Østrem's curve) during the study period (Table 2). Melt rates are highly variable beneath debris less than 3 cm. The mean uncertainty in the sub-debris melt rates is $\pm 0.06$ cm d$^{-1}$ and the standard deviation is 0.03 cm d$^{-1}$. The maximum uncertainty is 0.125 cm d$^{-1}$ and applies to three ablation stakes from which measurements were taken over a short 8 d period. These measurement uncertainties are negligible compared to the changes in melt rate with debris thickness (Fig. 6).

The mean ice cliff backwasting rate is 7.1 cm d$^{-1}$ and the standard deviation for the full population of measured ice cliffs is 2.5 cm d$^{-1}$. The maximum and minimum measured backwasting rates are 15 and 2.5 cm d$^{-1}$, respectively (Table 2). Figure 7 shows measured backwasting rates as they vary with elevation and aspect. There is no apparent aspect dependence on backwasting rates CE8 and ice cliffs backwasted at similar rates with and without ponds or streams at their base (Fig. 7). The mean backwasting rate uncertainty is $\pm 0.5$ cm d$^{-1}$ (Figs. 7 and S8). Maximum estimated uncertainty is $\pm 1$ cm d$^{-1}$ for 10 cliffs that were measured over the shortest interval (21 d). The standard deviation of uncertainty is $\pm 0.2$ cm d$^{-1}$.

**Table 2.** Statistics of debris- and melt-related in situ measurements for Kennicott Glacier.

| Measured variable | Mean | SD | Minimum | Maximum |
|---|---|---|---|---|
| Debris thickness [cm] | 13.7 | 13.9 | 0 | 100 |
| Sub-debris ablation [cm d$^{-1}$] | 4.0 | 1.8 | 0.8 (37 cm of debris) | 7.3 (1 cm of debris) |
| Ice cliff backwasting [cm d$^{-1}$] | 7.1 | 2.5 | 2.8 | 13.8 |

### 3.2 Remotely sensed ice cliff extent

#### 3.2.1 Performance of automatic ice cliff delineation methods

The adaptive binary threshold (ABT) method outperforms the Sobel edge delineation (SED) method. Averaged across the validation dataset, the ABT method correctly identifies 58 % of ice cliff area, with 21 % false positives. Percentages are relative to the hand-delineated validation dataset. The SED method yields a lower percentage of correctly identified ice cliffs (45 %) but also produces fewer false positives (14 %). In regions where we do not have manually digitized ice cliffs, our estimates of ice cliff area represent both true and false positives. Assuming our success rate is consistent across the glacier, we expect the ABT and SED approaches to detect 79 % and 69 % of the true ice cliff area, respectively.

Some systematic errors are evident, as anomalously light and dark regions of the glacier produce higher error. Regions of thin debris are especially problematic when using the SED method (Fig. 9; see also Herreid and Pellicciotti, 2018). To correct for this error in the SED results, where debris is very thin, we manually removed areas with highly erroneous ice cliff delineations; these only occur at higher elevations in the study area (Fig. 9). Due to its poorer performance, we do not use the SED-defined ice cliff area for distributed melt rate estimates.

#### 3.2.2 Spatial distribution of ice cliffs

The two delineation methods produce broadly similar ice cliff distributions. The SED method, specifically, overestimates ice cliff area at high elevation due to the thin, dark-colored debris. Over the 24.2 km$^2$ study area, we estimate that ice cliffs cover 2.14 km$^2$ (8.8 %) and 2.32 km$^2$ (9.7 %) of ice cliff plan view area using the SED and ABT methods, respectively (Fig. 10). We normalized ice cliff area by glacierized area within each elevation band, which we refer to as ice cliff fractional area or coverage. If we apply a bias correction to the SED (31 %) and ABT (21 %) estimates based upon under-delineation rates in manually digitized areas, the ice cliffs cover 11.4 % and 11.7 % of the glacier, respectively.

In total, 11.7 % of the debris-covered tongue of Kennicott Glacier is occupied by ice cliffs (see Anderson (2014) for an independent, consistent estimate of ice cliff coverage). Focusing on the ABT results, which provide the most accurate estimate, we find a "humped" profile in the elevational dis-

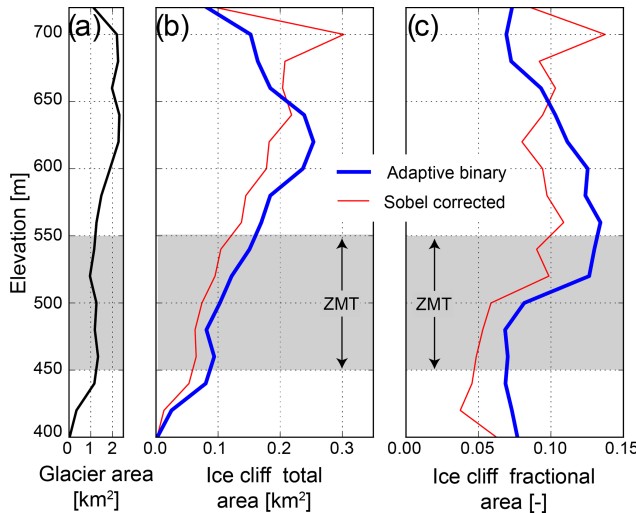

**Figure 10.** Results from the two ice cliff delineation methods with elevation. All panels use 20 m elevation bins. **(a)** Glacier area as a function of elevation. **(b)** Ice cliff area as a function of elevation. The red line shows results from the SED approach after false positives on dark-colored ice are removed. **(c)** Ice cliff area as a function of elevation, normalized by the glacier area within each elevation band. Note that fractional area ×100 is the percentage of ice cliff coverage.

tribution of ice cliff area (Fig. 10). Ice cliff fractional area is relatively uniform at 7 %–8 % except for a broad peak between 500–660 m a.s.l. within which fractional area reaches 13 % between 540 and 560 m.

### 3.3 Distributed estimates of melt

In Fig. 11, we show the best distributed melt estimate split into sub-debris and ice cliff contributions across the study area. When averaged across the entire study area, 74 % of melt is derived from sub-debris melt and 26 (with extreme bounds of 20, 40) % from ice cliff melt.

Figure 12 shows that the insulating effect of debris is more important in setting the area-averaged melt rate than ice cliffs, especially where debris is thinner. Modeled bare ice melt rates, which are meant to represent the hypothetical melt rate if debris were absent from the study area, increase towards lower elevations and range from 5.9 to 7 cm d$^{-1}$. Decreasing sub-debris melt downglacier, due to thickening debris, results in a deviation from the bare-ice melt rate below

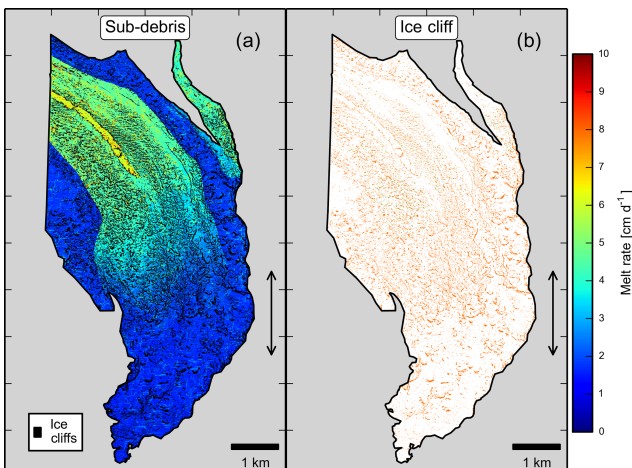

**Figure 11.** Distributed melt rates based on elevation and flow path (medial moraines). The zone of maximum thinning (ZMT) is defined by the double-headed arrows in each panel. **(a)** The best sub-debris melt rate estimate which decreases in magnitude downglacier in the central part of the glacier. Medial moraines near the edge of Kennicott Glacier were composed of thicker debris. **(b)** The best ice cliff backwasting rates which we assume are uniformly distributed across the study area with a value of $7.1\,\mathrm{cm\,d^{-1}}$. Note that no clear trends were present in ice cliff backwasting rate from medial moraine to medial moraine so the same backwasting–elevation CE9 relationship is applied across the study area (Fig. S8).

700 m a.s.l. Area-averaged sub-debris melt rates decline from $4.2\,\mathrm{cm\,d^{-1}}$ (3.2, 5.1) at the top of the study area to $1.6\,\mathrm{cm\,d^{-1}}$ (0.98, 2.0) near the terminus.

Ice cliffs, when their total melt contribution is averaged over the elevation bands, produce rates of $0.73\,\mathrm{cm\,d^{-1}}$ (0.31, 1.29) at the top of the study area and $0.69\,\mathrm{cm\,d^{-1}}$ (0.33, 1.4) near the terminus. The maximum contribution of ice cliffs to area-averaged melt occurs at 510 m and has a value of $1.3\,\mathrm{cm\,d^{-1}}$ (0.58, 2.4), close to where the ice cliff fractional area also maximizes. Ice cliffs between 500 and 520 m a.s.l. generate the highest percentage (42 % (34, 58 %)) of the total melt due to ice cliffs and sub-debris melt within the study area.

## 4 Discussion

We discuss the implications of our in situ mass balance measurements, our new automatic ice cliff delineation method, and finally our distributed melt estimates as they relate to the zone of maximum thinning.

### 4.1 In situ measurements

#### 4.1.1 Sub-debris melt rates

Our measured sub-debris melt rates are highly variable beneath debris less than 3 cm (Fig. 5). It appears that local

meteorology and/or surface hydrology are important controls on the melt-increasing effect of thin debris (see Mihalcea et al., 2006; Reid and Brock, 2010, for similar observations). Our sub-debris melt rates support the observations of Fyffe et al. (2020): there is no consistent melt enhancement under debris less than 3 cm. Debris typically forms parabolic-shaped medial moraines in cross section (Anderson, 2000), suggesting that the melt-reducing effect of debris dominates in the study area (and upglacier as well). Despite the scatter of melt rates under thin debris, the question remains: under what conditions does thin debris increase area-averaged melt rates relative to adjacent bare-ice melt rates?

Based on our debris thickness and sub-debris melt measurements, the characteristic debris thickness ($h_*$) was 8.17 cm. The relationship between melt rate and debris thickness from Kennicott Glacier is similar to those derived from other debris-covered glaciers at subpolar latitudes (Fig. S6). The consistent decline in sub-debris melt rates as debris thickens is not unexpected considering that the global mean value of $h_*$ is $6.6 \pm 2.9\,\mathrm{cm}$ (1 standard deviation; Anderson and Anderson, 2016).

#### 4.1.2 Ice cliff backwasting rates

The backwasting rates presented here are the first published from a debris-covered glacier outside of Eurasia, that the authors are aware of. Despite filling a new geographical niche, the average backwasting rates from Kennicott Glacier are similar to those from high-altitude Eurasian glaciers at lower latitudes with thicker debris cover (Table 3). The similarity in backwasting rates suggests that there are compensating effects between altitude, latitude, and day length. The backwasting rate data presented here are important for validating future regional and global mass balance estimates incorporating the effects of debris cover and ice cliffs.

Backwasting rate measurements were taken from 60 ice cliffs that varied with elevation, orientation, adjacent debris thickness, debris composition, and connection with ponds and streams (Figs. 7, S8). It is logical to expect that backwasting rates would be higher at lower elevations where more energy is available for melt, but significant scatter limits the clear establishment of a relationship with elevation, noting that a weak increase in backwasting rate is apparent towards low elevation when data are binned in 50 m elevation bands (Fig. S7). Measured backwasting rates do not consistently vary with orientation (Fig. 7). This observation contrasts with observations from lower-latitude debris-covered glaciers (Sakai et al., 2002; Buri and Pellicciotti, 2018), suggesting that there may be a latitudinal control on backwasting rates as they vary with orientation. Noting the small sample size, we also found that undercutting ponds ($n = 4$) and streams ($n = 8$) at the base of ice cliffs did not consistently increase backwasting rates (Fig. 7), though ponds may allow for the long-term persistence of ice cliffs (Brun et al., 2016; Miles et al., 2016). The scatter in our backwasting rate

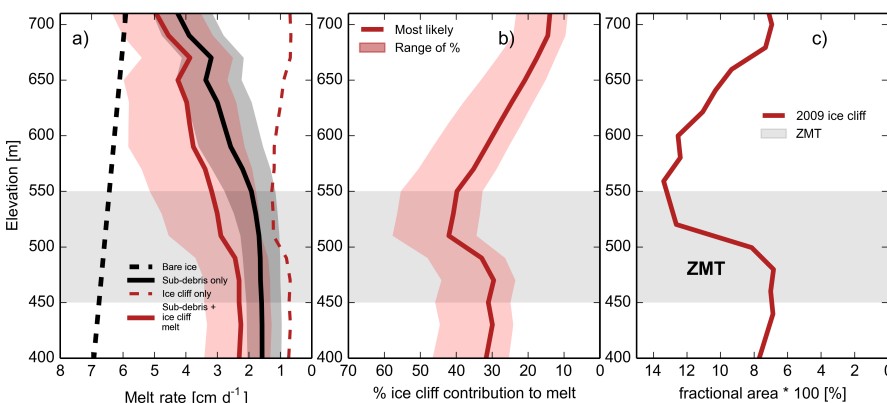

**Figure 12.** Distributed melt rate estimates with elevation. The zone of maximum thinning (ZMT) is represented by the grey bands for all panels. All panels use 20 m elevation bins. Elevations are relative to the 2013 glacier surface. **(a)** The elevation-band-averaged (*absolute*) melt rate over the study period. The red band contains an extreme range of sub-debris plus ice cliff melt based on compounding parameter choices such that 98.4 % of estimates lie within it (see Sect. 2.3.1). A total of 84.1 % of estimates for sub-debris melt are within the grey shaded band. Five additional distributed melt rate scenarios are presented in Sect. S3. Bare-ice estimates are based on the near-surface air temperature lapse rate from off-glacier meteorological stations and a degree-day factor for bare-ice melt (Sect. S1.6). The decrease in sub-debris melt rate at 670 m a.s.l. is related to the increased area of medial moraine no. 9 within the study area, which is covered with relatively thick debris. **(b)** The fractional (*relative*) contribution of ice cliffs to the area-averaged melt rate (sub-debris + ice cliff) with elevation. The red band contains the extreme range of melt contributions from ice cliffs. **(c)** The fractional area ×100 (%) coverage of ice cliffs.

**Table 3.** Comparison of ice cliff backwasting rates and debris thicknesses with other glaciers.

| Glacier | Region | Latitude [deg.] | Mean study area elevation [m] | Range of backwasting rates [cm d$^{-1}$] | Mean debris thickness [cm] | Reference |
|---|---|---|---|---|---|---|
| Kennicott | Alaska | 61 | 600 | 3–15 | 14 | This study |
| Miage | Alps, Italy | 46 | 2200 | 6.1–7.5 | 26 | Reid and Brock (2014) |
| Koxkar | Tien Shan, China | 42 | 3500 | 3–10 | 53 | Han et al. (2010); Juen et al. (2014) |
| Lirung | Himalayas, Nepal | 28 | 4200 | 7–11 | 50–100 | Brun et al. (2016) |
| Changri Nup | Himalayas, Nepal | 28 | 5400 | 2.2–4.5 | – | Brun et al. (2018) |

\* Sorted by latitude.

measurements precludes a clear establishment of cause and effect. The scatter is likely at least partially related to local topography and shading (Steiner et al., 2015), a control we do not explicitly consider here. Further field efforts with an even larger population of ice cliffs would allow for statistical analyses that reveal spatial controls on ice cliff backwasting rate.

We take an in situ measurement-based approach to quantify ice cliff backwasting rates. We assume that single measurements taken from the top of 60 ice cliffs represent the mean backwasting rate across the thousands of ice cliffs on Kennicott Glacier. It is tempting to turn towards process-based models of ice cliff backwasting rates, but modeling complicated processes necessitates a large number of free parameters. Most model parameters vary in unknown ways across debris-covered glacier surfaces. The best way to reduce parameter uncertainty and validate model results is to simply make more in situ measurements. Whether you model

ice cliff backwasting or follow a more empirical approach as we have here, the validity of the conclusions rests on the number and quality of measurements.

## 4.2 Remotely sensed ice cliff extent

### 4.2.1 Automatic ice cliff delineation methods

The adaptive binary threshold (ABT) method provides an especially accurate estimate of ice cliff area as it varies across a large debris-covered area. Both the ABT and SED ice cliff delineation methods underpredict ice cliff area somewhat. These methods require that ice cliffs are dark relative to surrounding debris cover, which is generally true for Kennicott and several other debris-covered glaciers in the Himalayas we examined. Ice cliffs may be brighter than the surrounding debris if the ice cliffs are not covered with thin debris films or if they are strongly illuminated. The ABT method will there-

fore tend to underpredict south-facing ice cliffs, although we observe many correct delineations.

The ABT approach is a promising, novel method for the large-scale delineation of ice cliffs. Because of the high accuracy of the method, its transferability to other glaciers should be tested using the parameters already tuned in this study and with new parameters tuned for other glaciers. Future improvements to the ABT method could be made by applying more advanced image segmentation techniques (e.g., Leyk and Boesch, 2010), by utilizing image texture analysis, or by allowing image processing parameters to adaptively vary across the glacier. Using multispectral imagery would also likely improve delineation, although such imagery is less readily available.

### 4.2.2 Spatial distribution of ice cliffs

The 11.7 % ice cliff coverage in the debris-covered tongue of Kennicott Glacier is the highest coverage from any glacier studied to date. The 11.7 % coverage is 60 % more coverage by percentage than the debris-covered portion of Changri Nup Glacier, the glacier with the second highest ice cliff coverage (Brun et al., 2018; Table 4). The debris-covered portion of Changri Nup Glacier is also considerably smaller in area ($1.5 \, \text{km}^2$) than the debris-covered tongue of Kennicott Glacier ($24.2 \, \text{km}^2$). Kennicott Glacier has the lowest mean debris thickness (13.7 cm) of glaciers with reported ice cliff coverage percentages and supports, by far, the highest percentage of ice cliffs. This implies that ice cliff coverage could vary with debris thickness or a variable that co-varies with debris thickness (e.g., debris mobility; Moore, 2018).

### 4.3 Distributed melt estimates

Our distributed melt rate estimates include potential slight biases towards higher melt rates. A total of 53 % of our debris thickness measurements were derived from the top of ice cliffs and topographic highs. Because debris tends to concentrate in topographic lows, our debris thickness measurements may be biased toward thinner debris and higher melt. Our measured ice cliff backwasting rates are based on repeated measurements at a single location at the top of each ice cliff. Maximum backwasting rates across each ice cliff are more likely to occur near the top (Buri et al., 2016; modeled from Lirung Glacier, Nepal). Applying our measurements across single ice cliffs or the entire ice cliff population may therefore also overestimate ice cliff melt.

On Kennicott Glacier, ice cliffs most likely contribute 26 % (with extreme bounds of 20 % and 40 %) of melt in the study area. For glaciers with mean debris thicknesses larger than 50 cm, where sub-debris melt rates are low, ice cliff relative contributions are larger than 26 % and as high as 40 %, despite having much lower ice cliff fractional coverage than Kennicott Glacier. This relationship holds when comparing individual debris-covered glaciers (Table 4) and as

debris thickness increases downglacier on Kennicott Glacier (Fig. 12b). Ice cliffs are *relatively* more important for mass loss the thicker the debris cover.

The debris-covered tongue of Kennicott Glacier provides an opportunity to test the importance of ice cliffs on debris-covered glacier mass balance. The thin debris leads to melt rates closer to bare-ice melt rates than most other studied debris-covered glaciers. Ice cliff backwasting rates are comparable to or higher than rates from other glaciers (Table 3). Kennicott Glacier also has the highest fractional coverage of ice cliffs, relative to other studied glaciers, which also serves to increase melt rates.

Despite this, ice cliffs on Kennicott Glacier do not compensate for the *absolute* melt-reducing effects of debris. Area-averaged melt rates, including ice cliff contributions through the study area, are lower than hypothetical bare-ice melt rates at the same elevation (Fig. 12a). Ice cliffs are therefore unlikely to counter the melt-reducing effects of debris on glaciers with thicker debris and/or lower ice cliff coverage.

The analysis above leads to the expectation that absolute area-averaged rates on debris-covered glaciers will tend to decline downglacier as debris thickens, an inference that is further supported by the analysis of Bisset et al. (2020) from selected glaciers across high-mountain Asia. Future efforts to represent the effect of ice cliffs on debris-covered glacier mass balance should consider using a modified debris thickness–melt relationship with a percentage melt enhancement based on remotely sensed ice cliff coverage and empirical relationships like those developed in this study.

### 4.3.1 Sensitivity tests: do ice cliffs maximize melt in the zone of maximum thinning (ZMT)?

We explore what hypothetical perturbations would be needed to produce the highest glacier-wide melt rates where the glacier has thinned the most. During the study period (mid-June and mid-August of 2011), the melt within the zone of maximum thinning (ZMT) was strongly reduced by debris cover. We assume that the ZMT – which was stable from 1957 to 2004 and 2000 to 2007 – remained in the same location during the summer of 2011. The ZMT was debris covered from at least 1957 to the present (Fig. S21). Ultimately, these sensitivity analyses show how extreme the parameter choices would need to be to maximize melt in the ZMT.

*Debris cover and sub-debris melt.* Debris thickness would have to decrease, specifically in the ZMT, from $\sim 20$ to 2 cm to produce maximum glacier-wide melt rates there. For melt to be maximized in the ZMT, where debris is $\sim 20$ cm thick, sub-debris melt rates would have to increase from $\sim 1.6 \, \text{cm} \, \text{d}^{-1}$ by a factor of 3 to $4.8 \, \text{cm} \, \text{d}^{-1}$. Our distributed melt-estimation approach assumes that small-scale debris thickness variability has a negligible effect on area-averaged melt rates, despite the non-linear debris–melt rate relationship. The sensitivity test in this paragraph reveals how

https://doi.org/10.5194/tc-15-1-2021                    The Cryosphere, 15, 1–18, 2021

**Table 4.** Comparison of ice cliff coverage and melt contribution with other debris-covered glaciers.

| Glacier | Region | Glacier area [km$^2$] | Study area [km$^2$] | Ice cliff fractional** area (%) | Ice cliff mass loss (%) | Mean debris thickness [cm] | Study |
|---|---|---|---|---|---|---|---|
| Ngozumpa | Nepal | 79.5 | 17.4 | 5 | 40*** | 0–300 | Thompson et al. (2016) |
| Lirung | Nepal | 5.8 | 1.1 | 2.0 | 36 | 50–100 | Buri and Pellicciotti (2018) |
| Kennicott | Alaska | 387 | 24.2 | 11.7 | 26 ($\pm$8) | 13 | This study |
| Changri Nup | Nepal | 2.7 | 1.5 | 7.4 | 24 ($\pm$5) | – | Brun et al. (2018) |
| Langtang | Nepal | 40.2 | 15.4 | 1.3 | 20 | – | Buri and Pellicciotti (2018) |
| Koxkar | China | 84 | 15.6 | 1.4 | 7.4-12 | 33 | Han et al. (2010); Juen et al. (2014) |
| Miage | Italy | 11 | 3.1 | 1.3 | 7.4 | 26 | Reid and Brock (2014) |

* Glaciers sorted by mass loss percent due to ice cliffs.
* Percent relative to each study area. CE10
** Combined contribution from ice cliffs, ponds, and streams.

improbable it is for small-scale debris variability to lead to maximum melt rates in the ZMT (see Sect. S3).

*Ice cliffs (and other melt hotspots).* In order for ice cliffs to increase melt and produce maximum glacier-wide melt rates in the ZMT, absolute backwasting rates would need to be 6.5 times higher than those measured in the summer of 2011. The hypothetical backwasting rates required to maximize melt in the ZMT are unrealistic; a compilation of previously published backwasting rates in Table 3 supports this. We implicitly assume that the peak melt season (mid-June to mid-August) is a good proxy for annual-average ablation rates. It is unlikely that this assumption affects our conclusions. In order for absolute annual area-averaged ablation rates to be maximized in the ZMT, ice cliff backwasting rates in shoulder seasons, specifically in the ZMT, would need to be 6.5 times those measured in the summer of 2011. These conditions would need to persist for at least 2 months outside of the peak melt season, despite reduced availability of energy for melt in these shoulder seasons.

While we do not explicitly document the melt rate of ponds and streams, we follow the approach of Kraaijenbrink et al. (2017) and assume that all melt hotspots melt at the same rate as ice cliffs. Using this assumption, in order for melt hotspots to compensate for the melt-reducing effects of debris in the ZMT, melt hotspots would need to cover 90 % of the glacier surface, specifically in the ZMT. Assuredly, this is not the case.

### 4.4 Importance of upglacier melt and ice dynamics

To consider what controls the ZMT, we return to the continuity equation for ice (Eq. 1). If we fail to account for the movement of ice, then local surface mass balance is the only factor that can cause ice thickness change, and the continuity equation reduces to

$$\frac{dH}{dt}(x) = \dot{b}(x) - \nabla \cdot Q(x) = \dot{b}(x) - 0. \tag{10}$$

If this equation were valid across Kennicott Glacier, then the zone of maximum thinning would align with the region of maximum area-averaged surface melt rates. However, melt is not maximized in the ZMT (Fig. 12). Previous studies have shown that ice is in motion in and above the ZMT (Armstrong et al., 2016, 2017). Equation (10) TS3 therefore cannot be applied to explain the location of the ZMT for Kennicott Glacier. The movement of ice (i.e., *ice dynamics*) down valley must play a role.

We now consider the scenario in which increased melt upglacier from the ZMT has led to dynamic thinning in the ZMT (also see Nye, 1960; Vincent et al., 2016). It is feasible that upglacier from the ZMT increased melt rates have reduced ice thicknesses through time, which in turn led to a reduction in ice speeds. Thinner ice and lower speeds upglacier from the ZMT reduce the volume of ice delivered per time unit CE13 ($Q$, ice discharge) to the ZMT. This scenario results in a downglacier gradient of ice discharge, d$Q$/d$x$, that is closer to zero across the ZMT. A d$Q$/d$x$ declining towards zero through time would in turn lower the ice emergence velocity, causing rapid surface lowering and thinning in the ZMT.

To advance our understanding of why rapid thinning occurs under melt-reducing debris cover, we must consider both terms in the continuity equation for ice (Eq. 1) and how they affect one another. We must also expand our perspective and consider the entirety of glaciers, including the debris-free portions upglacier from the debris cover.

### 5 Conclusions

Using novel methods, the spatial distribution of melt rate on a debris-covered tongue in Alaska has been quantified for the first time. We collected abundant in situ measurements on Kennicott Glacier allowing for the extrapolation of debris thickness, sub-debris melt rates, and ice cliff backwasting rates across the 24.2 km$^2$ study area. Debris thicknesses are extrapolated downflow units, as defined by medial moraines.

A newly developed automatic ice cliff delineation method is the first of its kind to use only high-resolution satellite imagery. The adaptive binary threshold (ABT) method robustly estimates ice cliff coverage for a particularly difficult test case (Kennicott Glacier) in which ice cliffs are abundant and often small. The method performs well even as debris color varies across nine medial moraines. With further testing, the ABT method could be applied efficiently across numerous glaciers.

Kennicott Glacier is the largest debris-covered glacier for which distributed melt has been rigorously quantified. Kennicott Glacier also exhibits the highest fractional coverage of ice cliffs documented on a debris-covered glacier (11.7 %), yet ice cliffs contribute only modestly to the average melt rate across the glacier tongue (26 %). Ice cliffs contribute a larger percentage of melt in areas where debris cover is thick, mirroring results from other studied glaciers in Eurasia. Despite this increasing *relative* importance of ice cliffs as debris thickens (Fig. 12b), the area-averaged absolute melt rates – which actually control glacier thinning and meltwater production – decline towards the terminus (Fig. 12a). While ice cliffs should not be neglected, our analysis suggests that increased attention be given to debris cover and how it varies across individual glaciers and regions.

The debris-covered tongue of Kennicott Glacier provides an opportunity to test the importance of melt hotspots for debris-covered glacier mass balance and thinning. Thin debris, high ice cliff backwasting rates, and abundant ice cliffs all compound to increase the likelihood that glacier-wide melt rates peak within the debris-covered tongue of Kennicott Glacier. The zone of glacier-wide maximum thinning (ZMT) is in a debris-covered, stable location upglacier from the terminus. However, even with extreme uncertainty scenarios, melt rates neither match hypothetical bare-ice melt rates nor result in glacier-wide maximum melt rates in the ZMT. We conclude that the reduction of ice discharge from upglacier is necessary to explain the rapid glacier thinning occurring beneath thick debris at Kennicott Glacier.

*Data availability.* Datasets are openly available at https://doi.org/10.5281/zenodo.4118672 (Anderson et al., 2020).

*Supplement.* The supplement related to this article is available online at: https://doi.org/10.5194/tc-15-1-2021-supplement.

*Author contributions.* LSA, WHA, and RSA designed the study. LSA composed the manuscript, collected all field data, and completed all analyses besides developing the automatic ice cliff delineation method. WHA developed the ice cliff delineation method, delineated ice cliffs, and wrote the associated text. RSA advised LSA and WHA through the study and contributed to the text and figures. PB added important discussion that improved the manuscript. All authors aided in composing the manuscript.

*Competing interests.* The authors declare that they have no conflict of interest.

*Acknowledgements.* We thank Craig Anderson and Emily Longano for invaluable field support. We thank Dirk Scherler, Indrani Das, Regine Hock, Martin Truffer, Jack Holt, Eric Peterson, Brandon Tober, Katy Barnhart, Andy Wickert, and Eric Winchell for thoughtful discussions. LSA thanks the organizers and participants of the 2010 Glaciological Summer School held in McCarthy, Alaska, which inspired this research. We thank Per Jenssen, Susan Fison, Ben Hudson, Patrick Tomco, Rommel Zulueta, the Wrangell-St. Elias Interpretive Rangers, the Wrangell Mountains Center, and Ted Scambos (NSIDC) for logistical support and the gracious loan of equipment. We thank Lucy Tyrell for facilitating outreach efforts. We also thank Joshua Scott, Wrangell-St Elias National Park, and the Polar Geospatial Center for access to satellite imagery as well as Etienne Berthier for sharing DEMs. WHA thanks Waleed Abdalati, Mahsa Mousavi, and Stefan Leyk for guidance in image processing.

An earlier version of this manuscript benefited from a review by David Rounce.

*Financial support.* Leif S. Anderson received support from a 2011 Muire Science and Learning Center Fellowship, NSF DGE-1144083 (GRFP), and funding from the European Research Council (ERC) under the European Union's Horizon 2020 research and innovation program under grant agreement no. 759639. Robert S. Anderson and William H. Armstrong received support of NSF EAR-1239281 (Boulder Creek CZO) and NSF EAR-1123855. William H. Armstrong received support from NSF OPP-1821002 and the University of Colorado at Boulder's Earth Lab initiative.

The article processing charges for this open-access publication were covered by a Research Centre of the Helmholtz Association.

*Review statement.* This paper was edited by Francesca Pellicciotti and reviewed by Evan Miles and one anonymous referee.

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

**Remarks from the language copy-editor**

CE1    This is not an official proper name and is thus not capitalized.

CE2    If "hyper" is modifying "fit", then this is correct as is. Furthermore, as this is also not an official proper name, it is also not capitalized.

CE3    This is an en dash, which is used to denote "and" or a relationship between two or more things. If you are talking about the relationship between thickness and melt, then this is correct as is.

CE4    The suggested change is ungrammatical and will not be inserted.

CE5    The suggested change is ungrammatical and will not be inserted.

CE6    Again, this does not seem to be a proper name. If it is, please provide multiple published scientific sources in which it appears as such.

CE7    See CE3.

CE8    The suggested change is ungrammatical and will not be inserted.

CE9    See CE3.

CE10    It is our house standard to separate notes beneath tables as individual sentences and thus use sentence-style capitalization.

CE11    It is our house standard to only italicize terms at their first instance in the abstract and main text as well as figure captions. This will stay as is throughout.

CE12    If you are referring to the relationship between debris and melt rate, then this is correct as is.

CE13    Note "unit" has been added since, as already explained, "per time" makes no sense in English. This addition does not restrict the time interval but still makes semantic and grammatical sense.

**Remarks from the typesetter**

TS1    Please give an explanation of why this needs to be changed. We have to ask the handling editor for approval. Thanks.

TS2    Yes, number with 5 digits have a space. Is 15 0 0 0 correct here?

TS3    Yes, it is our standard to show equation numbers like this.