# Peer review of "Debris cover and the thinning of Kennicott Glacier, Alaska: in situ measurements, automated ice cliff delineation and distributed melt estimates"

_The Cryosphere, 2019_

## Referee Comment (RC1) · Anonymous Referee #1 · 18 Oct 2019

Review of Anderson et al., part B, The Cryosphere, October 2019

In this second opus of their trilogy, Anderson et al. deduced the spatial pattern of melt due to ice cliff and under debris, and consider the distribution of supraglacial lakes to conclude that melt hot spots (cliffs and lakes) are not sufficient to explain the pattern of rapid thinning on Kennicott Glacier.

Overall this is a series of paper that bring a lot of new data and contribute to show that melt hot spots (ice cliff and lakes) only modestly contribute to the overall mass loss of a large debris covered tongue. A clear achievement has been to perform such measurements on a very large glacier in Alaska and proposed methods to extrapolate

the point wise measurements to the overall debris-covered glacier tongue.

General comments for the three papers.

1/ I am not convinced by the need to split this paper into three parts. It implies lot of repetitions and also mean that the reader as to refer to other parts of the article which is not convenient. Some data are plot several times in the three articles (debris thickness, dh/dt for 1957-2009 etc...) I think the authors missed here an opportunity to put everything together. Specifically in this part B, the discussion (section 4.2.1) whether ice cliff or debris can explain the zone of maximum thinning would be much more straightforward if Part B and C were merged. Right now this discussion is a lot of speculation to finally justify the need for a part C.

2/ One strong limitation (that needs to be emphasized more) is that field measurements over a short period of time in July 2011 are used to interpret a map of elevation change measured over a multidecadal time period. Authors need to recall to their reader that their results apply to 2-month period in summer. The whole discussion would have been much more meaningful if the elevation changes were also measured for the same time period where surface melt features are studied (but the DEM data are probably not available...).

General comments for part B.

3/ I miss a more thorough description of and comparison to earlier studies mapping ice cliff automatically. In particular Kraaijenbrink et al., RSE, 2016.

4/ I feel it would have been very interesting to see an evaluation of the ice cliff mapping algorithms using independent dataset, for example the Ragletti/Steiner cliff dataset on Lantang Glacier. Maybe this ice cliff automatic mapping part would have deserved a dedicated article, and all the rest of the results would then fit a single contribution?

5/ Uncertainties could be treated in a more systematic way so that results in the end should all be quoted together with their range of uncertainties. This applies to all three
parts.

6/ When authors provide % of melt, they should always make it clear that this is a percentage of the debris-covered tongue (and not the whole glacier!)

Specific comments.

L16 What does "enhancing" the mass balance mean. A mass balance can be increased or reduced. Is this formally demonstrated? I thought it was debated.

L21 "Total" is ambiguous. Tongue-wide or glacier-wide?

L41 One does not expect results in the introduction.

L51 "surface mass balance" would be a more appropriate way to refer to it

Eq 1. x,y are not defined.

L58. So do the authors neglect them? It should be stated unambiguously.

L88ff. Splitting the article into three parts leads to many repetitions such as this section. Problematic in my view.

L160. Unclear (understated) what meteorological data would have brought, if they had been available.

L169-170. This statement that 20% of ice cliff area need to be added is enigmatic at this stage in the paper.

Eq (3). How the type of fitting curve was chosen? It seems to come from nowhere. Can it be justified?

L183. I see in part A that your cliff backwasting rate neglect emergence velocity. This needs to be justified.

L191. A statement such as (here) "based on an analysis of 2-m ArcticDEMs" is too vague.
eq (7). Ice ciff area. Is it planar or real area? I think "i" must be added as superscript with bˆdot debris and bˆdot icecliff

L204. I do not understand why fitting a curve through 25% or 75% of the data points leads to "extreme" cases. Not clear. Why not a curve containing 67% of the data (to have 1-sigma uncertainties). See my general comment about treatment of error bars.

L226. "error checks" is a strange terminology. Why not "validation dataset"

L240. Percentage should be 21% and 31%, right?

L244. Where does "11.6%" come from? I read 11.4% and 11.7% above.

L245. This raise the question of whether all studies defined the "debris-covered tongue" the same way. Did the authors check carefully previous studies for this aspect?

L247. "This implies that ice cliff coverage varies with debris thickness". This seems like a hasty conclusion. . . other example from the literature to support the statement?

L257. One expect an error quantification for each term (81% and 19%).

L268. "Across all of the elevation bands, the ice cliffs between 500 and 520 m generate a maximum of 40% of the total mass loss due to ice cliffs and sub-debris melt." I am not sure I got the meaning here. Maybe reformulate for clarification. (it is clear from the figure, just a text improvement)

L273 "within" rather than "with" (I think)

L289. 19%. Lack error bars and also authors need to remind that this applies to a short period of time during summer 2011. So they cannot draw such broad conclusion. I would be curious to see a comparison of this number to the total glacier-wide ablation during this period if available. Is not it just a few percents? Do we need to really worry so much about ice cliffs for glacier-wide or region-wide application (and future projections)?

L307. the SMB cannot be "suppressed". It can be increase or decrease. (SMB is increased here, or less negative)

L321. "This required backwasting rate is well beyond potential biases introduced due to the summer of 2011 having anomalously low air temperatures". Statement not really explained and justified.

L324. Is this potential overestimation from the sampling strategy (at top of cliffs) included in the error bars, as it should?

L334. "mass loss" should be replaced by "melt rate" here.

L344. I did not get the point here.

L349. The wording suggests that 11.7 % of the glacier is covered with cliffs. No. This is the % of the debris-covered tongue.

Table2. For the ice cliff backwasting parameter f, the most likely value is not contained by the min/max interval. A typo? Or a real error? For the ice cliff area the most likely and max values are equal. This is also not really expected neither.

Table 3. Ice cliff fractional area, a percentage of what total area?

L415. I do not understand this note.

Figure 2. Are these data from Das et al? Did they use GDEM V2? This would be problematic because it has no defined time stamp. Explain the 1957 and 2015 grey boxes also.

Figure 4. Multiple reference to part A complicate the reading. 25 and 50% or 25 and 75? Is it "elevation bins"? Panel B. Why the order of values in both axis are reversed. Why not showing the Ostrem way? Authors could refer (in the article, not here) to a compilation by Kraaijenbrink et al., 2017 in their Nature study.

Figure 5. Can the authors show the location of this small area of the glacier?

Figure 6. Impressive maps.

Figure 8. Showing percentage for panel c (instead of fractional area) would facilitate correspondence with the text.

Figure 9 The sign only make sense if this is referred to as "surface mass balance rate". If the word "melt" is preferred then positive values should be shown.

Figure 10. See comment on Figure 9 for "melt"

Figure 11. Recall the period of dh/dt. In panel b rather than repeating dh/dt authors could show a map with the density / $m^2$ of ice cliff.

[Figure]

---

## Referee Comment (RC2) · Evan Miles (Referee) · 20 Jan 2020

**Review of 'Debris cover and the thinning of Kennicott Glacier, Alaska, Part B' by Leif Anderson et al., under consideration for The Cryosphere**

Part B of the Anderson et al trilogy aims to combine empirical relationships of surface properties and melt rates, based on the field measurements presented in Part A, with remote sensing observations of different surface features (particularly ice cliffs) in order to arrive at distributed estimates of melt rates for the period of observations. The analysis then uses these distributed melt values to address whether or not the identified melt hotspots can explain the thinning patterns evident at Kennicott. This is an important question as the glaciological community is trying to disentangle the influence of surface mass balance and ice dynamics for debris covered glacier evolution, and this is the first time the question has been addressed in Alaska, where surface debris is prevalent.

As such, this represents an important contribution to a current topic of research, and provides an answer to that question for Kennicott Glacier – these hot spots do have an important effect on the surface mass balance, but it is not plausible that they compensate for the overall melt reduction due to surface debris. The study could be more systematic to provide a definitive answer, and I have some criticisms regarding the empirical relationships presented in determining the distributed estimates of melt, as well as the difference in temporal scales between long-term elevation change (52 years) relative to single-year field measurements. However, although I have quite a few comments, no changes along these lines are likely to change the conclusions of the study. Rather, my principal concern is that the separation of this analysis from both Parts A and C reduces the strength and presentation of the entire analysis, while also leading to repetition of text and figures, as well as cross references. There are certainly gaps in the analysis because aspects have been included in Parts A or C rather than here, and some restructuring across the three manuscripts might improve the readability of all three. This is a choice for the authors and Editor to contemplate, but my opinion is that some consolidation would be beneficial.

Main comments:

My principal question in reading this manuscript was whether it should be standalone or integrated with Part A (and possibly C, which I have not reviewed). I appreciate that this is a difficult decision, and that Parts A, B, and C combined represent a substantial body of work. There are certainly some advantages to be considered for maintaining separation between the manuscripts, but my impression is that A and B are both weaker manuscripts separated from one another. The field measurements were clearly collected for the purpose of deriving distributed melt rate estimates, which leaves Part A without a compelling conclusion or discussion. At the same time, Part B requires frequent reference to Part A, or blind faith of the part of the reader with regards to the methods and results of the field data. As a consequence, there is also quite a bit of repeated material to cover (for example, content from 4 figures in Part A is also displayed in Part B). My instinct when reviewing Part A was that much of that material could be more meaningful if integrated with Part B, in supplementary information if not in the main text, and I now think that would greatly improve the readability of this manuscript.

I like the ice cliff delineation method in its simplicity (although some details need to be clarified, below), but its transferability is not very clear. Often when developing/proposing a new method it is necessary to see how robust the method is, but in this case the method has clearly been developed specifically to map cliffs in this particular scene, in order to apply the empirical melt relationship. As such, it is a relatively small part of the story in deriving the distributed melt estimates. At the same time, the maps of supraglacial ponds are not integrated very well into the story, while the exclusion of supraglacial streams (also mapped from satellite imagery) is a bit

The study aims to address the role of melt hotspots in explaining thinning rates. For this to be a definitive analysis in this regard, I feel like this needs to be done in a more systematic manner, whereas the present analysis seems to focus exclusively on ice cliffs. The distribution of supraglacial streams seems to be a major gap in the analysis of this part of the study, especially as the properties of supraglacial streams are assessed in Part C. Streams are mentioned throughout the background as hot spots and possible factors contributing to melt, but are then completely neglected in the methods and discussion. I understand that the role of surface streams cannot be assessed quantitiavely in this manuscript, but neither can the role of ponds. Similarly, although internal ablation is usually regarded as negligible in this type of specific mass balance assessment, there have been suggestions that this is a non-negligible term for extensively debris-covered glaciers. It is exceedingly unlikely that this mechanism could lead to the debris-cover anomaly, but for completeness I think it should be considered numerically along with the cliffs and ponds.

The relationships between elevation and debris thickness, and between elevation and ice cliff backwasting, are very weak. In neither case does elevation appear to be a primary control of the property. How different would your results be if you interpolated spatially (within flowlines or lithological units for example)? Did you consider alternative methods to provide a distributed debris thickness estimate? For example, one could consider the radiance measured by satellite thermal infrared imagery as related to debris thickness, and use your measurements to constrain this relationship locally to upscale in space much more meaningfully. For ice cliffs, justification of a fit to elevation needs to be more explicit in this Part of the study.

The conversion of backwasting rates into ice cliff melt misses a slope correction for the cliff area, which results in an underestimation of melt (see comment on L190). This raises a difficult question that has not been carefully considered yet for debris-covered glaciers, which is that the real surface area can be 10-20% higher than the planimetric area. For this study comparing geodetic thinning observations with estimated melt, that is an important aspect to consider, as melt occurs relative to the real surface area. This effect is especially pronounced for ice cliffs, but is also crucial for the 'background' melt rate if the glacier surface is highly variable.

It is too bad that more recent geodetic difference data were not included in the analysis. At present it is not clear how the long-term thinning rate relates directly to the 2011 observations. Would it be possible to use the ArcticDEM datasets to derive a recent-period thinning pattern? This would be more meaningful for a comparison to the contemporary distribution of surface features. The long-term perspective is still useful for understanding

dynamic changes, but comparing 50-year lowering rates to one-year melt patterns does not provide a definitive answer, especially for a clearly changing system.

Some rewriting is needed for readability and presentation standards in The Cryosphere. Although the ideas are well developed, some sections of the paper read as bullet points and/or the word choice has not been considered carefully, leading to some of my comments below.

Detailed comments:

L23. Presumably this 19% of melt is for the debris-covered area?

L25. Just a comment, for you to adopt or disregard: The literature has tended to use 'ponds' for these features as they are much smaller than supraglacial lakes on, e.g., the Greenland or Antarctic Ice Sheets.

L25. It would be nice to have the %areal coverage numbers here in the abstract, not just 'doubled'.

L27-27. This wording isn't very clear. By average melt rates you seem to mean the average for all surfaces, but as worded this seems to refer only to sub-debris melt. In the latter half of the sentence, do you mean that the overall melt relationship still follows an Ostrem-type relationship, even after accounting for cliffs and ponds?

L34. These are broken sentences. The first half needs a reference even if it is now well understood.

L95. Has this happened progressively in the past 82 years, or primarily over some later period?

L98. Why is 'partially' here?

L108. Which spectral bands of the image do you use?

L113. This is true in certain conditions, depending both on debris lithology and meteorology. Such conditions may be prevalent for Alaskan supraglacial debris and melt seasons, but it is important to think whether such a method is transferable.

L118. This is semantics perhaps, but I would argue that this whole workflow is your 'cliff detection method' (the name of step 2). It is a bit strange to have a 'cliff detection method' as the main step of a 'cliff detection workflow'.

L118. Is the histogram stretch just a linear min/max stretch? Is this histogram stretch applied to the image globally or locally within a patch? What spectral band(s) are used in either approach? I suppose that you also start with a debris outline (and glacier outline), which is important to acknowledge for a remote sensing approach.

L123. Is the saturation stretch part of your step 1, or a separate part of step 2?

L125. I see that the size of the moving window is included later as one of your parameters in the MC optimization. It might be good to give the reader a warning that that is the direction

the methods are going, and that you will start with a description of the implementation of each approach first.

L126-7. I would certainly not say that adaptive thresholding is insensitive to changes in debris cover and illumination, but it may be less sensitive. Ice cliffs are not uniform in surface character, and can appear both brighter and darker than the surrounding topography in different circumstances. It may be that these nuances are not so evident in the lower portion of Kennicott Glacier, but two particular cases would pose a major challenge for the ABT approach: 1) a population of ice cliffs with variable surface character (debris-free vs covered with fines) which will increase the spectral variance of the cliff population; 2) otherwise dark (potentially wet) debris. This is discussed later in the manuscript, but for a presentation of a new method, I think the accuracy and appropriate application needs some further advertisement/warnings.

L133. Some of this content is Results

L142. Was this 3% disagreement in total area? Can you provide a dice score for the two independent outlines? It is very possible to derive largely different cliff distributions but arrive with the same area.

L144-149. If I understand correctly, there are thus 6 parameters for the ABT and 5 for the SED implementations, with 4 shared between the two. How did this occur in practical terms? 2500 runs for each implementation, or 2500 runs used the same values for the shared parameters? More importantly, it is worth noting that with 5 parameters, 2500 runs results in an effective sampling of ~4.8x in each parameter (4.8^5 ~ 2500).

L152. 'The origin' is a bit ambiguous here, as it is true for figure 5, with the x-axis as the negation of the true positive rate. So really this is ranked by distance from (1,0) in your optimization space, correct?

L153. Why did you choose to reduce the FP rate (at the expense of TP) from the optimal parameter set? Can you please provide a dice coefficient for this parameter set for each approach?

L157. Process observation (2) actually refers to melt rates, rather than backwasting rates.

L158. The influence of lakes was noted earlier by Brun et al (2016) and Miles et al (2016), among others.

L163. In Part A it is clear that while elevation is a principal driver of debris thickness variability, there is considerable heterogeneity within any elevation bin. As your field measurements of debris thickness could not encompass the entire study area (that would not have been feasible), do you think they sufficiently characterise the unmeasured area (particularly the NW of the domain)? Have you tested the importance of debris thickness heterogeneity in your overall melt estimates? The subdebris melt relationship is not linear, so melt calculated with a mean thickness may not accurately approximate the mean melt rate.

L167. It is worth noting that you neglect internal ablation as well as other thermokarst processes (ponds, streams) in this computation for practical reasons.

L176. Please provide a goodness-of-fit for this empirical equation.

L180. It is interesting that as formulated, $b_{ice}$ is the measured clean ice melt rate near the top of the study area, rather than the lapsed melt rate for each debris point. This is much more practical, but ignores the real melt suppression by the debris as a shortcut to a rate. In any case, I presume that the equation (as in Anderson and Anderson 2016) is based on the measurements presented in Part A? Please provide a goodness-of-fit measure.

L185. I am sceptical of this linear fit given the spread of observations in Part A – a goodness of fit would be expected to be very low. If elevation is a secondary control, what might you presume is a primary control for the difference in backwasting rates?

L187. The similarity of backwasting rates for cliffs with/without lakes may be due to the observation type and period. Ponds and streams tend to incise thermoerosional notches, which can later collapse, thus enhancing the seasonal mass losses but not affecting what one would observe from the top of the cliff over a month or two. This is not a criticism of your work, it is just worth noting that this nonobservation doesn't mean a melt enhancement is not occurring.

L190. The correction in backwasting rates for cliff slope is correct, but it is also necessary to correct the cliff area from planimetric to surface area in order to correctly estimate melt from these inclined features, as melt occurs perpendicular to the surface. Thus

$$A_i = A/\cos(\theta)$$

With theta=40 degrees, this is a factor of 1.3 to all of the cliff-related melt calculations.

L192. I think it's reasonable to apply a constant slope for the melt calculation, but why is this the 'most likely case'? Can you please provide some supporting information as supplemental figures, etc?

L201. 'Most likely' is superfluous here; it is an estimate. It's nice that you provide bounds!

L202. This 'best estimate' is using the parameter values already given in the text, correct?

L213. I suppose you use the lapse rate (per timestamp? Hourly? Daily? Mean?) between the two stations for this estimation? It is notable that this lapse rate approach corresponded poorly to your on-glacier temperature observations (in the debris-covered area, Part A) – how do you think such an approach would correspond to the on-ice calculations?

L218. Did you attempt to digitize ice cliffs in 1957 as well? The mention of ponds (and long-term change) is quite sudden, and should maybe be better integrated with the text.

L219. 'insure' should be 'ensure'

L243. Should this be between 520 and 620 m? The fractional area is more meaningful than total area for understanding the cliff distribution.

L245. Importantly, this is of reported values.

L247. N is too small to note any meaningful correlation between debris thickness and ice cliff coverage, unless you have a physical mechanism to implicate.

L255 and 265. I am still confused about the ice cliff melt rate distribution, for 2 reasons. First, it appears that at high elevations in the study area, it appears that your modelled ice cliff melt rates are lower than the modelled subdebris melt rates. This is not plausible (or there would not be cliffs!). Second, the ice cliff melt rates in this region are also lower than for clean ice, which should be an approximate lower bound for ice cliff melt at all elevations: are nearly bare ice, but with surface debris well below the critical thickness (thus enhancing melt relative to bare ice, if we can neglect increased shading).

L257. These rates correspond to a mean cliff enhancement factor of 1.72 relative to the mean subdebris melt rate, which I suppose is lower than anywhere else due to the thin debris.

L259. 'Dominates' is a strange term here. Certainly the reduced melt rate due to debris thickness is apparent, and debris thickness differences are more important than the difference in cliff density.

L265. This appears to be a typo – the cliff melt rates are an order of magnitude lower than under debris?

L271. This is again very disjointed to the rest of the analysis. Also, the low lake coverage in the upper ZMT makes a lot of sense as this area has steeper surface slopes in 2009 (Fig 2).

L282. I appreciate consideration of the applicability and extension of this method to other sites/scenes. I think the biggest challenge for application to other scenes is that the tested parameter sets produced extremely variable results, and would need optimisation for every new site and image.

There are also seasonality patterns to consider- cliffs often retain snow longer than the debris surface, for example.

L324. The variability in observed backwasting rate is considerably stronger than any bias due to the observation location – the question is really where the mean lies.

L340. The potential distal effect of these features is conceptually well understood to be via internal ablation along englacial conduits, but is not possible to validate at present. See Benn et al (2001, 2012, 2017), Sakai et al (2002).

L344. This section/paragraph feels orphaned. It is worthwhile to note that even accounting for the hypsometric distribution of cliffs, the spatial pattern still emulated Ostrem's curve (just with different effective thicknesses), suggesting that this concept might be useful as a proxy for the altitudinal SMB pattern even where cliffs account for 40% of melt – just not directly comparable to stake measurements.

L353. 'counter' - should be singular

L356. 'trend' should be pattern

Table 3. I don't think that Buri and Pellicciotti (2018) is the most appropriate study for Lirung Glacier for this purpose. Why the comment on EB below the table?

Figure 2. It is not clear what the bars are in the upper left – is this the domain with supraglacial debris?

Figure 3. This is a very nice conceptual summary! Can you include a pond or stream?

Figure 4. In the caption for 'c', there is a reference to a 'black line' which corresponds to the 'solid' line I think.

Figure 5. This is a very nice summary of the method. Can you reproduce the same for the Sobel method to be included in the Supplementary Information?

Figure 6. Panel (b) does not depict the bare ice area outlines as in (c).

Figure 7. Nice depiction of the optimization. No colorscale is shown, though, and due to the different axis ranges, it is difficult to visualize the lowest Euclidean distance.

Figure 9.  See my comments in the text on line 255. Surely the lowest ice cliff melt rate (here 2.9cm/d) should correspond in space to the highest sub-debris melt rate (5.8 cm/d) – at the highest elevations. But then, the cliff melt rate should not be lower that the subdebris melt rate – that makes little sense. This suggests to me that the linear parameterization of ice cliff melt with elevation may not be appropriate.

Figure 10. Nice summary. Can you include a depiction of the cliff-only melt rates vs elevation in panel (a)? It is interesting that the cliff portion of melt is highest high in the ZMT, but still makes little difference in the mean melt rate profile. Also, it would be very meaningful to complete a version of panel (a) for the min and max melt parameterizations.  Effectively these estimates are generous uncertainty bounds for your results.

Figure 11. Do you have a depiction of supraglacial streams (density or otherwise) to complete the picture?

Figure 12. This is orphaned from the discussion and seems like an odd figure to close on.

---

## Author Comment (AC1) · 15 Feb 2020

Review of Anderson et al., part B, The Cryosphere, October 2019

Thank you kindly for taking the time to review these manuscripts.

In this second opus of their trilogy, Anderson et al. deduced the spatial pattern of melt
due to ice cliff and under debris, and consider the distribution of supraglacial lakes to
conclude that melt hot spots (cliffs and lakes) are not sufficient to explain the pattern of
rapid thinning on Kennicott Glacier.

Overall this is a series of paper that bring a lot of new data and contribute to show
that melt hot spots (ice cliff and lakes) only modestly contribute to the overall mass
loss of a large debris covered tongue. A clear achievement has been to perform such
measurements on a very large glacier in Alaska and proposed methods to extrapolate the point wise
measurements to the overall debris-covered glacier tongue.

Thank you for the kind summary.

General comments for the three papers.

1/ I am not convinced by the need to split this paper into three parts. It implies lot of repetitions and
also mean that the reader as to refer to other parts of the article which is not convenient.

We appreciate this perspective on the body of work. A single manuscript would allow the
development of the ideas without needing introductions or conclusions to transition through to see
the full breadth of the work.

What would happen with one manuscript though is that many of the conclusions from each Part (in
situ, ice cliff delineation/distributed melt, and holistic view) would be lost in a single contribution.
It would simply be an overwhelming manuscript, with a diverse swath of conclusions. We tried this
in a previous submission and the paper was rejected for being too complex.

Some data are plot several times in the three articles (debris thickness, dh/dt for 1957-2009 etc. . .) I
think the authors missed here an opportunity to put everything together.

We understand and accept that some repetition can be annoying. Here we tried to walk the line
between too much repetition and too much turning back and forth between manuscripts. But we
would like the reviewers and editors to consider how overwhelming one manuscript with all of
methods and ideas we outline across the full breadth of the 3 parts. That manuscript would be a
monster and less intelligible than what is presented here.

Specifically in this part B, the discussion (section 4.2.1) whether ice cliff or debris can explain the zone of maximum thinning would be much more straightforward if Part B and C were merged. Right now this discussion is a lot of speculation to finally justify the need for a part C.

We can see how the body of work could be split in different ways and certain parts would be easier to digest.

We feel that Part B presents a different perspective on thinning than Part C. First we look solely at melt and then we add in dynamics for additional evidence.

We don't feel that section 4.2.1 is much about speculation but rather addressing a new perspective not often discussed in debris-covered glacier literature. We find the arguments we present there are valuable for considering other debris-covered glaciers. Taking an end member approach allows us to consider these processes from a broader perspective than they often are.

Part C presents, additional evidence about whether ice cliffs can explain the zone of maximum thinning but there are additional new items in Part C. We try to bring together all of the observations in a holistic framework in Part C. We feel that is an important contribution that justifies more space than what a single manuscript would allow.

2/ One strong limitation (that needs to be emphasized more) is that field measurements over a short period of time in July 2011 are used to interpret a map of elevation change measured over a multidecadal time period. Authors need to recall to their reader that their results apply to 2-month period in summer. The whole discussion would have been much more meaningful if the elevation changes were also measured for the same time period where surface melt features are studied (but the DEM data are probably not available. . .).

We actually have dh/dt data that spans 2011 and will include it during revisions. The zone of maximum thinning is in the same location as the dh/dt maps from 1957 to 2009.

General comments for part B.

3/ I miss a more thorough description of and comparison to earlier studies mapping ice cliff automatically. In particular Kraaijenbrink et al., RSE, 2016.

We agree that a further analysis of this nature would be fruitful and interesting. But it seems that this comment contradicts this reviewer's idea that all 3 parts be combined into one.  We cannot add more content and have these 3 parts combined into one. Content would need to be trimmed down substantially to fit all three parts into one.

4/ I feel it would have been very interesting to see an evaluation of the ice cliff mapping algorithms using independent dataset, for example the Ragletti/Steiner cliff dataset on Lantang Glacier. Maybe this ice cliff automatic mapping part would have deserved a dedicated article, and all the rest of the results would then fit a single contribution?

We agree that the ice cliff mapping algorithm could be a separate paper in and of itself. This comment directly contradict's Evan's comment though that this ice cliff mapping approach is a minor part of this manuscript. We feel that this new approach does deserve more attention in this Part B and we will bring a bit more discussion of this method compared to others during the revisions.

5/ Uncertainties could be treated in a more systematic way so that results in the end should all be quoted together with their range of uncertainties. This applies to all three parts.

Please note that Figure 10 already includes a generous, extreme range of uncertainties.

6/ When authors provide % of melt, they should always make it clear that this is a percentage of the debris-covered tongue (and not the whole glacier!)

We will clarify this during revisions.

Specific comments.

L16 What does "enhancing" the mass balance mean. A mass balance can be increased or reduced. Is this formally demonstrated? I thought it was debated.

We will change 'enhancing' as suggested. We need to be more clear about what we mean. Because melt rates of ice cliffs are most certainly higher than sub-debris melt rates (where h_debris > 5 cm), the ice cliffs tend to move the surface mass balance more towards what melt rates would be if there was no debris at all. This is the 'enhancing' effect we refer too. But we will clarify this in revisions.

L21 "Total" is ambiguous. Tongue-wide or glacier-wide?

We will clarify this.

L41 One does not expect results in the introduction.

We can rephrase this sentence so it does not read like results. But the statement we make is based off of previous work from the area so it is more an observation to set the stage for the rest of the article.

L51 "surface mass balance" would be a more appropriate way to refer to it

Eq 1. x,y are not defined.

L58. So do the authors neglect them? It should be stated unambiguously.

We will state that we neglect them.

L88ff. Splitting the article into three parts leads to many repetitions such as this section. Problematic in my view.

It is true that some repeating is necessary, but at the same time combining all three parts would make a manuscript with 27 unique figures that combines, diverse in situ field measurements, MF correction, a new ice cliff delineation method, extrapolation of melt across a large glacier, discussion of lakes, streams, surface velocities, ice emergence, discussion of the uncertainty of each approach, a plethora of field measurements, and the presentation of what we feel is important and new theory in Part C.

We feel that a little repetition in each part is fine considering that a combined manuscript would require that we fundamentally change in the conclusions we draw. We would have to drop the discussion of surface processes presented in Part C as well as much of the very important field data from Part A.

It is important to also consider the drawbacks of combining the parts into one as well.

L160. Unclear (understated) what meteorological data would have brought, if they had been available.

This is more pointing to differences in our approaches from many debris-cover studies today. We can clarify this.

L169-170. This statement that 20% of ice cliff area need to be added is enigmatic at this stage in the paper.

We will clarify this.

Eq (3). How the type of fitting curve was chosen? It seems to come from nowhere. Can it be justified?

We will provide a proper method of justification for the curve fitting based on error metrics. Ultimately, the details of the shape of the curve are secondary (linear or non-linear) to the melt suppression effects of debris.

L183. I see in part A that your cliff backwasting rate neglect emergence velocity. This needs to be justified.

It is not clear what the reviewer is referring to here. The emergence velocity is not relevant to our backwasting rage here because we measured the rate in situ, on the glacier.

L191. A statement such as (here) "based on an analysis of 2-m ArcticDEMs" is too vague.eq (7). Ice ciff area. Is it planar or real area? I think "i" must be added as superscript with $\hat{b}$dot debris and $\hat{b}$dot icecliff

We will clarify and expand on this.

L204. I do not understand why fitting a curve through 25% or 75% of the data points leads to "extreme" cases. Not clear. Why not a curve containing 67% of the data (to have 1-sigma uncertainties). See my general comment about treatment of error bars.

We just need to add to line 204: 'for each dataset.'When every dataset is given 25 or 75 % uncertainties and then we use the relevant extreme curve fit for our estimation of melt across the study area an 'extreme' case results. This needs to be more clearly stated and emphasized in the manuscript. There already is a good uncertainty analysis here, but for some reason that was missed. For example these 'extreme' cases where something like 5% of possible cases are outside of the red band in Figure 10a and 10b.

We could use 1-sigma bounds but the ultimate results will not change.

Here, while we aren't using 1-sigma uncertainties we do provide a justifiable error analysis it just needs to be emphasized more.

L226. "error checks" is a strange terminology. Why not "validation dataset"

We are happy to change the terminology.

L240. Percentage should be 21% and 31%, right?

Yes we can fix this.

L244. Where does "11.6%" come from? I read 11.4% and 11.7% above.

L245. This raise the question of whether all studies defined the "debris-covered tongue" the same way. Did the authors check carefully previous studies for this aspect?

We used the same study area, but will check the numbers in codes again.

L247. "This implies that ice cliff coverage varies with debris thickness". This seems like a hasty conclusion. . . other example from the literature to support the statement?

There are not many studies that quantify ice cliff distribution and we don't see anything wrong with highlighting what 'could' be an interesting trend. Perhaps we could change the sentence to:

"This implies that ice cliff coverage could vary with mean debris thickness".

Since this is the first study

L257. One expect an error quantification for each term (81% and 19%).

The next sentence does just this for the 19% number, which also does the same for the 81% number so the error quantification is already present though it might not be in the expected format.

L268. "Across all of the elevation bands, the ice cliffs between 500 and 520 m generate a maximum of 40% of the total mass loss due to ice cliffs and sub-debris melt." I am not sure I got the meaning here. Maybe reformulate for clarification. (it is clear from the figure, just a text improvement)

We will fix this.

L273 "within" rather than "with" (I think)

L289. 19%. Lack error bars and also authors need to remind that this applies to a short period of time during summer 2011. So they cannot draw such broad conclusion.

We do not want to overwhelm the reader with uncertainties in every sentence. But the uncertainties are presented in the manuscript. We will highlight them more in revisions.

We will add in additional dh/dt data that spans the 2011 summer. We also make, what we think is a compelling argument in section 4.2.1 for why we do not expect the mass balance pattern to not primarily follow Ostrem's curve.

I would be curious to see a comparison of this number to the total glacier-wide ablation during this period if available. Is not it just a few percents? Do we need to really worry so much about ice cliffs for glacier-wide or region-wide application (and future projections)?

We could add in this analysis but we aren't sure how it would improve this study and the aims we outlined in the introduction. As we framed the study, we are interested in explaining the thinning patterns of debris-covered glaciers, and Kennicott Glacier specifically. Ice cliffs are an important, proposed contributor to this. Thinning patterns hold implications for longer term thinning patterns and hazards.

L307. the SMB cannot be "suppressed". It can be increase or decrease. (SMB is increased here, or less negative)

We will fix these terms.

L321. "This required backwasting rate is well beyond potential biases introduced due to the summer of 2011 having anomalously low air temperatures". Statement not really explained and justified.

What we mean is that the weather during the summer of 2011 was not anomalous in a way that would change the melt rate pattern we present in this study. We will make it more clear with further analyses.

L324. Is this potential overestimation from the sampling strategy (at top of cliffs) included in the error bars, as it should?

The assertion in the text as it stands is logically correct and we feel is a better way of arguing than adding error bars these error bars. We aren't sure how following this suggestion will actually improve the error estimation beyond what we already have presented. We could include error bars for each backwasting but how would that improve the legibility of our figures? How much of an over estimate is it? We do not know, so where do we end the error bar?

Rather if we know from other approaches that these are maximum estimates we can use that fact, as we do already in the discussion to present these ice cliff backwasting rates as generously high.

L334. "mass loss" should be replaced by "melt rate" here.

L344. I did not get the point here.

This lays the foundation for Part C. We will emphasize that here. In Part C we use the melt rate pattern developed in Part B to make further theoretical arguments about the interaction of surface mass balance, ice dynamics, and surface processes.

L349. The wording suggests that 11.7 % of the glacier is covered with cliffs. No. This is the % of the debris-covered tongue.

Table2. For the ice cliff backwasting parameter f, the most likely value is not contained by the min/max interval. A typo? Or a real error? For the ice cliff area the most likely and max values are equal. This is also not really expected neither.

Thank you for the comment but this is not an error. We point the reviewer to parameter 'g' immediately below parameter 'f' which is the y-intercept for the curve-fit. If you also look at Figure 4 you will see that these parameters are correct.

Table 3. Ice cliff fractional area, a percentage of what total area?

% of debris-covered glacier area. We will clarify this.

L415. I do not understand this note.

This note should be removed and will be in revisions.

Figure 2. Are these data from Das et al? Did they use GDEM V2? This would be problematic because it has no defined time stamp. Explain the 1957 and 2015 grey boxes also.

We will look into the time stamp issue and make sure that the profile we show has a time stamp. Ah the boxes are the extent of 'continuous' debris cover upglacier from the terminus. We will add this note.

Figure 4. Multiple reference to part A complicate the reading.

There is only one reference to Part A. I am not sure where the reviewer got this from. I am not convinced that referring to another paper to read about the in situ collection methods is really a big issue. It actually makes Part B more clear and less muddled. We find that when too many methods are presented in a single paper it becomes exceptionally hard to follow the main thread of the manuscripts. This is a major negative for combining these manuscripts.

But we will clarify what the reader can look for in Part A so it is more clear that it is the in situ data collection methods. Emphasizing again that all the reader needs to do is look in another manuscript that goes into detail about the in situ methods.

25 and 50% or 25 and 75?

This is a typo, sorry.

Is it "elevation bins"?

We will clarify this.

Panel B. Why the order of values in both axis are reversed. Why not showing the Ostrem way?

As we state below in the caption: 'The axes are flipped to be consistent with other figures.'

Authors could refer (in the article, not here) to a compilation by Kraaijenbrink et al., 2017 in their Nature study.

Figure 5. Can the authors show the location of this small area of the glacier?

Yes, we will add this.

Figure 6. Impressive maps.

Thank you, we feel that this figure shows the viability of our ice cliff detection method on a very complex example case.

Figure 8. Showing percentage for panel c (instead of fractional area) would facilitate correspondence with the text.

We agree and will change this.

Figure 9 The sign only make sense if this is referred to as "surface mass balance rate".
If the word "melt" is preferred then positive values should be shown.

We will fix this.

Figure 10. See comment on Figure 9 for "melt"

We will fix this.

Figure 11. Recall the period of dh/dt. In panel b rather than repeating dh/dt authors
could show a map with the density / m 2 of ice cliff.

We will show a shorter dh/dt period that covers the 2011 span. The pattern is similar. The cliff
density suggestion is helpful, thank you.

Thank you again for your efforts reviewing this manuscript.

In addition to the changes above we suggest that these changes are enacted:

**Part B: proposed changes**

We feel that there is more than enough new material here for a stand alone paper, but in order to
improve the manuscript we propose that we add these additional datasets/ideas to Part B:

- We will add additional text supporting the usefulness of our new ice cliff detection method.
  In the supplemental we will include additional satellite photos showing how ice cliffs tend to
  be darker than the surrounding debris so this method can therefore be applied on other
  glaciers. We will also compare our method with other approaches from other glaciers.

- We will present new DEM differences from 2007 to 2013. These dh/dt data show that the
  zone of maximum thinning remains in the same spot as for the period from 1957 to 2007.
  We will also include additional laser altymetry data from 2007 that shows a similar thinning
  pattern.
     -This will address one of the main criticisms from multiple reviewers.

- We will introduce back-of-the-envelope calculations of the possible effect of englacial melt,
  sub-glacial melt, melt under pond surfaces, and melt by streams. This will clear up any
  issues related to this manuscript not being comprehensive with regards to melt hotspots.
  ◦ We will not include stream digitizations in this manuscript because we cannot possibly
    digitize all streams on the glacier surface (imagery is too coarse). The streams play more
    into the feedbacks in Part C. We will instead make arguments about the surface area
    coverage of streams and their plausible effect on surface melt.

- We will use a uniform curve fit through the ice cliff backwasting data. And also explore the
  effect of other curve fits, producing different figure 10a and 10bs which we will put in the
  supplemental and discuss in the main text.

- Add in the paragraph description that links each of the three papers and helps guide the reader through each manuscript.
- Make sure it is clear how generous the uncertainty estimates already are in this paper. One of the reviewers missed these error estimates completely.
- Emphasize the increasing importance of ice cliffs under thicker and thicker debris.

---

## Author Comment (AC2) · 15 Feb 2020

Review of 'Debris cover and the thinning of Kennicott Glacier, Alaska, Part B' by Leif Anderson et al., under consideration for The Cryosphere

Thank you kindly for taking the time to review our manuscripts!

Part B of the Anderson et al trilogy aims to combine empirical relationships of surface properties and melt rates, based on the field measurements presented in Part A, with remote sensing observations of different surface features (particularly ice cliffs) in order to arrive at distributed estimates of melt rates for the period of observations. The analysis then uses these distributed melt values to address whether or not the identified melt hotspots can explain the thinning patterns evident at Kennicott. This is an important question as the glaciological community is trying to disentangle the influence of surface mass balance and ice dynamics for debris covered glacier evolution, and this is the first time the question has been addressed in Alaska, where surface debris is prevalent.

As such, this represents an important contribution to a current topic of research, and provides an answer to that question for Kennicott Glacier – these hot spots do have an important effect on the surface mass balance, but it is not plausible that they compensate for the overall melt reduction due to surface debris. The study could be more systematic to provide a definitive answer, and I have some criticisms regarding the empirical relationships presented in determining the distributed estimates of melt, as well as the difference in temporal scales between long-term elevation change (52 years) relative to single-year field measurements. However, although I have quite a few comments, no changes along these lines are likely to change the conclusions of the study. Rather, my principal concern is that the separation of this analysis from both Parts A and C reduces the strength and presentation of the entire analysis, while also leading to repetition of text and figures, as well as cross references. There are certainly gaps in the analysis because aspects have been included in Parts A or C rather than here, and some restructuring across the three manuscripts might improve the readability of all three. This is a choice for the authors and Editor to contemplate, but my opinion is that some consolidation would be beneficial.

Thank you kindly for taking the time to review these manuscripts!

Main comments:

My principal question in reading this manuscript was whether it should be standalone or integrated with Part A (and possibly C, which I have not reviewed). I appreciate that this is a difficult decision, and that Parts A, B, and C combined represent a substantial body of work.

Thank you for your thoughts on this we reply to all comments regarding structure in the main response.

There are certainly some advantages to be considered for maintaining separation between the manuscripts, but my impression is that A and B are both weaker manuscripts separated from one another. The field measurements were clearly collected for the purpose of deriving distributed melt rate estimates, which leaves Part A without a compelling conclusion or discussion.

It is worth noting that while it may seem that these are weaker manuscripts apart the question in our mind is: Do all three work together and complement one another.

We have version of the manuscript where we did combine Parts A and B and we simply found that there was too much in the manuscript, too many directions, too many diverse methods discussed.

A and B were presented in 2 parts so we could explore the in situ data alone. If we combine the manuscripts we will lose the analysis of the in situ data as well as the presentation of many measurements relevant for melt modelers. Additionally combing A and B means that the ice cliff detection method becomes a footnote instead of a primary contribution. There are other manuscripts that just present an ice cliff detection method.

We agree that Part A can be improved and we outline how that can happen in the proposed changes. We have more data we can add to Part A such that is takes more of a unique perspective but we excluded that data in favor of streamlining Parts A, B, and C. Please see our primary response about the structure of the manuscripts.

At the same time, Part B requires frequent reference to Part A, or blind faith of the part of the reader with regards to the methods and results of the field data. As a consequence, there is also quite a bit of repeated material to cover (for example, content from 4 figures in Part A is also displayed in Part B).

We appreciate this comment but we feel this issue is overemphasized. The reviewer finds that 1) you need to read Part A before understanding Part B and 2) that information is repeated between the two manuscripts.

Both of these issues would occur if Part A was accepted first and then Part B was submitted independently later. From Part A we pass on three datasets, and numerous observations to support the inferences made in Part B.

(for example, content from 4 figures in Part A is also displayed in Part B).

Content from A is in only in 4 panels in 2 figures (alaska context figure and curve fits to debris, sub-debris melt rate, and ice cliff backwasting). Which is perfectly acceptable going from one paper to the next. We can also leave out the data figure that is repeated but we feel that would require the reader to look back so we include what is really needed to understand Part B.

My instinct when reviewing Part A was that much of that material could be more meaningful if integrated with Part B, in supplementary information if not in the main text, and I now think that would greatly improve the readability of this manuscript.

We appreciate this view but when the manuscript was previously combined it was bloated and difficult to outline all the pieces needed for Part B with the detail needed.

I like the ice cliff delineation method in its simplicity (although some details need to be clarified, below), but its transferability is not very clear. Often when developing/proposing a new method it is necessary to see how robust the method is, but in this case the method has clearly been developed specifically to map cliffs in this particular scene, in order to apply the empirical melt relationship. As such, it is a relatively small part of the story in deriving the distributed melt estimates.

Thank you for highlighting this. We present the new ice cliff detection method as a proof of concept for a rather tough case. Kennicott Glacier has more dense ice cliffs than any other debris-covered glacier that we know of. Slope threshold approaches do not appear to be as effective on this glacier.

We could include a more thorough comparison with other methods we feel that should be the focus of another manuscript.

At the same time, the maps of supraglacial ponds are not integrated very well into the story, while the exclusion of supraglacial streams (also mapped from satellite imagery) is a bit

Thank you for pointing this out. We actually feel that simple fact that ponds do not align with the zone of maximum thinning shows that they cannot explain the pattern. Please see Figure 10c.

[Figure]

It is also important to note that the red range in 10a and 10b are the extreme melt estimates.

We state this in the manuscript. It is not clear how lakes can control the zone of maximum thinning if they are not present there? We show that lakes are not present across this zone of maximum thinning from 1957 and 2009.

In Part A we also highlight that ice cliffs with ponds at their base do not backwaste at rates higher than ice cliffs with streams or ponds. We will emphasize that more here.

We can include a discussion of surface streams and do a back-of-the-envelope calculation for the amount of strain heat produced by the streams. We would be very surprised if streams would contribute enough melt to compensate for the melt reducing effect of debris, especially considering that ice cliffs would need to cover 90% of the debris-covered tongue to compensate for the effects of surface debris.

The study aims to address the role of melt hotspots in explaining thinning rates. For this to be a definitive analysis in this regard, I feel like this needs to be done in a more systematic manner, whereas the present analysis seems to focus exclusively on ice cliffs.

Thank you for raising this point. The important point that is emphasized in the manuscript is that this glacier has the highest concentration of ice cliffs of any that we know of, and is thus a good place to test if ice cliffs, especially can compensate for the melt reducing effects of debris.

The distribution of supraglacial streams seems to be a major gap in the analysis of this part of the study, especially as the properties of supraglacial streams are assessed in Part C. Streams are mentioned throughout the background as hot spots and possible factors contributing to melt, but are then completely neglected in the methods and discussion.

We think streams do have an important effect (and we discuss them at length in Part C). We are happy to discuss streams here in Part B. But we do address them in Part C where we do digitize many streams.

It is important to note that there are very few studies that actually address the effect of streams on debris-cover glacier mass balance and to date none have really quantified an effect. Here is where having space to describe our methods in Parts A and C is an advantage. We mention that cliffs with streams do not backwaste faster than cliffs without streams based on our measured 60 ice cliffs. This implies that they have a secondary effect on mass balance. We aren't aware that this has been noted before and we feel this is support for keeping Part A in its current form so we have space to point out these important observations based on the in situ data.

We can do the calculation here in this manuscript for the amount of strain heat produced by an extreme concentration of streams on Kennicott Glacier. We hypothesize now that while streams locally enhance melt they are largely inconsequential melt contributors when averaged across the glacier. See Section 4.2.1 for support of this argument here.

While streams appear to help maintain ice cliffs (see Part C) we are skeptical that they will be a major contributor to debris-covered tongue wide melt rates. We will address this in the revisions. If adding them to Part B helps in the continuity of the 3 manuscripts we are happy to add them.

I understand that the role of surface streams cannot be assessed quantitiavely in this manuscript, but neither can the role of ponds. Similarly, although internal ablation is usually regarded as negligible in this type of specific mass balance assessment, there have been suggestions that this is a non-negligible term for extensively debris-covered glaciers. It is exceedingly unlikely that this mechanism could lead to the debris-cover anomaly, but for completeness I think it should be considered numerically along with the cliffs and ponds.

We will explore back of the envelope estimates for englacial mass loss due to strain heating of water along with the surface streams. They could be interesting additions to the analysis.

The relationships between elevation and debris thickness, and between elevation and ice cliff backwasting, are very weak. In neither case does elevation appear to be a primary control of the property.

We are a bit confused about the elevation versus debris thickness statement because lower in this review you state: "In Part A it is clear that while elevation is a principal driver of debris thickness variability, there is considerable heterogeneity within any elevation bin." As we suggested in Part A it would be a nice idea to extrapolate down flowlines, but we felt like this was a methological development to focus on in later studies.

Despite the scatter we agree on, Figure 5 Part A (see below) shows that debris thickness does increase down glacier. We can argue about whether or not that is linear or not but the box-plots show the increase. Here is a reason why Part A standing alone is good for presenting the body of work. We are able to show the box plots that really reveal the thickening of debris down glacier.

[Figure]

**Figure 5.** Pattern of debris thickness with elevation. a) In situ debris thickness measurements. b) Debris thickness boxplots in 50 meter elevation bins. Outliers are represented as +'s.

How different would your results be if you interpolated spatially (within flowlines or lithological units for example)? Did you consider alternative methods to provide a distributed debris thickness estimate?

We did consider different approaches but we felt that the number of ice cliffs and variable surface topography was a strong hinderace to using thermal infrared imagery. A full consideration of this would require an additional study.

For example, one could consider the radiance measured by satellite thermal infrared imagery as related to debris thickness, and use your measurements to constrain this relationship locally to upscale in space much more meaningfully.

We did consider this approach but because of the number of ice cliffs in the study area we felt that this could be another manuscript in of itself, requiring significant justification and method development. We would need to remove the bare ice effects from each pixel before estimating debris thickness.

For ice cliffs, justification of a fit to elevation needs to be more explicit in this Part of the study.

We agree with this elevation does not need to be the main control here. We will explore other curve fits but a uniform distribution with elevation seems most defensible.

The conversion of backwasting rates into ice cliff melt misses a slope correction for the cliff area, which results in an underestimation of melt (see comment on L190).

The reviewer is correct and we have included an updated Figure 10 below. The change in the distributed melt estimates was minor but not negligible though.

[Figure]

This raises a difficult question that has not been carefully considered yet for debris-covered glaciers, which is that the real surface area can be 10-20% higher than the planimetric area. For this study comparing geodetic thinning observations with estimated melt, that is an important aspect to consider, as melt occurs relative to the real surface area. This effect is especially pronounced for ice cliffs, but is also crucial for the 'background' melt rate if the glacier surface is highly variable.

Having access to a 5 m DEM of the glacier we noticed this same effect when including the local surface for sub-debris melt rate. We agree and are happy to show this effect for sub-debris melt in a figure or two in the supplemental.

It is too bad that more recent geodetic difference data were not included in the analysis. At present it is not clear how the long-term thinning rate relates directly to the 2011 observations. Would it be possible to use the ArcticDEM datasets to derive a recent-period thinning pattern? This would be more meaningful for a comparison to the contemporary distribution of surface features. The long-term perspective is still useful for understanding dynamic changes, but comparing 50-year lowering rates to one-year melt patterns does not provide a definitive answer, especially for a clearly changing system.

We have these more recent geodetic estimates and we will include them in revisions. There is also laser altymetry data that supports the stable location of the zone of maximum thinning in the long run as well as the more recent decades.

Some rewriting is needed for readability and presentation standards in The Cryosphere. Although the ideas are well developed, some sections of the paper read as bullet points and/or the word choice has not been considered carefully, leading to some of my comments below.

We agree that there is some ambiguity in the writing that needs to be corrected. This review helped us understand where we can tighten the wording up considerably. Thank you kindly!

Detailed comments:

L23. Presumably this 19% of melt is for the debris-covered area?

Yes we will make sure the study area is defined correctly.

L25. Just a comment, for you to adopt or disregard: The literature has tended to use 'ponds' for these features as they are much smaller than supraglacial lakes on, e.g., the Greenland or Antarctic Ice Sheets.

We will switch to ponds. Helpful suggestion and we should be consistent with the literature.

L25. It would be nice to have the %areal coverage numbers here in the abstract, not just 'doubled'.

L27-27. This wording isn't very clear. By average melt rates you seem to mean the average for all surfaces, but as worded this seems to refer only to sub-debris melt. In the latter half of the sentence, do you mean that the overall melt relationship still follows an Ostrem-type relationship, even after accounting for cliffs and ponds?

We will clarify this as it is really important, making sure we emphasize which components we are including in the distributed melt.

Thank you for pointing this out. We will clarify this

Suggested change: Despite abundant ice cliffs and expanding surface lakes, average melt rates are suppressed by debris, with the primary control on elevation-band-average melt rates appearing to be debris thickness.

L34. These are broken sentences. The first half needs a reference even if it is now well understood.

We will join with a comma

L95. Has this happened progressively in the past 82 years, or primarily over some later period?

This is outlined in the citation, Rickman and Rosenkrans (1997). We will add a sentence describing the timing of expansion of lakes.

L98. Why is 'partially' here?

Good catch.

We will change this to: The presence of ice cliffs, surface lakes, and variations in debris thickness on debris-covered glaciers makes distributed estimates of mass balance difficult.

L108. Which spectral bands of the image do you use?

We will add:

On 109, before "The 2009 WV image…" add "We use the panchromatic band, which integrates radiance across the visible spectrum and provides the highest spatial resolution".

L113. This is true in certain conditions, depending both on debris lithology and meteorology. Such conditions may be prevalent for Alaskan supraglacial debris and melt seasons, but it is important to think whether such a method is transferable.

This is a good point. We looked into the universality of this observation that ice cliffs are generally darker than the surrounding debris cover using WorldView imagery in the Digital Globe's EVWHS viewer. We attach screenshots of Miage, Koxhar, and Lirung glaciers below. These are debris covered glaciers from the Alps and Tibet, which are found at different latitudes and lithologies than Kennicott Glacier. We find that, consistent with our Kennicott Glacier observations, ice cliffs on these other glaciers are generally darker than the surrounding debris cover. We can find examples where this is not true, for example along parts of the southwestern margin of Miage Glacier in the snapshot below. These regions may go undetected in the method we present here. We address this limitation in the manuscript (L281 in the original submission). To address this comment, we add at the end of this paragraph "This observation of darker ice cliffs is generally true for Kennicott and several other debris covered glaciers we examined, but this relationship should be verified before application to a different glacier of interest. There are situations (e.g., variable debris-covered and debris-free ice) where this method could detect darker regions that are not related to ice cliffs. Output should be examined to ensure that such conditions do not contaminate results". This language also partly addresses your L126-127 comment as well.

Miage Glacier (45.80 N, 6.85 E)

[Figure]

Koxkar (41.78 N, 80.1 E)

[Figure]

Lirung Glacier (28.23 N, 85.56 E)

[Figure]

L118. This is semantics perhaps, but I would argue that this whole workflow is your 'cliff detection method' (the name of step 2). It is a bit strange to have a 'cliff detection method' as the main step of a 'cliff detection workflow'.

Yes we need to use consistent terminology here and we will address it in revisions.

We have changed "detection" to "delineation" throughout.

L118. Is the histogram stretch just a linear min/max stretch? Is this histogram stretch applied to the image globally or locally within a patch? What spectral band(s) are used in either approach? I suppose that you also start with a debris outline (and glacier outline), which is important to acknowledge for a remote sensing approach.

We use a linear histogram stretch uniformly across the entire image. We do not clip the raster to the glacier, and include both on-ice and off-ice areas. The method ends up not being sensitive to this because we are tuning histogram stretch parameters to optimize performance on debris-covered ice, so it doesn't affect the stretch. We have added language at the end of the paragraph to clarify these points.

The step we call "detection" is the 'heart' of the ice cliff delineation method. The other steps are essentially pre- and post-processing. We have added these terms to the numbered steps to make this clearer.

Change to "1) processing: stretching the image brightness histogram to a suitable range for our ice cliff detection methods; 2) detection: applying an ice cliff detection method; and 3) post-processing: morphologically filtering of the detected ice cliffs (Fig. 5). We apply a linear histogram stretch uniformly across the image, including both the glacier and surrounding off-ice areas."

L123. Is the saturation stretch part of your step 1, or a separate part of step 2?

The saturation stretch is a pre-processing step (Step 1). However, we use different stretch values depending on which method we use (edge detection or adaptive binary), because the methods perform best with different exposure levels. We have added language to clarify this.

Change to "In pre-processing, we use separate saturation stretches (Fig. 5) for each method by applying the exposure function in the scikit-image package (skimage). The different methods perform best with different exposure levels, so we create two separate stretched orthoimages in pre-processing

L125. I see that the size of the moving window is included later as one of your parameters in the MC optimization. It might be good to give the reader a warning that that is the direction the methods are going, and that you will start with a description of the implementation of each approach first.

Thanks for this suggestion, we added language to foreshadow this progression.

Add at L120 after (Fig. 5): These steps introduce several processing parameters, which we select using a Monte Carlo optimization method. Below, we first present the processing steps, followed by our parameter optimization procedure.

L126-7. I would certainly not say that adaptive thresholding is insensitive to changes in debris cover and illumination, but it may be less sensitive. Ice cliffs are not uniform in surface character, and can appear both brighter and darker than the surrounding topography in different circumstances. It may be that these nuances are not so evident in the lower portion of Kennicott Glacier, but two particular cases would pose a major challenge for the ABT approach: 1) a population of ice cliffs with variable surface character (debris-free vs covered with fines) which will increase the spectral variance of the cliff population; 2) otherwise dark (potentially wet) debris. This is discussed later in the manuscript, but for a presentation of a new method, I think the accuracy and appropriate application needs some further advertisement/warnings.

We changed this to state that the approach is less sensitive to these variations than a global threshold. We added specific reference to this potentially problematic situation of alternating debris-free vs. debris-covered in the text we describe in response to your comment on L113. In response to that comment, we also added "warnings" that the output should be examined to look for spurious results, and that a user should verify that ice cliffs are in fact darker for their glacier of interest.

Change L126-127 to : "Because the brightness threshold varies across the image, the *ABT* approach is less sensitive to changes in illumination and debris color than a global threshold."

L133. Some of this content is Results

Yes, these can be considered results, but our purpose for including this language here is to motivate our last step in the processing procedure (morphological filtering). We have added a sentence to the start of this paragraph to make this intent clear.

Add to L132 start of paragraph: "The last step in our processing process is morphological filtering to remove spurious data".

L142. Was this 3% disagreement in total area? Can you provide a dice score for the two independent outlines? It is very possible to derive largely different cliff distributions but arrive with the same area.

Yes, the 3% was total area. WHA to look into Dice score.

L144-149. If I understand correctly, there are thus 6 parameters for the ABT and 5 for the SED implementations, with 4 shared between the two. How did this occur in practical terms? 2500 runs for each implementation, or 2500 runs used the same values for the shared parameters? More importantly, it is worth noting that with 5 parameters, 2500 runs results in an effective

sampling of ~4.8x in each parameter (4.8^5 ~ 2500).

We added text to clarify this procedure. In each of the 2500 iterations, we select the value of every parameter at random using a uniform probability distribution across a set range of possible values for that parameter. The ranges were determined using operator judgement to cover all physically-meaningful values. This method allows searching a wider parameter space with fewer iterations than the approach you describe. Interactions between processing parameters across their entire possible ranges are captured. The parameter space is not sampled as systematically as you describe, but it is covered more broadly.

Change L148 sentence starting with "We ran…" to "We ran the ice cliff detection algorithm 2500 times with differing parameter choice. In each iteration, every parameter is randomly selected using a uniform probability distributions over that respective parameters range of possible values (Duan et al., 1992). This method allows us to efficiently test performance across a wide range of parameter values and is sensitive to interaction between selected parameters across their ranges."

L152. 'The origin' is a bit ambiguous here, as it is true for figure 5, with the x-axis as the negation of the true positive rate. So really this is ranked by distance from (1,0) in your optimization space, correct?

Thanks for pointing out this potential confusion. We added that we mean the origin on Figure 7, which you are right is complicated by the "1 – true positive" term. Perfect model performance is TP,FP=(1,0) like you say, this just becomes (0,0) on that plot due the "1-TP" term.

Change to "Euclidean distance from the origin on Figure 7, which defines…"

L153. Why did you choose to reduce the FP rate (at the expense of TP) from the optimal parameter set? Can you please provide a dice coefficient for this parameter set for each approach?

We will look into the dice coefficient approach. Thank you for pointing this out.

L157. Process observation (2) actually refers to melt rates, rather than backwasting rates.

Good catch we will correct this.

L158. The influence of lakes was noted earlier by Brun et al (2016) and Miles et al (2016), among others.

We will cite these works.

L163. In Part A it is clear that while elevation is a principal driver of debris thickness variability, there is considerable heterogeneity within any elevation bin. As your field measurements of debris thickness could not encompass the entire study area (that would not have been feasible), do you think they sufficiently characterise the unmeasured area (particularly the NW of the domain)? Have you tested the importance of debris thickness heterogeneity in your

overall melt estimates? The subdebris melt relationship is not linear, so melt calculated with a mean thickness may not accurately approximate the mean melt rate.

This is a good point that is difficult to address for all folks working on debris-covered glaciers. Because we are trying to find out if ice cliffs can compensate for sub-debris melt this bias actually makes our estimate more generous.

Because many of our debris thickness measurements were taken at the top of ice cliffs though we suspect that our estimates underestimate debris thickness. We take our debris thickness measurements to be minimum estimates which means that sub-debris melt rate are likely to be even lower than what we estimate throughout this study, and through the study area.

L167. It is worth noting that you neglect internal ablation as well as other thermokarst processes (ponds, streams) in this computation for practical reasons.

Yes, we will note this.

L176. Please provide a goodness-of-fit for this empirical equation.

We will add this.

L180. It is interesting that as formulated, b_ice is the measured clean ice melt rate near the top of the study area, rather than the lapsed melt rate for each debris point. This is much more practical, but ignores the real melt suppression by the debris as a shortcut to a rate.

This is a good point and you laid out the reasoning, it is simply practical. At one point does emphasizing the details of physical processes (i.e. that bare ice melt rate increases a bit downglacier) get in the way of simple representation of the essence of sub-debris melt? The question really comes down to how important is it that bare-ice lapsed melt rates increase a little downglacier compared to the effects of debris thickening debris. On Kennicott Glacier it is clear to us that thickening debris is much more important than increasing energy for melt at lower elevations. The equation from Anderson and Anderson (2016) can easily be used with increasing bare ice melt rates if the user desires it.

In any case, I presume that the equation (as in Anderson and Anderson 2016) is based on the measurements presented in Part A? Please provide a goodness-of-fit measure.

Yes equation in Anderson and Anderson, 2016 is based on the data from part A. But note that the other fits from other glaciers in Anderson and Anderson (2016) are not necessarily taken across an elevation range. We will report an RMSE for the curve fits, optimizing for h_star.

L185. I am sceptical of this linear fit given the spread of observations in Part A – a goodness of fit would be expected to be very low.

We will provide additional curve fits for ice cliff backwasting and additional versions of Figure 10 to show the effect of this curve fit on our broader results. These figures can be placed in the supplemental.

If elevation is a secondary control, what might you presume is a primary control for the difference in backwasting rates?

We could address the scatter in backwasting rates more formally in Part A. We suspect that it is local debris thickness but a more formal analysis is needed. This could be a nice contribution to Part A. Because of the number of angles to look at the data and analyses in the full body of work we feel that multiple parts are justified.

L187. The similarity of backwasting rates for cliffs with/without lakes may be due to the observation type and period. Ponds and streams tend to incise thermoerosional notches, which can later collapse, thus enhancing the seasonal mass losses but not affecting what one would observe from the top of the cliff over a month or two. This is not a criticism of your work, it is just worth noting that this nonobservation doesn't mean a melt enhancement is not occurring.

This is a very thoughtful comment. We agree. It is a really neat next line of research! We have in fact observed this very collapse phenomenon on a glacier in Switzerland!

L190. The correction in backwasting rates for cliff slope is correct, but it is also necessary to correct the cliff area from planimetric to surface area in order to correctly estimate melt from these inclined features, as melt occurs perpendicular to the surface. Thus

$A\_i = A/\cos(\theta)$

With theta=40 degrees, this is a factor of 1.3 to all of the cliff-related melt calculations.

We will look into one more code but we think the reviewer is correct here. We have updated figure 10 with the correction (noting that the red bands show the extreme range discussed in the text):

[Figure]

L192. I think it's reasonable to apply a constant slope for the melt calculation, but why is this the 'most likely case'? Can you please provide some supporting information as supplemental figures, etc?

This a good point. The 'most likely estimate' based on our data collection and analysis.

L201. 'Most likely' is superfluous here; it is an estimate. It's nice that you provide bounds!

We will change to 'most likely estimate.'

L202. This 'best estimate' is using the parameter values already given in the text, correct?

Yes we will clarify this.

L213. I suppose you use the lapse rate (per timestamp? Hourly? Daily? Mean?) between the two stations for this estimation? It is notable that this lapse rate approach corresponded poorly to your on-glacier temperature observations (in the debris-covered area, Part A) – how do you think such an approach would correspond to the on-ice calculations?

This is good to clarify. Based on our understanding of melt factors (MFs) they are always relative to the temperature data used to tune the MF. And that off-glacier lapse rates are often a better representation of the lower 1 km of the atmosphere independent of the glacier. We could also just draw a vertical line in Figure 10a, assuming that temperature did not increase at lower elevations and the ice cliffs would still not compensate for the insulating effects of debris.

L218. Did you attempt to digitize ice cliffs in 1957 as well? The mention of ponds (and long-term change) is quite sudden, and should maybe be better integrated with the text.

We did not attempt to digitize ice cliffs, it would not be possible. We will better integrate this text but we feel this adds some good context to the work.

L219. 'insure' should be 'ensure'

we will change this.

L243. Should this be between 520 and 620 m? The fractional area is more meaningful than total area for understanding the cliff distribution.

Good catch we will discuss fractional area here.

L245. Importantly, this is of reported values.

We aren't exactly sure what is meant here but we could provide a more robust estimate of debris-cover from the other glaciers. We will search for this.

L247. N is too small to note any meaningful correlation between debris thickness and ice cliff coverage, unless you have a physical mechanism to implicate.

We agree that N is too small. But we do have a physical mechanism to describe shy this might be the case it is actually outlined more broadly in Part C. But we can reference that here and make it more clear in Part C as well. Emphasizing how debris has this effect.

This is another place where we feel that we need Part C. Here there is an interesting observation based on the data presented in Part B but we cannot explain this physical mechanism in Part B because we need to support the idea that the melt pattern follows Ostrem's curve before we can really get into the explanation of the physical mechanisms in Part C.

L255 and 265. I am still confused about the ice cliff melt rate distribution, for 2 reasons. First, it appears that at high elevations in the study area, it appears that your modelled ice cliff melt rates are lower than the modelled subdebris melt rates.

This is a good observation. And something we can improve in the manuscript. This is happening because are using empirical fits to the data. At the highest elevations where we measured ice cliff backwasting rate there were some ice cliffs associated with crevasses and thin debris that retreated slowly.

This is not plausible (or there would not be cliffs!). Second, the ice cliff melt rates in this region are also lower than for clean ice, which should be an approximate lower bound for ice cliff melt at all elevations: are nearly bare ice, but with surface debris well below the critical thickness (thus enhancing melt relative to bare ice, if we can neglect increased shading).

Also a good point. This also gets to the issue raised by this reviewer for our curve fits for backwasting rate with elevation. Perhaps a more physical way of representing the backwasting rate with elevation would be to pin the minimum melt rate of the ice cliffs to the bare-ice melt rate (6 cm/d) or use a uniform backwasting rate with elevation (at say the bare ice melt rate 6 cm/d).

While these changes will improve the analysis but we are confident, as this reviewer points out that these changes will not change the overall conclusions. Thank you for pointing this out.

L257. These rates correspond to a mean cliff enhancement factor of 1.72 relative to the mean subdebris melt rate, which I suppose is lower than anywhere else due to the thin debris.

That makes sense to us. This is a nice way to present the relative effects of the two.

L259. 'Dominates' is a strange term here. Certainly the reduced melt rate due to debris thickness is apparent, and debris thickness differences are more important than the difference in cliff density.

We can re-phrase this emphasizing that the melt rates follow more closely the sub-debris melt rate curve than the ice cliff backwasting curve (independent of the curve fit through the backwasting data).

L265. This appears to be a typo – the cliff melt rates are an order of magnitude lower than under debris?

We need to clarify the text here so it it more clear what we are referring to here. If you took all melt from ice cliffs in each elevation band and then calculated how much that lowered the entire area of the elevation band (ice cliffs + debris area) then ice cliffs lower the entire surface by the rates quoted in this sentence.

L271. This is again very disjointed to the rest of the analysis.

We will work to incorporate this better. We also feel that adding in streams to Part B would also feel disjointed and disrupt the flow of the manuscript.

Also, the low lake coverage in the upper ZMT makes a lot of sense as this area has steeper surface slopes in 2009 (Fig 2).

We agree with this.

L282. I appreciate consideration of the applicability and extension of this method to other sites/scenes. I think the biggest challenge for application to other scenes is that the tested parameter sets produced extremely variable results, and would need optimisation for every new site and image. There are also seasonality patterns to consider- cliffs often retain snow longer than the debris surface, for example.

We have not rigorously tested application of this procedure to other glaciers and scenes because our primary goal was to estimate ice cliff area on Kennicott Glacier for this study. It is plausible these processing parameters would perform well on other scenes of Kennicott Glacier, and perhaps for other glaciers as well. Illumination and sun angle can vary from scene-to-scene, and debris color can vary across different glaciers. However, Kennicott itself has several debris colors and textures, and the method does not appear to systematically differ in performance from one debris band to the next – this suggests the routine is not strongly sensitive to debris color. Varying illumination may change the ideal processing parameters, but the fact that the adaptive binary threshold is normalized by the brightness of pixels surrounding ice cliffs should mitigate sensitivity to this issue. That being said, you are correct that optimal performance could require training data (i.e., manually delineated ice cliffs) for a new scene to find optimal parameters. Manually digitizing ice cliffs in a few training areas is not incredibly time intensive, so we do not view this as a critical shortcoming of this method. We added text stating that the transferability of these processing parameters requires further investigation.

At L285 before "Using multispectral imagery…" add "The transferability of optimal processing parameters (both across time and space) requires further investigation."

L324. The variability in observed backwasting rate is considerably stronger than any bias due to the observation location – the question is really where the mean lies.

We agree with this and will provide additional analyses to address the curve fit through the backwasting data.

L340. The potential distal effect of these features is conceptually well understood to be via internal ablation along englacial conduits, but is not possible to validate at present. See Benn et al (2001, 2012, 2017), Sakai et al (2002).

We will take a closer look at these papers. We agree that these effects are related to englacial conduits. We are wondering further how then englacial conduit melt out relates to the rest of the debris-covered glacier system. We believe that internal ablation though has a small effect on overall thinning. It isn't clear what physically will produce the heat needed to cause the thinning rates observed from debris-covered glaciers.

L344. This section/paragraph feels orphaned. It is worthwhile to note that even accounting for the hypsometric distribution of cliffs, the spatial pattern still emulated Ostrem's curve (just with different effective thicknesses), suggesting that this concept might be useful as a proxy for the altitudinal SMB pattern even where cliffs account for 40% of melt – just not directly comparable to stake measurements.

Nicely worded. We will work to incorporate this paragraph in, because this is really meant to be a transition and set up for Part C where we take the melt curve and compare it with a number of other analyses.

L353. 'counter' - should be singular

L356. 'trend' should be pattern

Table 3. I don't think that Buri and Pellicciotti (2018) is the most appropriate study for Lirung Glacier for this purpose. Why the comment on EB below the table?

We will adjust the table accordingly.

Figure 2. It is not clear what the bars are in the upper left – is this the domain with supraglacial debris?

Oops that was dropped from the caption, yes it is the extent of continuous supraglacial debris transversely across the glacier.

Figure 3. This is a very nice conceptual summary! Can you include a pond or stream?

Yes we will add those.

Figure 4. In the caption for 'c', there is a reference to a 'black line' which corresponds to the 'solid' line I think.

Yes we will correct this.

Figure 5. This is a very nice summary of the method. Can you reproduce the same for the Sobel method to be included in the Supplementary Information?

Thank you. We have included a similar workflow figure for the sobel edge detection method in the supplements.

Figure 6. Panel (b) does not depict the bare ice area outlines as in (c).

We have added the bare ice cliff outline to panel b.

Figure 7. Nice depiction of the optimization. No colorscale is shown, though, and due to the different axis ranges, it is difficult to visualize the lowest Euclidean distance.

It is true that the different axis ranges make the Euclidean distance harder to visualize. There can be many more false positives than false negatives. In the limit, you can have as many false positives as you have pixels, whereas false negatives can only occur on ice cliff pixels. We chose the axis limits to show the range of possible outcomes rather than omit many data points to have the plot be at equal scale. We omit a color bar because there is already a lot going on with this figure, the colors are not crucial for understanding the figure, but rather facilitate visualizing distance, and we state the meaning of the colors in the caption.

Figure 9. See my comments in the text on line 255. Surely the lowest ice cliff melt rate (here 2.9cm/d) should correspond in space to the highest sub-debris melt rate (5.8 cm/d) – at the highest elevations. But then, the cliff melt rate should not be lower that the subdebris melt

rate – that makes little sense. This suggests to me that the linear parameterization of ice cliff melt with elevation may not be appropriate.

We agree and can adjust the 'most likely estimate' curve fits to reflect these physical realities.

Figure 10. Nice summary. Can you include a depiction of the cliff-only melt rates vs elevation in panel (a)?

This is a good idea. Yes we can, though we are somewhat hesitant to make the figure overly busy.

It is interesting that the cliff portion of melt is highest high in the ZMT, but still makes little difference in the mean melt rate profile.

Thank you kindly. It is precisely because of observations like this that we feel it is best to separate the manuscripts as we have. Such that there is space to discuss these items in more depth.

Also, it would be very meaningful to complete a version of panel (a) for the min and max melt parameterizations. Effectively these estimates are generous uncertainty bounds for your results.

The red bands are in fact the extreme uncertainty bounds. We need to better emphasize this in the caption and text because seeing that the red band as extreme bounds makes the analysis in section 4.2.1 much more compelling. Just need to clarify this for the reader.

Figure 11. Do you have a depiction of supraglacial streams (density or otherwise) to complete the picture?

We discuss streams in Part C and discuss feedbacks between the cliffs and streams there as we feel there is more room to discuss them there. We also note that we aren't sure what the actual effect of the streams are on ice cliffs. Because of this we feel that discussing streams in relation to ice cliff distribution as we do in Part C is a good idea as opposed to introducing them here. But this is a good item to consider as it would make for a more complete picture for Part B.

Figure 12. This is orphaned from the discussion and seems like an odd figure to close on.

Yes we see what you mean. We can move it to the supplement.

In addition to the changes described above we also propose these changes:

**Part B: proposed changes**

We feel that there is more than enough new material here for a stand alone paper, but in order to improve the manuscript we propose that we add these additional datasets/ideas to Part B:

- We will add additional text supporting the usefulness of our new ice cliff detection method. In the supplemental we will include additional satellite photos showing how ice cliffs tend to be darker than the surrounding debris so this method can therefore be applied on other glaciers. We will also compare our method with other approaches from other glaciers.

- We will present new DEM differences from 2007 to 2013. These dh/dt data show that the zone of maximum thinning remains in the same spot as for the period from 1957 to 2007.

We will also include additional laser altymetry data from 2007 that shows a similar thinning pattern.

   -This will address one of the main criticisms from multiple reviewers.

- We will introduce back-of-the-envelope calculations of the possible effect of englacial melt, sub-glacial melt, melt under pond surfaces, and melt by streams. This will clear up any issues related to this manuscript not being comprehensive with regards to melt hotspots.
  - We will not include stream digitizations in this manuscript because we cannot possibly digitize all streams on the glacier surface (imagery is too coarse). The streams play more into the feedbacks in Part C. We will instead make arguments about the surface area coverage of streams and their plausible effect on surface melt.

- We will use a uniform curve fit through the ice cliff backwasting data. And also explore the effect of other curve fits, producing different figure 10a and 10bs which we will put in the supplemental and discuss in the main text.

- Add in the paragraph description that links each of the three papers and helps guide the reader through each manuscript.
- Make sure it is clear how generous the uncertainty estimates already are in this paper. One of the reviewers missed these error estimates completely.
- Emphasize the increasing importance of ice cliffs under thicker and thicker debris.

---

## Author Response (AR1)

Thank you for your substantial contributions to the process and our manuscripts! They were very helpful. However, even after substantial revisions, we feel that Part C requires further work. As a result we have withdrawn that contribution. We have combined what was Part A and B to stand on its own as a singular submission. We greatly appreciate both the editor and reviewers managing the original three manuscript format. We further hope these changes will ease and simplify the future reviewing and editing of our submission.

Changes made to what was Parts A and B:

0

- Parts A and B are combined into one new manuscript following the editor's recommendation.
  - All sections of the text were updated following the reviewer comments
  - Uncertainties are reported in the text, figures and supplemental.
  - Thermal conductivities, on-glacier air temperatures, and degree-day factor methods have been moved from Part A to the supplemental materials of the new contribution.
    - Internal debris temperatures through time are plotted there.
    - A figure has been added showing the negligible effect of correcting in situ melt measurements to the full study period.
    - Curve fits are justified in detail, especially the sigmoidal fit to debris thickness.
  - Debris thicknesses were extrapolated down medial moraines
    - Debris thickness measurements are plotted down each medial moraine in the supplemental
  - Ice cliff backwasting is now applied uniformly through the study area for distributed melt estimates
    - Ice cliff backwasting is plotted by medial moraine and if streams or ponds are at their base in the supplemental
    - For ice cliff slope uncertainty we completed a compilation of published mean ice cliff slopes from previous studies. This allows for the establishment of uncertainty bounds in the distributed melt estimates
    - We also include in situ measurements of ice cliff slope taken from the summer of 2011.
  - Uncertainty estimates, very similar to those already in the original Part B, are now more clearly stated with the percentage of simulations that fall within the uncertainty bounds clearly stated.
  - 5 new sensitivity tests are applied to explore the robustness of our distributed melt estimates, following reviewer comments. They are in a new greatly expanded supplemental.
  - Thinning data from Das et al., 2014 are moved into the main text from the previous supplemental to support the stable location of the zone of maximum thinning up until 2007.
  - Most figures have been updated.
  - We have cut down on speculation and emphasized in some locations that we are presenting hypotheses.
  - Ponds were moved out of the contribution.
  - Conclusions have been expanded and placed in bullet points.

Replies to the four reviews of the original Parts A and B follow.

**Reply to Reviewer 1 Part A**

Review of "Debris cover and the thinning of Kennicott Glacier, Alaska, Part A: in situ mass balance measurements" by Anderson et al.

**Thank you for taking the time to review our manuscripts. You comments were very helpful.**

This study is the first part of three publications that investigate debris cover on Kennicott Glacier in Alaska. Given the limited number of studies that measure properties and melt rates of debriscovered glaciers, these measurements and results are important for advancing our understanding of debris-covered glaciers. This is especially true when one considers the limited knowledge of debris-covered glaciers in Alaska. The measurements and results are presented well. For the most part, the study is easy to follow, well-written, and has sufficient references.

There are a few sentences/paragraphs that could be modified to improve their readability though. The only major comment is to make sure that this study is discussing results that specifically pertain to this part of the three-part study. There are also a couple places where additional detail or analysis would provide useful context to the modeling community; however, this would only require minimal additional work. Therefore, I recommend accepting this manuscript for publication subject to minor revisions. Please see my detailed comments below.

Thank you kindly. We very much appreciate your efforts, especially considering you reviewed two manuscripts.

**Main Comments**

The reasons for studying Kennicott Glacier largely come across as reporting results across the three papers as opposed to stating what each paper does. For example, L55-57 state that the debris is thinner than most previously studied, but there is no reference to any studies concerning debris thicknesses on Kennicott Glacier. Similarly, L58 states there are more ice cliffs than those previously studied without a reference to a study that shows this. Hence, these appear to be results (and results from other papers) that are stated in the introduction.

**This is remedied by combining Parts A and B.**

Furthermore, the introduction states multiple times that the thinner debris increases the likelihood that melt hotspots will compensate for the insulating affects; however, thinner debris has higher melt rates, so it's unclear why melt hotspots would be more important for debris-covered glaciers with thinner debris because there would be less contrast between the sub-debris and ice cliff melt rates. If this is a hypothesis, then please state it this way. If this is supported by a physical basis, then please explicitly state this reasoning.

It is not about the relative contribution of hotspots to sub-debris melt but rather a comparison of absolute melt rates. That is a big point here that we will emphasize better. The absolute melt rate is what matters for the debris covered anomaly.

It is the net melt (sub-debris + hotspots) compared to the bare-ice melt rates at the top of the debris cover. Or another way to put it is: where is the maximum glacier-wide melt rate? And does it correspond with the zone of maximum thinning.

Lastly, the interpretation of the transverse variations of debris thickness appear to be poorly supported by the present figures and text. L135-139 state that mean debris thicknesses increase

near the glacier margins. However, site a appears to be closest to the center of the glacier, yet it has thicker debris. Similarly, site c is between sites b and d. Perhaps this is complicated by how far downglacier these sites are, but this needs to be elaborated upon. The same is true for the conclusion, where this is discussed. I would suggest removing this from the conclusion.

We see the reviewer's point, but the trend we discuss is also present. We have moved this material out of the new manuscript. We ultimately present distributed debris thickness estimates down medial moraines in the new manuscript.

Specific Comments Italics indicate suggested grammatical changes

L26 - use of "thick" and "thin" is a relative term. I suggest adding in parentheses what constitutes thick and thin.

We add this distinction.

L35 – consider "and, when thick, suppresses melt rates." or "and suppresses melt rates when thick."

**This sentence was removed.**

L39 – this sentence is missing its subject, so it's an incomplete sentence. Consider using a semicolon instead or adding the subject "Alternatively, this anomaly could be caused by...". Also, "or" and "alternatively" are repetitive.

**This sentence was removed.**

L41 – referring to the debris-cover anomaly here almost across as a result, i.e., Kennicott Glacier experiences the debris cover anomaly. If this is already known, then the reference should be added. If this is not known, then consider changing this sentence to give a broader overview of what's being done, e.g., constrain patterns of ... to understand the role of surface melt and ice dynamics on the surface lowering of Kennicott Glacier.

This is simply an observation derived from the data presented in Das et al., 2014, so we leave it in the introduction of part 1.

L55 - it's not entirely clear why thinner debris would affect the anomalous glacier thinning explained by melt hotspots, since thinner debris will have melt rates that are closer to clean ice. Also, are there previous debris thickness measurements of Kennicott Glacier? If so, this should be cited; otherwise, the fact that Kennicott Glacier has thinner debris than those previously studied is a result.

Thank you for the comment. If rapid thinning under debris cover is primarily caused by melt (hot spots + sub-debris melt) then we are most likely to see this effect where debris is thin and sub-debris melt rates are high. The basic logic we use throughout the former 3 parts is: if melt rates are the primary control on thinning then melt must also be maximized where thinning is greatest. Having thin debris already creates high melt rates, adding a high coverage of ice cliffs means that both components (hot spots and sub-debris melt) are extreme for Kennicott Glacier. We make this clear in the updated manuscript.

L60 – It remains unclear as to why thin debris increases the likelihood that melt hotspots will compensate for the insulating effects of debris.

It is very simply that thin debris reduces melt less than thick debris relative to hypothetical, local bare ice melt rates. If you have a glacier with thick debris cover melt suppression will be higher relative to hypothetical local bare-ice melt rates and will require a much higher contribution of melt from hot spots to compensate for the insulating effects of debris. This is about absolute melt total not the relative contribution of ice cliffs to sub-debris melt.

Conversely, the way the argument is stated sounds like melt hotspots cannot compensate for the insulating effects of debris on glaciers; however, because the debris on Kennicott Glacier is thinner, the sub-debris melt rates are closer to clean ice melt rates and hence the melt hotspots are less important because there's less of a difference to compensate for. The key here seems to be more on the sub-debris melt rates of thin debris than the melt hotspots. Please clarify this.

We are happy to clarify this in the text. Thin debris represents an extreme case where melt rates are already higher, adding on a high concentration of ice cliffs means that we are likely to get high melt rates in the debris-covered area. It is not about the relative contribution of hotspots versus debris, it is about the absolute value of melt.

L61 - typo "similar" should be "similar"

Fixed.

L72 – typo in the reported elevation range? Also, is there a reference for this data? RGI inventory perhaps?

Added the RGI reference and corrected the range.

L73 – consider "... and our study area, the debris-covered tongue of Kennicott Glacier (24.2 km 2 ), is only..."

Fixed.

L77 – be consistent with reporting elevations. Perhaps "Above 700 m a.s.l.". This should be done throughout the manuscript as well, e.g., L90, L131, L134, caption of Figure 1 "located at 1240 m a.s.l.", etc.

Fixed throughout.

L77-79 – is there a reference for these observations?

Based on our observations from travel on the glacier surface.

L86 – What do you mean by "Kennicott Glacier debris"? The debris properties? If so, state this "Because the debris properties of Kennicott Glacier have not been..."

**This was removed.**

L88 – consider "internal and surface debris temperatures, and …"Figure 1 – delete the ")" after panel b in the caption. Change to elevations to m a.s.l. May Creek meteorological station is not shown on the map. I suggest adding this – perhaps it is covered by

one of the legends.

We prefer not to add the May Creek station to this figure. We now add to the caption where the station is relative to McCarthy direction and distance wise.

Figure 2 – caption is unclear. "Dead" ice portion has daily mean surface velocities greater than 5 cm d -1 only during sliding events? Is this meant to be less than 5 cm d -1 with the exception of sliding events? Also, what does "and the observations of Rickman and Rosenkrans, 1997" refer to? Fix this reference.

We remove the dead ice discussion. The text is also clarified.

L107 – Avoid the use of unnecessary acronyms like LR for lapse rate. This only makes the manuscript less readable, especially for readers who may not be as familiar with a specific acronym.

We moved the LR to the supplemental and removed the LR acronym.

Figure 4 – The 4 panel figure is highly repetitive (e.g., shortwave radiation is shown in all 4 panels, and the MWS air temperature is shown in both panels). I would recommend using only 2 panels. Air temperature can easily show the 3 sites, and the two lapse rates can easily be shown on the same figure by using different colors or styles.

The problem is that the figures become too difficult to read following this suggestion. We are not sure that this is a big issue. We moved this to the supplemental.

L128 – This line doesn't make sense "at 109 locations at the same locations we also measured". Is it means to be two sentences? Otherwise, perhaps "around the locations where we measured ...".

**Fixed.**

Table 2 - is 0.001 cm an actual measurement? That is incredibly precise and thin for a debris thickness, which is hard to believe.

**Changed to 0 cm.**

L135-139 - It would be helpful to provide context to the specific sites (panels) for each of these sentences, e.g., "debris thickness did not exceed 15 cm (Fig. 6c)"

**Helpful thank you. We apply this throughout.**

L144 – Given the use of MF (used by Pellicciotti et al. 2005) instead of DDF (used by Hock 2003), I would consider either changing the "MF" to "DDF" or add the example citation of Pellicciotti et al. (2005). Note that in some fields MF or DDF could refer to multiplying multiple variables. I leave it up to the authors as to whether they want to maintain this original convention or adopt newer uses of it (e.g., degree-day factors shown as f ice (Radić and Hock, 2011)).

**Also very helpful. We now use DDF everywhere.**

L148 - Why the use of off-glacier air temperatures when you have data from on-glacier air

temperatures? It would be interesting to see the off-glacier air temperatures over the same period of time – perhaps this could be added to Figure 4 as this would provide some indication of how much the debris warms the air temperature?

We use the off-glacier air stations because we do not have measurements for the full time period of the field campaign from on glacier. There is a local station in McCarthy but it is not automatic and is recorded only during work hours for the airport. For melt factors it is also common to use off glacier sites and they actually perform better than on glacier sites often times. The idea is that on glacier sites are affected by the ice surface itself but really what is controlling available energy for melt is the integrated temperature from the lower 1 km of the atmosphere.

In addition to this the meltfactor correction provides a minor correction to the melt rates. We include this to be complete and correct measurements for difference measurement intervals.

L153-157 – Given the impressive amount of data collected, it is disappointing that the authors do not provide a "best-fit" Østrem curve for comparison with other sites. While there is considerable variability in surface lowering, especially over thin debris that is dependent on local conditions as the authors state, this is clearly something that would affect all previous curves. Is there a good reason the authors did not do this? This could be a highly beneficial product for modelers. If uncertainty is the issue, the authors could easily add uncertainty bounds to the curves.

A curve fit is in the old Part B. The curve fits and extreme curve fits are clear in the new manuscript. The curve fit parameters are also shown.

L176-177 – What does the "mean" debris surface temperature refer to? Is this the mean temperature over the entire study period (at least one week) or was this used to estimate conductivity on a shorter time period? I assume it is the former, but it may be good to be explicit, e.g., "... we then calculate K e for each temperature profile over the entire duration of the temperature measurements.". This would avoid any misunderstandings because the effective thermal conductivity could vary over time, e.g., if there was a change in debris moisture.

Thank you for this comment. We changed this in the text. It is the mean temperature over the entire study period (at least one week). This is moved to the supplemental material.

Figure 9 – Why is there a point for a debris thickness of 0 with an effective thermal conductivity of 0 W C -1 m -1 ? This seems to be unphysical. I also question the "nonlinear" increase in thermal conductivity as a function of debris thickness. There appears to be a fair amount of scatter such that a linear fit might also produce a reasonable fit? Furthermore, if the (0,0) point is discarded, then the linear fit will likely cross the x-axis around 0.4 - 0.5 W C -1 W -1, which is near the lower range of that estimated based on physical constants (L181; Nicholson and Benn, 2006). Hence, this would be more physically based. Lastly, why is thermal conductivity varies due to debris thickness and not the other way around. Hence, the debris thickness is the independent variable (typically plotted on the x-axis) and the thermal conductivity is the dependent variable.

These are all good points and we remove the zero point and curve fit. Moved all conductivity methods and results to the supplemental.

L181 – I question "The apparent non-linear increase". See comment above. It would be good to at least see a linear fit as well.

**We remove the non-linear fit.**

L182 – typo, "may be due to..."

L185 – it would be valuable to make assumptions concerning the specific heat capacity and porosity such that a comparison could be shown for the differences in thermal conductivity based on the method.

This is a nice suggestion but we are just moving this data to the supplemental.

L206 – type "were made..."

L205-206 – were these debris thicknesses already known from the previous debris thickness and ablation stake measurements or were these new measurements? Furthermore, how many "data points" were collected?

L214-218 - why the switch from backwasting rates to backwasting melt factors? It would be easier to read if it were consistent. Backwasting melt factors are only discussed in the supplemental now.

Figure 12 – caption, "based on the individual melt factor..."

**Moved to the supplemental.**

L227 – shouldn't have to restate acronym, although see previous comment about removing it altogether.

**Removed the acronyms.**

L233 – "related to the large areas..."L245-248 – consider changing these sentences so that two sentences in a row don't start with "But..." as this should make it easier to read and understand.

**Text revised.**

L255 - Please state the percentage of debris thickness measurements that were derived from the top of ice cliffs to provide the reader with some sense of if this was for 50% of 100% of the measurements. "The majority (X%) of our debris thickness measurements..."

**This % was added to the new manuscript.**

L278-279 – This sentence about Part B is confusing. What does estimate if ice cliff melt rates correspond to the location of maximum thinning under thick debris on Kennicott Glacier mean? Is "under thick debris" meant to refer to the debris-covered glacier? A specific part of the glacier? Or literally the areas where the debris is thickest? I assume this is generally referring to the debris-covered glacier, but please clarify to avoid confusion.

**Removed.**

L281-285 - Is (1) different than (2)? Or is the poor representation of air temperature due to using the off-glacier meteorological data, which does not account for the variations in air temperature above the debris? Also, having sentences in the middle of these various points is very hard to read. I would suggest making these three separate sentences.

We see what the reviewer means. This could be clarified with a bit more explanation. This section was moved to the supplemental material.

L285 – What does this sentence of the portion of fine material have to do with ice cliffs? This seems very out of place and appears to refer to the section on thermal conductivities.

This section is actually about ice cliffs (3.4 Ice cliff backwasting). Just needs a bit more of a clear explanation. The section has been re-written to clarify our meaning here.

L297 - missing Oxford comma, which seems to be used throughout the rest of the manuscript

**Corrected.**

L300 – "transverse debris thickness patterns broadly correspond with surface velocities" is out of place and perhaps meant for paper B or C. This paper showed no data on surface velocities.

**We moved out of the new manuscript.**

L302 – may want to acknowledge the limitations that were described in the discussion, i.e., that most debris thickness measurements were from on top of ice cliffs and so caution should be used when using these for tuning and validating distributed debris thickness estimates as they may underestimate the actual debris thickness.

We now report the % of debris thickness measurements from the top of ice cliffs. That way the reader can decide.

L305 – reconsider "non-linear" relationship. See comment above. Furthermore, is the larger point that "water" or "porosity" plays an important role in heat transfer? They are certainly related to one another, but most of the discussion seemed to focus on the role of finer debris and porosity. This should be consistent in the conclusion.

We removed the non-linear relationship. All conductivity methods and results are moved to the supplemental.

L308 – there is no evidence in this paper that the ice cliffs counteract the insulating effects of thick debris. More appropriate would be to summarize how the backwasting melt rates compared to the sub-debris melt rates. If this is a conclusion from Part B, then it belongs in that paper.

This is a general statement not a specific statement about the Kennicott. But we clarify this issue throughout.

References (thanks for including these. Very kind)

Pelliciotti, F., Brock, B., Strasser, U., Burlando, P., Funk, M., and Corripio, J. (2005). An enhanced temperature-index glacier melt model including the shortwave radiation balance: development and testing for Haut Glacier d'Arolla, Switzerland, Journal of Glaciology, 51(175):573-587.

Radić, V. and Hock, R. (2011). Regionally differentiated contribution of mountain glaciers and ice caps to future sea-level rise, Nature Geoscience, 4:91-94.

**Reply to Review 2 Part A**

Review of 'Debris cover and the thinning of Kennicott Glacier, Alaska, Part A' by Leif Anderson et al., under consideration for The Cryosphere

**Thank you kindly for taking the time to review our manuscripts. Your comments greatly improved our work.**

The manuscript by L Anderson, et al., presents a variety of field measurements on debris-covered Kennicott Glacier, and characterises the debris properties and melt rates under debris or at ice cliffs. These data are an extremely useful contribution to understanding of debris covered glaciers in distinct settings. Very few measurements of debris-covered glaciers are available in Alaska, despite the extensive debris coverage of glaciers in the region. The data presented cover an extensive set of topics, and will be useful in calibrating and applying models developed for other regions to Alaskan sites.

Although there are only minor points of criticism relating to the data presented, the manuscript at present lacks cohesion. The results from this manuscript are key in laying the foundation for Parts B and C of the study by Anderson et al, but I can't shake the feeling that this would better fit as (largely) supplementary material for Part B, or as a submission to the EGU journal Earth Systems Science Data; the content is unusual for The Cryosphere. In the latter case or if the manuscript will remain as an independent paper in The Cryosphere, I would recommend expanding the discussion of the varied data collected; some opportunities for expanded discussion are identified in my comments below.

**This manuscript (Part A) has been combined with the original Part B following this reviewer's suggestion.**

**Major Points**

As a presentation of diverse field measurements, the manuscript lacks a storyline. I appreciate the effort and value of collecting these measurements, but there is no methodological development, and the results and discussion seem geared towards briefly placing the measurements in the context of observations in High Mountain Asia. The few major outcomes (e.g. aspect dependence of ice cliffs) are not investigated or discussed in much detail, as it is very clear that these measurements are geared towards supporting Part B. Consequently, I feel as though many of the results could be included in Part B without a separate Part A; rather by including these measurements as supplementary material, as they follow more-or-less established methods.

The manuscript organisation is awkward at times. In part this is because measurements and results are presented together, but also because figures are not always associated with the text that pertains to them. More problematic is the lack of an integrating discussion – the individual measurements are discussed but there is not much of a summary characterisation of Kennicott. I appreciate that this is difficult to do from such diverse field measurements. Again, this is in part because the paper is unusual for content in The Cryosphere, and this is another reason why I think this work could be integrated into Part B (or as a manuscript in the EGU journal Earth Systems Science Data, rather than a distinct manuscript.

Data availability. In the modern spirit of open data, I would strongly recommend that these measurements be archived in an open repository.

Off-glacier air temperatures are used to correct short-period met measurements to the full period of

record, but these stations have been shown in this manuscript to represent entirely different altitudinal temperature differences compared to on-glacier stations. The use of the off-glacier stations needs to be robustly evaluated at the stations, and the on-glacier stations need to be used to determine melt factors (for the on-glacier air temperature subperiod). Even if this does notchange the pattern of relative melt factors, this represents a (possibly major) uncertainty in all of the analysis.

Thank you for your comment. The DDF approach is always relative to the station data (be it onglacier or off-glacier). Because we have measured melt rates we can optimize the melt factor for each ablation stake. On-glacier weather stations often perform worse when applying DDFs. We have moved this discussion into the supplement. A figure in the supplemental of the new manuscript shows how small the DDF correction is.

Furthermore, the on glacier temperatures are effected by the glacier surface. While on-glacier sites are best for energy balance approaches, off glacier sites have been shown to perform better when using DDF approaches (Ohmura, 2001; Wheler et al., 2014).

It is actually, as well, and exceptional occurrence that there is a meteorological station above the study area. In Alaska, this is a rather special occurrence. In short we have already made the best correction of theses data possible with what is available and this is simply a negligible issue.

Uncertainty in measurements or calculations is not considered at all in the manuscript. Since these measurements are used in two linked following studies, and to draw important conclusions about the dynamics of debris-covered glaciers, I think it is important to frame the results in terms of uncertainty from the start.

All uncertainties are now described in the new manuscript for all measurements and the supplemental material.

Minor Points L34. 'when thick it supresses melt rates' – although common knowledge, it is worthwhile to specify a reference here

We add a citation here in the new manuscript.

L41. Not just explain but also examine; we have evidence of the 'debris-cover anomaly' in High Mountain Asia but not before in Alaska, to my knowledge.

The DC anomaly is present in Das et al., 2014. We just highlight that in the introduction here.

L53. Missing 'glacier' - debris-covered glacier mass balance

Fixed.

L55-64. I agree that Kennicott is an interesting case, and a great opportunity to examine the debriscover anomaly. However, I don't entirely agree with these two justifications in their present form, possibly because a bit more explanation is needed. The presence of thinner debris means that there is less melt enhancement due to cliffs and ponds (ie they may not melt much 'more' than the subdebris ablation), even if their areal coverage is extensive. Your implied point is that the thin debris should lead to less of a melt difference between clean and debris-covered areas, and so the chance of cliffs/ponds/other mechanisms to make up for this is greater. That needs to be made explicit; at present the second rationale is unclear. Thank you for pointing this out and David's review has a similar comment. The point here is that the relative contribution of ice cliffs versus sub-debris melt is entirely irrelevant from our perspective. Rather what matters are the absolute melt rates.

We clarify this in the new manuscript.

For the third rationale, it would be beneficial to identify the actual density of ice cliffs in the study area (although this is an output from part B).

This is remedied by combining Parts A and B into one manuscript.

Readers should not have to jump between the manuscripts to understand the rationale.

This is remedied by combining Parts A and B into one manuscript.

L80. The reference to Mount Blackburn does not fit into the text very well – what is the relevance to Kennicott? Debris supply mechanisms? Lithology?

It is simply an important local landmark. But we remove it.

L83. The multiple clauses with commas are a bit awkward.

Fixed.

L88. For consistency, this should be debris internal temperature and debris surface temperature.

This is moved to the supplemental and corrected.

L93. I suggest changing 'vary' to 'differ'. Boundary layer conditions also vary widely for debrisfree glaciers, and for debris-covered glaciers; without a doubt there is overlap in this variability, but the distributions of conditions differ, which is your point.

This is moved to the supplemental and corrected.

L106. It would be good to include a very brief description of this important transition, or to simply state that this location is at the base of a prominent bulge. It would also be useful to refer to readers to a more specific area of Part C.

**This was moved to the supplemental.**

L107. These lapse rates are extremely steep, which makes me wonder if the positions themselves are sufficiently representative of the glacier surface. As elevation tends to be a less direct control on air temperatures over debris, I would recommend fitting the regression to all three observations at once (rather than a 2-step regression).

This is moved to the supplemental material and leave the analysis as is. We just reference the data in the main text. This suggestion could be fruitful, we just mention the local conditions now instead of adding new analyses.

It is highly likely that topographic prominence and proximity to water are both controls on both wind and air temperature over debris (e.g. Shaw and Steiner publications, also Miles et al, 2017 [Frontiers], Supplementary Material).

These are great suggestions, thank you! Here we really highlight the proximity to wide stretches of bare ice, which is the case on the Kennicott, unlike many previously studied DCG. We add this to the text in the supplemental of the new manuscript.

L114. 'was' should be 'were' as LRs is plural.L128.

Move to the supplement, fixed.

It is not clear from Figure 2 which are the 109 locations with debris thickness measurements, as there are more than 109 points when combining sub-debris melt, ice cliff backwasting, and debris temperature.

A single debris thickness measurement may represent several backwasting rates if they are measured close to one another. The same applies to ablation stakes which may be measured multiple times.

We drop the surface temperature measurements from the body of work because they emphasized the measurement of thick debris in order to increase the range of debris surface temperatures made.

L130. It would be good to identify these thinner debris positions (especially those with multiple measurements) spatially in Figure 2, rather than just with elevation.

This will make the figure too complex. And there is already a ton in these manuscripts.

L136. The presentation of these data seems to occur with Figure 7, which is not mentioned here but is quite a jump through the paper.

Fixed with combining Parts A and B.

L140-142. Were repeated sub-debris melt measurements made at the same positions? Did the debris thickness change when re-exhuming the stakes? What uncertainty is there in your debris thicknesses or melt rates due to the removal and reburial of debris? (Especially if this occurs repeatedly). A key consideration is that supraglacial debris often presents as sorted, but it is extremely difficult to replace debris in the same state which it was found. This of course is not a problem unique to your measurements, but it should be acknowledged and considered.

Yes this is a potential issue to all sub-debris melt measurements. Is there a citation showing that this effect actually matters for sub-debris melt rates? It seems like a potential minor issue. We consider it negligible but briefly mention it in the supplemental.

L145. This melt factor determination negates SW and LW inputs (and their variability), which may be very important for debris covered glacier surfaces (e.g. Reid, Steiner, Buri ice cliff studies, also Carenzo et al 2016). Although this may not affect your overall results in terms of total melt, it will definitely affect the aspect dependence of subdebris and ice cliff melt. Also, this is clearly determining the mean melt factor for each location; how variable were different melt subperiods for each site?

To us, the degree-day factor (DDF) approach we already use includes these aspect effects. If the melt is higher in a southerly direction then the DDF would be higher. If a north facing ice cliff retreats slower than the MF would be lower. The SW and LW effects may be able to produce more accurate estimates of melt but that would play more of a role if cloudiness changed and the relative effect of SW to LW fluxes changes. But a simple MF approach would also include these effects if the relative effect of SW to LW changes as well.

Our approach here is not to use the most sophisticated melt model possible, that requires may more data input (we are in a relatively data poor region for glacier studies and have no access to these fluxes locally) and increased constraint of parameters. The simple approach used here is effective, and please note how small the corrections are in figure 12 of the original Part A. It won't matter which melt model we use the change will be small because we are correcting the rates for a difference of a few weeks between individual measurements. But the differences between melt models is a worthy target of research.

L148-150. Please explain this estimation of T\* more clearly. Are you using the LR between the two off-glacier stations to estimate T\* at each location? If so, this estimation needs to be further evaluated relative to the multi-step on-glacier LRs (for the shorter period of measurements for those stations), which differ considerably for the environmental lapse rate.

Yes we are using two off glacier meteorological stations which has been shown to provide good estimates of melt. As far as we understand off glacier sites provide a better sense of the temperature of the lowest km of the atmosphere which works well for predicting melt. Note too what we use these DDFs for: It is just to correct difference in measurement period so we are deriving DDF from a couple of weeks to estimate melt for another couple of weeks. The effect on backwasting rates is very small and does not change the story here or in part B, even if we used a more complex model we aren't convinced that anything would change, because of the increase in parameter uncertainty.

These air temperatures were moved to the supplemental.

At present, the dependence on off-glacier measurements is not very robust, as your on-glacier air temperature measurements indicate a significant deviation from off-glacier air temperature spatial variability. This will have the effect of smoothing your ice cliff MFs with elevation.

We do not see how this really matters. We use the closest, viable meteorological station. The difference in temperatures observed from the on-glacier stations will be included in differences in the DDF between sites. See Wheler et al., 2014 article on the use of DDF from Canada. That is the advantage of the DDF approach. The DDF includes all these differences in physical variability. Even if an energy balance model includes all the Energy transfer pathways the number of parameters skyrockets such that the issue becomes the constraint of these parameters. If we were using an energy balance approach then yes we need on glacier temperatures, but we are not and we feel that based on a number of studies this approach is well justified.

L156-7. This is an interesting comparison, and should be explored a bit in the Discussion. Is this due to latitudinal controls on Ta or SWin? Presumably these glaciers have differing lithologies, and they certainly differ in climatic setting, so perhaps this is a coincidence? I note that there is still a factor of 2 difference between the other glaciers.

We agree that this is interesting and it is now included in the supplemental.

L161. This is not shown in Fig 2.

**This figure was moved to the supplemental.**

L176. Please justify the use of a linear extrapolation to surface temperature, which differs from interpretation of many debris internal temperature profiles I've seen (often an exponential form is noted when there are sufficient thermistors).

When integrating over more than a week the temperature profile becomes linear when heat is transferred by conduction (Conway and Rassmussen, 2000).

It would also be good to include 1-2 plots of the internal temperatures – diurnal variations and means.

This is new plot is included in the supplemental.

L181. I have some qualms with the 'non-linear' increase, which is only because you have imposed (0,0) as an additional point for your fit. Surely, an infinitesimally small debris thickness (which is of course unrealistic) should converge on the thermal conductivity of the rock material itself (i.e. no longer an effective conductivity, but the true conductivity of the material). If you neglect the (0,0) point, this looks most like a linear trend crossing the x-axis at about 0.4 W (C m) -1 . Also, I think that the non-linearity, if true, needs more consideration and discussion – what are the effects of sorting, for example? Does this imply a bulk density difference between the upper and lower debris layers?Also, what do you expect conductivity to look like for layers thicker than 1 m (e.g. these would exceed the range estimated by Nicholson and Benn (2006).

We remove the non-linear fit.

L199. It would be good to show the distinct lithological mixes in Figure 9.

We are no longer discussing this.

L205. Please indicate the accuracy of the Fluke Infrared Thermometer.

We removed the surface temperature data from all Parts.

L204-208. This section does not clearly follow the past sections, and also does not integrate very well with the rest of the study at present.

**We moved this to the supplemental.**

L216. Did you classify cliffs based on the presence of streams as well? Part of the results of Brun et al (2016) and others is that any moving water can have the same effect as ponds. In my opinion (not demonstrated) supraglacial streams are even more effective cliff maintenance mechanisms.

We did take notes on the presence of streams at the base but we found no correlation with the backwasting rate. Ice cliffs with lakes and streams are now designated in a figure in the new manuscript.

L223. It is worth considering these climatological and latitudinal controls in slightly more detail. Is Kennicott really cloudier in the melt season than Lirung (site of Buri and Pellicciott, 2018)? The latitudinal control is not unexpected, but deserves more consideration. Effectively, during the

ablation season there should be less diurnal variation in solar zenith angle at high latitude (solar zenith and azimuth are of course correlated seasonally at any latitude).

We no longer consider this in the manuscript.

L233-234. Both instances of 'effected' should be 'affected'.

Corrected.

L264. Are these the (unmodified) measured melt rates or your estimated melt rates from section 2.3?

It doesn't matter which. We will include a plot of measured versus corrected melt rates in the supplemental. The points virtually plot on top of one another the changes are tiny.

L265. The comma here is awkward. Perhaps use 'as compared to'

Removed.

L273. This was only demonstrated for north-facing cliffs in Buri et al (2016b).

Does the reviewer mean south facing? Since north-facing ice cliffs are preserved.

L282. I agree that the representation of air temperatures from off-glacier stations is not robust. This deserves careful comparison of estimated air temperatures from lapse rates derived from your on-glacier stations (for the shorter period) before an extrapolation across the glacier. More importantly, this could lead to a major uncertainty in your MFs for both debris and cliffs, even if the patterns do not change with more realistic air temperatures. At the very least an evaluation of the accuracy of the off-glacier stations for representing the on-glacier observed air temperatures is needed.

We disagree. There is no need for the off glacier temperature to be compared to on glacier sites. DDF are relative parameters only relevant to the air temperature measurements. In addition we could also not do the DDF correction and the melt rate results would be almost the same.

As long as the DDF derived from the air temperature data we used is then used with air temperatures from the same stations the MF extrapolation is viable. We feel that this point is over emphasized. DDF and lapse rates are only relative to the temperatures at station.

See Wheler et al., 2014: Effects of Temperature Forcing Provenance and Extrapolation on the Performance of an Empirical Glacier-Melt Model

We move the on-glacier air temperatures to the supplemental.

L304-307. This list of summary statements is not terribly satisfying, and feels like a list of bullet points. More interesting is whether Kennicott's debris properties generally fit within the range of previous distributions (they seem to) which is meaningful as there are few published debris properties in Alaska generally. At the very least, it would be nice to have some numbers in the text?

These conclusions no longer matter with the re-combination of parts A and B.

Table 1. The estimated debris surface temperature difference is not described in the text.

**Moved to the supplemental.**

Table 2. I would describe the contents of this table as 'measurements' rather than 'variables'.

**Changed.**

Table 3. It seems odd to choose Buri and Pellicciotti (2018) to represent Lirung, as that study was primarily modelling synthetic cliffs rather than reporting backwasting measurements. I think the most appropriate study here would be Brun et al (2016).

**Reference changed.**

Figure 1. At what interval are these contours?

**Fixed.**

Figure 2. It would be useful to identify the sources and dates of the WV and aerial imagery in this caption or in the text.

**All images are referenced now.**

Figure 3. I like this schematic, but it's not quite complete: missing are the thermistor strings and air temperature measurements (possibly others). Also, it would be fantastic to include some field photographs demonstrating the measurements.

**This is moved to the supplemental.**

Figure 4. Since you rely on the May Ck and Gates air temperature measurements, it would be very beneficial to show them here. Perhaps it would also be possible to combine panels (a) and (c), and (b) and (d).

We feel that combining the panel will make an unintelligible figure. We could add in the off-glacier data but we aren't sure how it really matters. DDF are all relative to the temperature data they are derived from as long as data from the same stations used to derive the DDF is used for extrapolation the principle holds. There is no absolute DDF it is always relative to the temperature data.

Figure 5. Can you indicate the lithology of the debris thickness in panel (a)?

We did not quantify the lithology, it is not mentioned in the new manuscript, except briefly in the supplemental.

Figure 6. This seems to be referred to out of place in the text. Also, I'd suggest switching the axes (so that elevation is the y axis) for easier comparison with Figures 1 and 5.

**We move this to the supplemental.**

Figure 7. I didn't catch a description of the bare-ice melt rate – what elevation was this at? In addition, this content is almost entirely repeated in Figure 8, so I'd suggest eliminating the figure, but depicting the bare ice melt rate in Figure 8.

We will clarify bare-ice melt rate, it is described in the manuscript now. This data is only shown in one plot now.

Figure 9. As described with my comment on L181, I don't think the point at the origin is justified, in which case a linear fit is entirely appropriate. Also, I'm a bit disappointed that we don't see any of the thermistor data!

We will remove the point at 0 and the curve fit. This is moved to the supplemental as well.

Figure 10. I would suggest to merge this with Figure 9, as the content is very closely related. Also, I note that the units here  $(m \ 2 \ s \ -1)$  differ from that in the text  $(mm \ 2 \ s \ -1)$ .

Moved to the supplemental.

Figure 11. Over what time period were these temperature measurements taken?

From 10 am to 4 pm. We remove these surface temperatures from the contribution.

Figure 12. Is it possible to identify the cliffs that bordered ponds or streams within one of these panels?

This is now included in the new manuscript.

**Reply to Reviewer 1 Part B**
Thank you kindly for taking the time to review these manuscripts. Your comments greatly improved this work.

In this second opus of their trilogy, Anderson et al. deduced the spatial pattern of melt due to ice cliff and under debris, and consider the distribution of supraglacial lakes to conclude that melt hot spots (cliffs and lakes) are not sufficient to explain the pattern of rapid thinning on Kennicott Glacier.

Overall this is a series of paper that bring a lot of new data and contribute to show that melt hot spots (ice cliff and lakes) only modestly contribute to the overall mass loss of a large debris covered tongue. A clear achievement has been to perform such measurements on a very large glacier in Alaska and proposed methods to extrapolate the point wise measurements to the overall debris-covered glacier tongue.

**Thank you for the kind summary.**

General comments for the three papers.

1/ I am not convinced by the need to split this paper into three parts. It implies lot of repetitions and also mean that the reader as to refer to other parts of the article which is not convenient.

Some data are plot several times in the three articles (debris thickness, dh/dt for 1957-2009 etc. . .) I think the authors missed here an opportunity to put everything together. Specifically in this part B, the discussion (section 4.2.1) whether ice cliff or debris can explain the zone of maximum thinning would be much more straightforward if Part B and C were merged. Right now this discussion is a lot of speculation to finally justify the need for a part C.

**We appreciate this perspective on the body of work and we have combined Parts A and B into one manuscript.**

2/ One strong limitation (that needs to be emphasized more) is that field measurements over a short period of time in July 2011 are used to interpret a map of elevation change measured over a multidecadal time period. Authors need to recall to their reader that their results apply to 2-month period in summer. The whole discussion would have been much more meaningful if the elevation changes were also measured for the same time period where surface melt features are studied (but the DEM data are probably not available. . .).

The field data is from June to August 2011. We actually have dh/dt estimates that cover the time period derived from ArcticDEMs. New dhdt estimates are provided in what was Part C in another contribution to be re-submitted. The zone of maximum thinning is in the same location as the dh/dt maps from 1957 to 2004, 1957 to 2000, and from 2000 to 2007 from Das et al., 2014. These plots from Das et al., 2014 were in the supplemental material, but are now in the main text of the revised manuscript.

In the discussion of the new manuscript we focus on just the measurement period in 2011.

General comments for part B.

3/ I miss a more thorough description of and comparison to earlier studies mapping ice cliff automatically. In particular Kraaijenbrink et al., RSE, 2016.

We now cite this paper and provide a broader discussion of ice cliff mapping approaches.

4/ I feel it would have been very interesting to see an evaluation of the ice cliff mapping algorithms using independent dataset, for example the Ragletti/Steiner cliff dataset on Lantang Glacier. Maybe this ice cliff automatic mapping part would have deserved a dedicated article, and all the rest of the results would then fit a single contribution?

Interesting point, but ultimately this paper is about evaluating ice cliff extent on a single particularly difficult glacier. If an independent, reliable validation dataset was available for Kennicott Glacier we would use it. But we have already put in a substantial effort to validate this method, having digitized  $\sim 7 \%$  of the entire surface.

5/ Uncertainties could be treated in a more systematic way so that results in the end should all be quoted together with their range of uncertainties. This applies to all three parts.

Please note that Figure 10 already includes a generous, extreme range of uncertainties, that has been over looked by this reviewer. We report errors more clearly throughout and added a new heading titled uncertainty in distributed melt estimates. We also added 6 additional uncertainty cases in the supplemental.

"results in the end should all be quoted together with their range of uncertainties."

We consider this to be a style choice. We provide extreme uncertainty estimates because our goal is to determine if ice cliff + sub-debris melt is maximized in the zone of maximum thinning. Including too many different ways of representing uncertainty will confuse the reader. So we place these additional uncertainty estimates in the supplemental.

6/ When authors provide % of melt, they should always make it clear that this is a percentage of the debris-covered tongue (and not the whole glacier!)

We have clarified this.

Specific comments.

L16 What does "enhancing" the mass balance mean. A mass balance can be increased or reduced. Is this formally demonstrated? I thought it was debated.

We will clarify 'enhancing' as suggested. In this case just changed to 'increasing.'

L21 "Total" is ambiguous. Tongue-wide or glacier-wide?

**We will clarify this.**

L41 One does not expect results in the introduction.

The statement we make is based off of previous work from the area so it is more an observation to set the stage for the rest of the manuscript, not our results.

L51 "surface mass balance" would be a more appropriate way to refer to it

See text immediately below L51.

Eq 1. x,y are not defined. Fixed.

L58. So do the authors neglect them? It should be stated unambiguously.

We state that we neglect them.

L88ff. Splitting the article into three parts leads to many repetitions such as this section. Problematic in my view.

A and B are now a single manuscript.

L160. Unclear (understated) what meteorological data would have brought, if they had been available.

**Removed.**

L169-170. This statement that 20% of ice cliff area need to be added is enigmatic at this stage in the paper.

**We will clarify this.**

Eq (3). How the type of fitting curve was chosen? It seems to come from nowhere. Can it be justified?

We will provide a proper method of justification for the curve fitting based on error metrics. Ultimately, the details of the shape of the curve are secondary (linear or non-linear) to the melt suppression effects of debris. We have added significant supplementary data and plots to show how the specifics of the curve fit do not effect our final conclusions.

L183. I see in part A that your cliff backwasting rate neglect emergence velocity. This needs to be justified.

It is not clear what the reviewer is referring to here. The emergence velocity is not relevant to our backwasting rage here because we measured the rate in situ, on the glacier.

L191. A statement such as (here) "based on an analysis of 2-m ArcticDEMs" is too vague.eq (7). Ice ciff area. Is it planar or real area? I think "i" must be added as superscript

with bdot debris and bdot icecliff

This has been clarified and fixed. The 'i' superscript is not needed because the melt rates are varying pixel to pixel within elevation band 'i.'

L204. I do not understand why fitting a curve through 25% or 75% of the data points leads to "extreme" cases. Not clear. Why not a curve containing 67% of the data (to have 1-sigma uncertainties). See my general comment about treatment of error bars.

We use 25% or 75% of the data points for debris cover because using 1-sigma uncertainties would result in negative debris thicknesses. There is really no way around this for debris as it varies with elevation.

In the uncertainty estimates we now use 1-sigma bounds for ice cliff backwasting, ice cliff slope, and sub-debris melt. As stated above we don't see away around the Interquartile range approach for debris thickness.

L226. "error checks" is a strange terminology. Why not "validation dataset"

**Adjusted.**

L240. Percentage should be 21% and 31%, right?

Fixed.

L244. Where does "11.6%" come from? I read 11.4% and 11.7% above.

Typo, fixed.

L245. This raise the question of whether all studies defined the "debris-covered tongue" the same way. Did the authors check carefully previous studies for this aspect?

We clarify what we mean by study area early on.

L247. "This implies that ice cliff coverage varies with debris thickness". This seems like a hasty conclusion. . . other example from the literature to support the statement?

There are not many studies that quantify ice cliff distribution and we don't see anything wrong with highlighting what 'could' be a trend.

L257. One expect an error quantification for each term (81% and 19%).

The next sentence does just this for the 19% number, which also does the same for the 81% number so the error quantification is already present.

L268. "Across all of the elevation bands, the ice cliffs between 500 and 520 m generate a maximum of 40% of the total mass loss due to ice cliffs and sub-debris melt." I am not sure I got the meaning here. Maybe reformulate for clarification. (it is clear from the figure, just a text improvement)

**Clarified.**

L273 "within" rather than "with" (I think)

Section was moved out of the new contribution.

L289. 19%. Lack error bars and also authors need to remind that this applies to a short period of time during summer 2011. So they cannot draw such broad conclusion.

19% is the most likely case. We choose not to overwhelm the reader with uncertainties quoted in every sentence. The uncertainties were already quoted previously. We also add 6 additional cases to explore the uncertainty.

We do not attempt to extrapolate the melt rates beyond the summer of 2011 now.

I would be curious to see a comparison of this number to the total glacier-wide ablation during this period if available. Is not it just a few percents? Do we need to really worry so much about ice cliffs for glacier-wide or region-wide application (and future projections)?

We could add in this analysis but we aren't sure how it would improve this study and the aims we outlined in the introduction. As we framed the study, we are interested in explaining the thinning patterns of debris-covered glaciers, and Kennicott Glacier specifically. Ice cliffs are an important, proposed contributor to this. Thinning patterns hold implications for longer term thinning patterns and hazards.

L307. the SMB cannot be "suppressed". It can be increase or decrease. (SMB is increased here, or less negative)

Changed to 'melt' instead of SMB.

L321. "This required backwasting rate is well beyond potential biases introduced due to the summer of 2011 having anomalously low air temperatures". Statement not really explained and justified.

What we mean is that the weather during the summer of 2011 was not anomalous in a way that would change the melt rate pattern we present in this study. We looked at pdds from year-to-year and the summer of 2011 is not an outlier in that regard. But we deleted this and just focus on the summer of 2011 in the discussion now.

L324. Is this potential overestimation from the sampling strategy (at top of cliffs) included in the error bars, as it should?

The assertion in the text as it stands is logically correct and we feel is a better way of arguing than adding error bars. We aren't sure how following this suggestion will actually improve the error estimation beyond what is presented. We could include error bars for each backwasting but how would that improve the legibility of our figures? How much of an over estimate is it? We do not know, so where do we end the error bar?

Rather if we know from other approaches that these are maximum estimates we can use that fact, as we do already in the discussion to present these ice cliff backwasting rates as generously high. With the revisions they are even more generous.

The backwasting errors are now presented, and are very small compared to the extreme uncertainty estimates included in the original manuscript.

L334. "mass loss" should be replaced by "melt rate" here.

All lake discussion is moved to out of the new manuscript.

L344. I did not get the point here.

All lake discussion is moved to out of the new manuscript.

L349. The wording suggests that 11.7 % of the glacier is covered with cliffs. No. This is the % of the debris-covered tongue.

All lake discussion is moved to out of the new manuscript. But we are more careful throughout with the study area definitions.

Table2. For the ice cliff backwasting parameter f, the most likely value is not contained by the min/max interval. A typo? Or a real error? For the ice cliff area the most likely and max values are equal. This is also not really expected neither.

Thank you for the comment but this is not an error. We point the reviewer to parameter 'g' immediately below parameter 'f' which is the y-intercept for the curve-fit. If you also look at the original Figure 4 you will see that these parameters are correct.

Table 3. Ice cliff fractional area, a percentage of what total area?

% of the study area. We clarified this.

L415. I do not understand this note.

Removed.

Figure 2. Are these data from Das et al? Did they use GDEM V2? This would be problematic because it has no defined time stamp. Explain the 1957 and 2015 grey boxes also.

They use a May 4 2004 ASTER DEM. Clarified.

Figure 4. Multiple reference to part A complicate the reading.

Fixed by combining Parts A and B into one.

25 and 50% or 25 and 75?

This is a typo, sorry. Fixed.

Is it "elevation bins"?

**Fixed.**

Panel B. Why the order of values in both axis are reversed. Why not showing the Ostrem way?

Reversed in revisions.

Authors could refer (in the article, not here) to a compilation by Kraaijenbrink et al., 2017 in their Nature study.

It is not clear what the reviewer is specifically wanting us to cite.

Figure 5. Can the authors show the location of this small area of the glacier?

Done.

Figure 6. Impressive maps.

**Thank you.**

Figure 8. Showing percentage for panel c (instead of fractional area) would facilitate correspondence with the text.

We note the difference in the caption.

Figure 9 The sign only make sense if this is referred to as "surface mass balance rate". If the word "melt" is preferred then positive values should be shown.

Fixed.

Figure 10. See comment on Figure 9 for "melt"

Fixed.

Figure 11. Recall the period of dh/dt. In panel b rather than repeating dh/dt authors could show a map with the density / m 2 of ice cliff.

This figure was moved out of the revised mansucript.

Thank you again for your efforts reviewing this manuscript.

**Reply to Reviewer 2 Part B**

Review of 'Debris cover and the thinning of Kennicott Glacier, Alaska, Part B' by Leif Anderson et al., under consideration for The Cryosphere

**Thank you kindly for taking the time to review our manuscripts!**

Part B of the Anderson et al trilogy aims to combine empirical relationships of surface properties and melt rates, based on the field measurements presented in Part A, with remote sensing observations of different surface features (particularly ice cliffs) in order to arrive at distributed estimates of melt rates for the period of observations. The analysis then uses these distributed melt values to address whether or not the identified melt hotspots can explain the thinning patterns evident at Kennicott. This is an important question as the glaciological community is trying to disentangle the influence of surface mass balance and ice dynamics for debris covered glacier evolution, and this is the first time the question has been addressed in Alaska, where surface debris is prevalent.

As such, this represents an important contribution to a current topic of research, and provides an answer to that question for Kennicott Glacier – these hot spots do have an important effect on the surface mass balance, but it is not plausible that they compensate for the overall melt reduction due to surface debris. The study could be more systematic to provide a definitive answer, and I have some criticisms regarding the empirical relationships presented in determining the distributed estimates of melt, as well as the difference in temporal scales between long-term elevation change (52 years) relative to single-year field measurements. However, although I have quite a few comments, no changes along these lines are likely to change the conclusions of the study. Rather, my principal concern is that the separation of this analysis from both Parts A and C reduces the strength and presentation of the entire analysis, while also leading to repetition of text and figures, as well as cross references. There are certainly gaps in the analysis because aspects have been included in Parts A or C rather than here, and some restructuring across the three manuscripts might improve the readability of all three. This is a choice for the authors and Editor to contemplate, but my opinion is that some consolidation would be beneficial.

**Thank you kindly for taking the time to review these manuscripts!**

**Main comments:**

My principal question in reading this manuscript was whether it should be standalone or integrated with Part A (and possibly C, which I have not reviewed). I appreciate that this is a difficult decision, and that Parts A, B, and C combined represent a substantial body of work.

**Thank you for your thoughts. We are submitting one manuscript combining Parts A and B.**

There are certainly some advantages to be considered for maintaining separation between the manuscripts, but my impression is that A and B are both weaker manuscripts separated from one another. The field measurements were clearly collected for the purpose of deriving distributed melt rate estimates, which leaves Part A without a compelling conclusion or discussion.

**Parts A and B are combined into the new manuscript.**

At the same time, Part B requires frequent reference to Part A, or blind faith of the

part of the reader with regards to the methods and results of the field data. As a consequence, there is also quite a bit of repeated material to cover (for example, content from 4 figures in Part A is also displayed in Part B).

**Parts A and B are combined into the new manuscript.**

(for example, content from 4 figures in Part A is also displayed in Part B). My instinct when reviewing Part A was that much of that material could be more meaningful if integrated with Part B, in supplementary information if not in the main text, and I now think that would greatly improve the readability of this manuscript.

**Parts A and B are combined into the new manuscript.**

I like the ice cliff delineation method in its simplicity (although some details need to be clarified, below), but its transferability is not very clear. Often when developing/proposing a new method it is necessary to see how robust the method is, but in this case the method has clearly been developed specifically to map cliffs in this particular scene, in order to apply the empirical melt relationship. As such, it is a relatively small part of the story in deriving the distributed melt estimates.

Thank you for highlighting this. We present the new ice cliff detection method as a proof of concept for a rather tough case. Kennicott Glacier has more dense ice cliffs than any other debris-covered glacier that we know of. Slope threshold approaches do not appear to be as effective on this glacier. We could include a more thorough comparison with other methods we feel that should be the focus of another manuscript.

But we disagree that the ice cliff delineation method is a small part of the study. We feel it is a clear achievement to delineate ice cliffs so well on such a complex example!

At the same time, the maps of supraglacial ponds are not integrated very well into the story, while the exclusion of supraglacial streams (also mapped from satellite imagery) is a bit

**Thank you for pointing this out. We move lakes and stream methods out of the new submission.**

The study aims to address the role of melt hotspots in explaining thinning rates. For this to be a definitive analysis in this regard, I feel like this needs to be done in a more systematic manner, whereas the present analysis seems to focus exclusively on ice cliffs.

**We will tone down the language about melt hotspots and focus on ice cliffs.**

The distribution of supraglacial streams seems to be a major gap in the analysis of this part of the study, especially as the properties of supraglacial streams are assessed in Part C. Streams are mentioned throughout the background as hot spots and possible factors contributing to melt, but are then completely neglected in the methods and discussion.

**We mention streams once or twice in the new manuscript.**

It is important to note that there are very few studies that address the effect of streams on debriscover glacier mass balance and to date none have really quantified an effect. We simply will focus on ice cliffs in the new submission. I understand that the role of surface streams cannot be assessed quantitiavely in this manuscript, but neither can the role of ponds. Similarly, although internal ablation is usually regarded as negligible in this type of specific mass balance assessment, there have been suggestions that this is a non-negligible term for extensively debris-covered glaciers. It is exceedingly unlikely that this mechanism could lead to the debris-cover anomaly, but for completeness I think it should be considered numerically along with the cliffs and ponds.

We will not address this issue as the reviewer states, "It is exceedingly unlikely" to matter despite the suggestion in the literature. Rather we will just focus on ice cliffs in the new submission.

The relationships between elevation and debris thickness, and between elevation and ice cliff backwasting, are very weak. In neither case does elevation appear to be a primary control of the property.

They might be weak but our generous uncertainty analysis makes this point moot.

We are a bit confused about the elevation versus debris thickness statement because lower in this review you state: "In Part A it is clear that while elevation is a principal driver of debris thickness variability, there is considerable heterogeneity within any elevation bin." As we suggested in Part A it would be a nice idea to extrapolate down flowlines, but we felt like this was a methological development to focus on in later studies.

Despite the scatter we agree on, Figure 5 Part A (see below) shows that debris thickness does increase down glacier. We can argue about whether or not that is linear or not but the box-plots show the increase. Here is a reason why Part A standing alone is good for presenting the body of work. We are able to show the box plots that really reveal the thickening of debris down glacier.

Figure 5. Pattern of debris thickness with elevation. a) In situ debris thickness measurements. b) Debris thickness boxplots in 50 meter elevation bins. Outliers are represented as +'s.

How different would your results be if you interpolated spatially (within flowlines or lithological units for example)? Did you consider alternative methods to provide a

distributed debris thickness estimate?

We re-did the distributed debris thickness and melt estimates by extrapolating down medial moraines.

For example, one could consider the radiance measured by satellite thermal infrared imagery as related to debris thickness, and use your measurements to constrain this relationship locally to upscale in space much more meaningfully.

We did consider this approach but because of the number of ice cliffs in the study area we felt that this could be another manuscript in of itself, requiring significant justification and method development. We would need to remove the bare ice effects from each pixel before estimating debris thickness.

For ice cliffs, justification of a fit to elevation needs to be more explicit in this Part of the study.

**We now use uniform backwasting rates. The primary results do not change.**

The conversion of backwasting rates into ice cliff melt misses a slope correction for the cliff area, which results in an underestimation of melt (see comment on L190).

**This was actually already included but not stated in the manuscript.**

This raises a difficult question that has not been carefully considered yet for debris-covered glaciers, which is that the real surface area can be 10-20% higher than the planimetric area. For this study comparing geodetic thinning observations with estimated melt, that is an important aspect to consider, as melt occurs relative to the real surface area. This effect is especially pronounced for ice cliffs, but is also crucial for the 'background' melt rate if the glacier surface is highly variable.

**We agree and it was already included in the original contribution.**

It is too bad that more recent geodetic difference data were not included in the analysis. At present it is not clear how the long-term thinning rate relates directly to the 2011 observations. Would it be possible to use the ArcticDEM datasets to derive a recent-period thinning pattern? This would be more meaningful for a comparison to the contemporary distribution of surface features. The long-term perspective is still useful for understanding dynamic changes, but comparing 50-year lowering rates to one-year melt patterns does not provide a definitive answer, especially for a clearly changing system.

We brought in the Das et al., 2014 datasets from the supplemental into the main text. More recent thinning data was going to be in the new Part 2 but we feel it is better to submit that manuscript elsewhere.

Some rewriting is needed for readability and presentation standards in The Cryosphere. Although the ideas are well developed, some sections of the paper read as bullet points and/or the word choice has not been considered carefully, leading to some of my comments below.

We agree that there is some ambiguity in the writing that needs to be corrected. This review helped us understand where we can tighten the wording up considerably. Thank you kindly!

Detailed comments:

L23. Presumably this 19% of melt is for the debris-covered area?

Yes we will make sure the study area is defined correctly.

L25. Just a comment, for you to adopt or disregard: The literature has tended to use 'ponds' for these features as they are much smaller than supraglacial lakes on, e.g., the Greenland or Antarctic Ice Sheets.

We will switch to ponds. Helpful suggestion and we should be consistent with the literature.

L25. It would be nice to have the %areal coverage numbers here in the abstract, not just 'doubled'.

L27-27. This wording isn't very clear. By average melt rates you seem to mean the average for all surfaces, but as worded this seems to refer only to sub-debris melt. In the latter half of the sentence, do you mean that the overall melt relationship still follows an Ostrem-type relationship, even after accounting for cliffs and ponds?

We will clarify this as it is really important, making sure we emphasize which components we are including in the distributed melt.

Thank you for pointing this out. We will clarify this

Suggested change: Despite abundant ice cliffs and expanding surface lakes, average melt rates are suppressed by debris, with the primary control on elevation-band-average melt rates appearing to be debris thickness.

L34. These are broken sentences. The first half needs a reference even if it is now well understood.

We will join with a comma

L95. Has this happened progressively in the past 82 years, or primarily over some later period?

This is outlined in the citation, Rickman and Rosenkrans (1997).

L98. Why is 'partially' here?

Good catch.

We will change this to: The presence of ice cliffs, surface lakes, and variations in debris thickness on debris-covered glaciers makes distributed estimates of mass balance difficult.

L108. Which spectral bands of the image do you use?

We will add:

On 109, before "The 2009 WV image..." add "We use the panchromatic band, which integrates radiance across the visible spectrum and provides the highest spatial resolution".

L113. This is true in certain conditions, depending both on debris lithology and meteorology. Such conditions may be prevalent for Alaskan supraglacial debris and melt seasons, but it is important to think whether such a method is transferable.

This is a good point. We looked into the universality of this observation that ice cliffs are generally darker than the surrounding debris cover using WorldView imagery in the Digital Globe's EVWHS viewer. We attach screenshots of Miage, Koxhar, and Lirung glaciers below. These are debris covered glaciers from the Alps and Tibet, which are found at different latitudes and lithologies than Kennicott Glacier. We find that, consistent with our Kennicott Glacier observations, ice cliffs on these other glaciers are generally darker than the surrounding debris cover. We can find examples where this is not true, for example along parts of the southwestern margin of Miage Glacier in the snapshot below. These regions may go undetected in the method we present here. We address this limitation in the manuscript (L281 in the original submission). To address this comment, we add at the end of this paragraph "This observation of darker ice cliffs is generally true for Kennicott and several other debris covered glaciers we examined, but this relationship should be verified before application to a different glacier of interest. There are situations (e.g., variable debris-covered and debris-free ice) where this method could detect darker regions that are not related to ice cliffs. Output should be examined to ensure that such conditions do not contaminate results". This language also partly addresses your L126-127 comment as well.

Miage Glacier (45.80 N, 6.85 E)

---

## Referee Report (RR1)

**Review of 'Debris cover and the thinning of Kennicott Glacier, Alaska: in situ measurements, automated ice cliff delineation and distributed melt estimates' by Anderson et al, submitted to The Cryosphere**

The revised manuscript by Anderson et al has combined parts A and B to focus on distributed melt estimates for most of the debris-covered area of Kennicott Glacier, Alaska. The scope is much clearer and the study is greatly improved in this formulation, which I appreciate has been a major undertaking for the authors. The presentation and analysis of observations from debris covered glaciers in Alaska are a welcome contribution to The Cryosphere, and this study uses empirical models to complement the remote sensing observations and inferences of Brun et al (2018) regarding the importance of ice cliffs and sub-debris melt, and the resulting patterns of glacier-wide melt rates.

At this point I have few substantive comments, although I think the study would benefit aesthetically from additional careful editing and textual revision. In particular, some content seems misplaced between Methods/Results/Discussion, and the storyline is not entirely clear for the discussion, which comes up just short of stating explicitly that since melt rates are not the key factor driving the zone of maximum thinning, reduced ice fluxes must play an important role.

My comments below are extensive, but primarily relate to the presentation of the study. There are a few more substantive comments, but none that require additional analysis.

Summary comments:

1. A need for careful final editing in terms of grammar and accuracy. At present the writing style is not particularly satisfying to read: many sentences start with 'it,' 'but,' 'and,' 'because,' etc that should be linked to the preceding thought, and the editing/proofreading has not been thorough.
2. When uncertainty values are given, it is not clear how they were derived or what they correspond to. In terms of representativeness, the authors seem to make some key unstated assumptions: 1) that the peak ablation season is a good proxy for annual-average ablation rates (possibly not true for ice cliffs), 2) small-scale debris thickness variability is not important to assess glacier-scale melt rates despite the nonlinear relationship between debris thickness and melt). Although I disagree with both of these assumptions, I do not think they affect the conclusions of the study. I do, however, think those assumptions should be made clear; this study makes good use of empirical methods to ask an important theoretical question, but the reader needs to be reminded of the fundamental assumptions of the framework of analysis.
3. Some content related to the continuity equation in the introduction and methods, but then this isn't really discussed in the results and discussion at the end. I guess these parts of the analysis have been retained for later publication, but it feels like an unfinished thought here; perhaps not entirely necessary (yet) or a bit of discussion would close the gap to simply indicate clearly that based on this analysis it should be tested whether a reduced ice flux can explain the ZMT.

Comments:

L18. 'melt reducing' is a compound modified and should normally be hyphenated

L19. This sentence is a bit complex due to the frequent commas. There should not be a comma after 'thinning'. The 'but' is counterintuitive here and is better replaced with 'and' as Kennicott is an example of the dynamics represented above. However, the 'under insulating debris cover' is only demonstrated by your manuscript later on (it is not yet background knowledge that the debris is thick), so perhaps best to remove 'insulating'.

L22. Although an excellent contribution, this is a semi-automated method as it requires training data.

L23. Why the comma after 'relationships'?

L25. 'which' should be 'and' for the sentence to make sense grammatically: Ice cliffs cover 11.7% of the debris-covered tongue [, … ,] and contribute…'

L26. 'with a' should be 'which has'. As written, the literal meaning is that the ice cliffs have a mean debris thickness of 13.7cm, but I suppose that this should refer to the debris-covered area.

L29-30. This is the first time that the decline in ice discharge is mentioned in the abstract. Introducing at the end of current line 24 with a succinct methodological description would clarify to readers where this comes from. Otherwise, the abstract should also have a line along the lines of 'We find a decline in ice discharge from up-glacier…'

L37. The debris is not itself expanding, but the debris-covered areas are.

L38-39. I suggest joining these sentences as they are directly linked theses. It's rarely a good idea to start a sentence with 'But'. The first clause, however, is a hypothesis that should have a reference. In addition, the only study I know of that has assessed whether debris is actually *thickening* rather than simply expanding in coverage is (Gibson et al., 2017).

L41. (F. Brun et al., 2019) is a better reference for this phenomena, although the 2018 study is also appropriate.

L43. Again, I suggest linking this sentence to the one before: ', and has been documented…' Also, Asia has already been mentioned just two sentences before, so this could be 'and has also been documented in Europe'.

L46. This phenomena being global or not is a bit out of scope for this paper. It's an interesting supposition but the important aspect for the background of your study is that it also occurs in Alaska.

L47. No need for 'from' in 'from within'

L48. 'Compelling' is out of place or a word is missing: 'This is a compelling ___ because …'

L54-55. I suppose that this question is the overarching drive of this analysis, and should be highlighted as such rather than as a rhetorical question in the text.

Eq 1. The equation requires units for each of the quantities to be the same (m/a or m w.e./a) which should be noted somewhere, since b* usually differs in units from dH/dt and the flux divergence. Also, the flux divergence terms are not strictly correct, they should be $dQ\_x/dx$ and $dQ\_y/dy$ and are partial derivatives.

L61-75. It would be good to include a reference for the continuity equation and its simplified version here; this has been done before and deserves a reference. Cuffey and Paterson is the obvious choice

because it contains all these concepts, but the continuity equation has been applied to mountain glaciers extensively in Peru and the Alps since the 1990's, and since the 1970's at least for the ice sheets. I'm not suggesting a comprehensive list, but a reference or two would be value-added.

L72. Benn et al (2017) is one of the few studies that have tried to estimate $b^*\_e$, and is highly relevant.

L76. This is a nice way to present the various hypotheses.

L80. I believe you intended to italicize the 'M' in 'Melt' as well.

L84. The comma should be removed from this statement.

L85. This is not necessarily upglacier of the debris, but upglacier of the *thick* debris. Typo: 'thinning *and* reduced ice flow'

L87. (Fanny Brun et al., 2018) summarized this concept very well in their Figure 11.

L89. Typo: 'revealing'

L90. It's definitely marginally acceptable to reference your own thesis for a paper on the same topic.

L92. 'from across Kennicott Glacier' is already clear from this sentence

L93-95. The mention of 2011 appears from nowhere since it hasn't been introduced that this is the period you have data for. Perhaps it is tidier to leave the questions unconstrained temporally, but then in the next paragraph to introduce the summer of 2011 as the period of observations. You then still need to be careful to indicate that the same patterns are expected to be apparent in the summer of 2011 as for long term/annual mass balance and thinning, for example based on Das et al (2014) and possibly the ITS_LIVE velocities?

L98-99. I think it's fine to relegate the analysis of streams and ponds to another paper, but if these are also potential contributors to surface mass balance, doesn't this cut your thesis short? I.e. you are no longer able to make a statement for all melt hotspots combined, but only for cliffs. Now, we certainly know that there are few ponds on Kennicott, and their location is different to the TMZ; why not simply pretend they melt at the same rate as cliffs (somewhat as in (Kraaijenbrink, Bierkens, Lutz, & Immerzeel, 2017)) so that your ablation budget is complete? Streams are even more problematic, since they are prevalent and unconstrained. I make this point not to criticize your work; I am convinced that the hotspots are only part of the explanation and that ice dynamics are vital to explain the debris cover anomaly. I just this your huge amount of work here will be much stronger by representing these somehow (even as a hypothesis test).

L105. I believe this should be WorldView-1.

L117. Is this 20% figure from your own digitization (if so, of what imagery and when?), or from (Scherler, Wulf, & Gorelick, 2018)? Just best to indicate the source (or 'approximately'), as these numbers often get recycled.

L130. There is an extra space before 'duration'. Understanding that the scope of this study is the bulk of the ablation season for this glacier, have you considered the representativeness of this period to the full-year ablation budget? This is not likely to change your outcomes, but it is worth thinking whether ablation hotspots such as cliffs and ponds 'turn off' their melt contributions at the same time as the general debris-covered tongue.  This probably depends based on the glacier-specific site and characteristics, but cliffs and ponds could have positive surface energy balance

during periods of the year when conduction through debris is already negligible. The assumption here (with regards to the research question) is that the peak ablation season corresponds well to the average annual ablation across the glacier. I don't think this assumption is particularly bad, but it should be made explicit.

L136-142. I completely agree with all of this content, but it belongs in the Introduction.

L147. Perhaps combine this sentence with L148-149 'by digging through…'?

L152. It is not clear what these values correspond to – the uncertainty and variability of the measurements themselves, or (as it appears in Fig 5) of the altitudinal curve-fit. For each of these, a bit of additional information is needed.

How did you get an thickness uncertainty of 0.3cm? This is incredibly precise considering the challenge of the measurement – excavation to the ice surface without disturbing the ice surface, finding the reference height, ensuring a vertical measurement, etc. For a 30cm pit, this implies a measurement within 6 degrees of vertical, ignoring all other uncertainties, or for 60cm, 4 degrees.

What does the standard deviation correspond to – the variation between measurements at the same elevation?

I guess that the 'maximum error' is the maximum standard error of the elevation's mean debris thickness?

L159. Again, the 'how' of the uncertainty is as important as the value. In this case I would suspect that this uncertainty represents the uncertainty in the stake height measurements, spread over time? Similarly for L176.

L165. Typo in 'nonetheless'

L166-171. This is nice content and a very tidy synthesis of key points from past work on cliffs, but shouldn't this be in the Introduction? No methods appear until L171-2

L196-197. The content of the stereoimagery is out of place here, and interrupts the description of the ice cliff mapping method, for which it seems irrelevant. Perhaps it could be part of 2.3, since you use the elevation to prescribe the melt rates across the glacier?

L222. As written, I'm not sure which problems are for 'ABT and SED' or 'SED only'. This becomes clearer later in the paragraph, so maybe simplify and remove the indication of which method suffers from which problems here?

L231-2. Makes sense to reference (Steiner, Buri, Miles, Ragettli, & Pellicciotti, 2019) here. This difference is 3% of total cliff area or 3% area mismatch (non-overlapping areas)? Please make it clear in the text.

L259-270. I am impressed by the effort to represent the debris thickness variability based on flow units, which is a key aspect of spatial heterogeneity in deriving glacier-wide melt estimates based on (necessarily) spatially-biased measurements.

L274. It's nice to see h* explained in very clear terms (the half-melt thickness) and related to physical meaning. Please do add units to the definitions of each variable.

L295. Indicate the unit of degrees in the equation (for 90).

L312. Verb tense issue: '… the curve fits … *are* calculated …'

L317. 'ice cliffs' should be singular (ice cliff slopes) or possessive (ice cliffs' slopes)

L318. Doubled %%

L340. Please do not start this sentence with 'And'.

L346. A suitable reference for this is (Mertes, Thompson, Booth, Gulley, & Benn, 2016).

L383. Nice simple inference worth testing further. Most likely this relates to the difficulty of mobilizing debris, see e.g. (Moore, 2017).

L392. It's not immediately clear what the values in parentheses mean. Are these the lower/higher estimates? Just make it clear in the first instance.

L421. Is this in cm? This content is a bit unrelated; the consistent decline in sub-debris melt is not unexpected for many other reasons as well, which h* just indicates in aggregate.

L428. Awkward comma placement. Best to get rid of 'Becuase' here, and connect to the next sentence (which begins with 'But'). Again, see (Moore, 2017) for a review of these processes.

L423-446. This is interesting content but largely conjecture and lacking a synthesis – so there are many possible reasons for heterogeneity in cliff backwasting rates as you measured, but what does it mean for science? What is the point? That we need large N? Or high-quality measurements and models? Or…?

L457. This sentence undermines itself: 'Our method works if you correct it for bias.' I think the point here is that your method presents the spatial variability of ice cliff areas very well, even if it underestimates the total cliff area somewhat; this spatial variability is key for you to derive the subdebris melt estimates! It would be nice to see how consistent those biases are between the distinct evaluation patches, possibly as a panel in the SuppMat. A lot of content, time and effort has gone into the cliff mapping method and it would be nice to see the high quality results in more detail there.

L489. Great synthesis. My only suggestion would be to also clearly acknowledge that ice cliffs greatly enhance the overall melt rates for the debris-covered area; as presently written, the implication is that one only has to account for the increasing debris cover to get a reasonable melt estimate; this would underestimate melt by 26% in your results. Clearly the debris dominates the glacier-wide melt rate and wrt elevation, but also it is clear that you need to take cliffs into account somehow.

L515. Convincing hypothesis tests above. For completeness, you may as well perform a back-of-envelope for cliffs and streams to emphasize that, even poorly-constrained, these features cannot source sufficient energy for melt to be the only driver of the ZMT.

L515-517. Agreeing that melt is not maximized in the ZMT, isn't it also clear that cliff and debris melt is maximized (although still less than for clean ice) leading into the ZMT? I.e. aren't cliffs and high subdebris melt rates important drivers of the reduced ice flux into the ZMT?

L519. 'debris-covered glacier termini' appears to have been accidentally used twice.

L522. (Bisset et al., 2020) examined selected glaciers across HMA, not only in the Everest region

L524. The 42% figure comes from nowhere – do you mean Ngozumpa via Thompson et al (2016)? Anyways I think the suggestion is a good simplified debris thickness representation – if you can figure out how to estimate the apportioning of melt to cliffs!

L530. To my knowledge it is the first time that melt has been quantified for a debris-covered tongue in Alaska, and one of the few times it has been done convincingly, with in situ measurements, for a debris-covered area globally. Nice work.

L531-569. Please rework this long list (12 items!) of bullet points into cohesive paragraphs.

L572. Please just archive the data in a repository. You can still force people to request them or let you know what they intend to use the data for.

Table 1. The values for ice cliff backwasting in the 'min' and 'max' cases do not seem to correspond to Table 2 or Figure 7 (these would be 4.6 and 9.6). Which values did you use?

Table 4. Note that (Thompson, Benn, Mertes, & Luckman, 2016) lumped all cliff-associated influence by necessity, whether due to cliffs/ponds/streams etc.

Figure 1. On panel (c) the legend need some work to show the variation (dH/dt) and units clearly. Please also consider using a proper color ramp – I know it is a pain in QGIS. I don't think there is a need for the statistics from the Das et al (2014) study in the caption, but if you want to include them, it's not clear what you mean by the mean error and 1 std – is the 'mean' the glacier-wide error of the mean, and 1 std (can be sigma) the standard deviation across the glacier? This is easily misunderstood.

Figure 3. Very nice synthesis.

Figure 4. As for figure 1, the legend in panel (a) needs to be tidied.

Figure 5. Correction to caption text: 'The red bars *are* the median…'. For completeness of the manuscript itself, and also as a nice comparison to this relationship, I would prefer to see the relationships of debris thickness with elevation for the other moraines here.

Figure 6 caption. '… is smaller *than* the marker …'. The RMSE units are presumably cm w.e. d^-1?

Figure 7. I suggest to indicate 'ZMT' next to the arrow for clarity. In (b) the axis labels are not terribly clear: 'aspect' is not labelled anywhere (nor the units) and the ice cliff backwasting axis values are a bit confusing due to their location.

Figure 8. In some respects the results without morphological opening look even better, because the opening has also removed thin slivers of true ice cliff area. Did you try a connected-pixels morphological clean? This would eliminate (for example) 'cliffs' with less than 10 pixels, and would generally leave the thin cliffs unmodified. Perhaps you are happy with the current performance, but it's just a suggestion.

Figure 9. In the caption, there is mention of 'thin red lines' but I don't see these anywhere. Perhaps use a different colour for these?

Figure 10. Please indicate the elevation bin size for the hypsometry.

Figure 11. Why not delineate the ZMT here using elevation contours (thus corresponding directly to the rest of the paper)? Otherwise I find myself wanting to flip back several pages to see where the contours are, etc.

Figure 12. Small typo: '84.1% of estimate*s* are within the grey shaded band.'

Bisset, R. R., Dehecq, A., Goldberg, D. N., Huss, M., Bingham, R. G., & Gourmelen, N. (2020). Reversed Surface - Mass - Balance Gradients on Himalayan Debris - Covered Glaciers Inferred from Remote Sensing. *Remote Sensing*, *12*(1563). https://doi.org/10.3390/rs12101563

Brun, F., Wagnon, P., Berthier, E., Jomelli, V., Maharjan, S. B., Shrestha, F., & Kraaijenbrink, P. D. A. (2019). Heterogeneous Influence of Glacier Morphology on the Mass Balance Variability in High Mountain Asia. *Journal of Geophysical Research: Earth Surface*, 1331–1345. https://doi.org/10.1029/2018JF004838

Brun, Fanny, Wagnon, P., Berthier, E., Shea, J. M., Immerzeel, W. W., Kraaijenbrink, P. D. A. A., … Arnaud, Y. (2018). Ice cliff contribution to the tongue-wide ablation of Changri Nup Glacier, Nepal, central Himalaya. *The Cryosphere*, *12*(11), 3439–3457. https://doi.org/10.5194/tc-12-3439-2018

Gibson, M. J., Glasser, N. F., Quincey, D. J., Mayer, C., Rowan, A. V., & Irvine-Fynn, T. D. L. (2017). Temporal variations in supraglacial debris distribution on Baltoro Glacier, Karakoram between 2001 and 2012. *Geomorphology*, *295*, 572–585. https://doi.org/10.1016/j.geomorph.2017.08.012

Kraaijenbrink, P. D. A., Bierkens, M. F. P., Lutz, A. F., & Immerzeel, W. W. (2017). Impact of a global temperature rise of 1.5 degrees Celsius on Asia's glaciers. *Nature*, *549*(7671), 257–260. https://doi.org/10.1038/nature23878

Mertes, J. R., Thompson, S. S., Booth, A. D., Gulley, J. D., & Benn, D. I. (2016). A conceptual model of supra-glacial lake formation on debris-covered glaciers based on GPR facies analysis. *Earth Surface Processes and Landforms*. https://doi.org/10.1002/esp.4068

Moore, P. L. (2017). Stability of supraglacial debris. *Earth Surface Processes and Landforms*. https://doi.org/10.1002/esp.4244

Scherler, D., Wulf, H., & Gorelick, N. (2018). Global Assessment of Supraglacial Debris-Cover Extents. *Geophysical Research Letters*, *45*(11), 11,798-11,805. https://doi.org/10.1029/2018GL080158

Steiner, J. F., Buri, P., Miles, E. S., Ragettli, S., & Pellicciotti, F. (2019). Supraglacial ice cliffs and ponds on debris-covered glaciers: Spatio-temporal distribution and characteristics. *Journal of Glaciology*, *65*, 617–632. https://doi.org/10.1017/jog.2019.40

Thompson, S., Benn, D. I., Mertes, J., & Luckman, A. (2016). Stagnation and mass loss on a Himalayan debris-covered glacier: processes, patterns and rates. *Journal of Glaciology*, 1–19. https://doi.org/10.1017/jog.2016.37

---

## Editor Decision (ED1)

**Editor decision, 13 Nov 2020**

Dear Leif and co-authors,

Thank you very much for the revised paper, which I am happy to accept for publication after technical corrections. I am happy to see that you considered almost all issues raised – and that you addressed some of the more major ones raised by reviewer 2. I agree with your choice of how to treat the assumptions you made and the sensitivity/uncertainty analysis (mostly moved to the SI).

I have few technical details for you to address, as I feel the English of the text could still be more polished. I have suggested a restructured version of the abstract below, which maintains all your content of course. I feel the order of the paragraphs was not entirely logical, and some of the wording could be improved. Feel free to use that or a modified, more fluent and tidier version of yours. I would encourage the authors to go through their text once more, and try to polish it some more and correct style/grammar, based on some of my comments below (in few instances the text contains still grammar errors). This is something that the co-authors could help with.

Finally, I hope the authors are satisfied with this final paper version, which I feel is substantially improved. I also hope that they appreciate the constructive and very thorough reviews of their paper, which remained all constructive all the time. There is no need for confrontational wording (e.g. lines 527-528: "From our view that is a presumptuous statement about the method we develop here".).

I really look forward to see this paper published.
All the best,

Francesca

REVISED SUGGESTED ABSTRACT:

Many glaciers are thinning rapidly beneath melt-reducing debris cover, including the Kennicott Glacier in Alaska. The zone of maximum thinning at Kennicott Glacier is located under debris. Scattered within the debris cover, melt hotspots, such as ice cliffs, locally increase melt rates. We explore the roles of debris and ice cliffs in controlling rapid thinning under thick debris at Kennicott Glacier.

We collected abundant in situ measurements of debris thickness, sub-debris melt, and ice cliff backwasting allowing for extrapolation across the debris-covered tongue (the study area and the lower 24.2 km 2 of the 387 km 2 glacier). A newly developed automatic method, the Adaptive Binary Threshold, uses for the first time only optical satellite imagery to delineate ice cliffs. We show that the method accurately estimates ice cliff coverage even where ice cliffs are small and debris color varies.

Kennicott Glacier exhibits the highest fractional area of ice cliffs (11.7 %) documented to date. Ice cliffs contribute 26% of total melt across the glacier tongue. Although the relative importance of ice cliffs to area-average melt is significant, the absolute area-averaged melt is dominated by debris. At Kennicott Glacier, glacier-wide melt rates are not maximized in the zone of maximum thinning, and we show that this rapid thinning is due to a decline in ice discharge through time. There is more debris-covered ice in Alaska than any other region on

Earth. Through our efforts, Kennicott Glacier is now the first in glacier in Alaska and the largest globally where melt across its debris-covered tongue has been rigorously quantified.

COMMENTS

COMMENTS RELATED TO THE REBUTTAL TEXT

_On line 265 of the document (in the authors' reply to reviewer 1): you show as original and new paragraph exactly the same text! Please clarify here if and how you made changes to that text.

_Text on line 590: there is a "in" missing I assume?

Original text:

"Expanding and thickening debris cover should reduce glacier thinning relative to glaciers without debris (Banerjee, 2017; Gibson et al., 2017), but the melt-suppressing effect of debris is not always apparent the observed thinning patterns of glaciers even when debris is thick and debris coverage is extensive (e.g., Kaab et al., 2012; Gardelle et al., 2013)".

It should be:

Expanding and thickening debris cover should reduce glacier thinning relative to glaciers without debris (Banerjee, 2017; Gibson et al., 2017), but the melt-suppressing effect of debris is not always apparent IN? the observed thinning patterns of glaciers even when debris is thick and debris coverage is extensive (e.g., Kaab et al., 2012; Gardelle et al., 2013)".

_Lines 765 and following: In response to a comment by Reviewer 2: Krajeenbrink et al (2017) only consider ponds, not ice cliffs, in their melt estimates for HMA. I would also say that the evidence available until now suggests that the two do not melt at the same rate. So I suggest you rephrase the sentence you have added:

*"While we do not explicitly document the melt rate of ponds and streams (i.e., melt hotspots) we follow Kraaijenbrink et al. (2017)'s approach and assume they melt at the same rate as ice cliffs. Using this logic, in order for ice cliffs (melt hotspots) in the ZMT to compensate for the insulating effects of debris, ice cliff (melt hotspot) area would need to increase from 11.7% to 90% of the glacier surface. Ice cliffs, ponds, and streams assuredly do not occupy 90% of the ZMT. This again suggests that ice cliff and other melt hotspots do not control the location of the ZMT."*

I would recommend that in this paragraph you simply state that you do not address ponds and streams here, that you will do it in a follow up publication, but that, given current estimates of their contributions to ablation, their inclusion will not change your main conclusions.

_Line 802: provide the source of that 20% value (your own digitisation).

_Line 876 and following: on uncertainty of ablation measurements: how do you combine the uncertainty in the marking and the uncertainty due to the tilt of the stake? Can you please explain?

_Line 946: the reviewer suggested added reference: "*L231-2. Makes sense to reference (Steiner, Buri, Miles, Ragettli, & Pellicciotti, 2019) here*". And I would agree with that, even if this is one of our studies.

_Line 970: I agree with the reviewer that you should provide units.

COMMENTS RELATED TO THE REVISED TEXT

The following comments and lines refer to lines of the revised paper (in track-changes):

_Line 1340:

«Greater surface elevation changes»: say if these are positive of negatine, I assume you refer to thinning patterns here?

_Line 1434: I would remove this sentence

"leaving a detailed examination of other melt hotspots for another contribution.", or rephrase it. The sentence has it is grants an explanation of why they are not included here, and does not add much to the paper. There will most likely be a second contribution, and readers will be aware of it.

_Line 1442

I would add, after (DEMs), "but showed that the method is sensitive to the DEMs resolution and its predictive skills diminish for coarser resolution".

_Line 1448

Consider removing: "thereby addressing the questions outlined above»

_Line 1496: I would remove "better"

_Line 1566: I would rephrase as: We DEVELOP an automated algorithm to delineate ice cliffs from optical satellite imagery.

_Line 1567 and previous occurrences: WorldView (WV) satellite- check where you use WorldWiew as a word for the first time, provide the acronym there and be consistent in its use. You have used both acronym and extended word in the text before this occurrence.

---

## Author Response (AR2)

**Editor comments**

I have now received two very thorough and useful reviews, both of which indicate minor revisions but also list a number of comments that should all be addressed before the paper can be accepted for publication. Both reviewers point to the need to improve the presentation of your article, and the Discussion and Conclusions sections especially.

We worked extensively to make these resolve these important issues and the manuscript is much improved. All changes followed the advice of the reviewers. Of note is a new section requested by Reviewer 2 at the end of the discussion tying back into the continuity equation for ice. The conclusions are now all in paragraph form and follow the highlights suggested by the reviewers. Following reviewer 2's advice we made sure all introductory text is in the intro, all methods text in the methods, etc. The discussion was streamlined and all repeated text was removed. We also improved the legibility of a few figures.

The Supplementary Material also needs improvement, from formatting to style and avoiding repeating figures that are in the main paper.

We removed repeated figures and referenced them now in the main text so it is easy to interface between the main text and the supplementary material. We also removed several figures to reduce the size of the supplementary materials.

I would also encourage you to carefully address reviewer 2 main point 2 on the underlying assumptions of your methods, and make them clear to the readers. I notice that few of the comments in the previous round of reviews have not been addressed and would encourage you to do it now.

We discuss each of these assumptions mentioned by Reviewer 2 in the text in section 4.3. We appreciate the opportunity to clarify our methods.

Thank you for your efforts!

**Report #1**

Thank you for taking the time to review this manuscript!

Suggestions for revision or reasons for rejection (will be published if the paper is accepted for final publication)

Review the revised version of Anderson et al., TC, September 2020

This is a revised version of a paper that was initially made of 3 parts. Part A and B have been merged into this single manuscript and part C will be re-submitted elsewhere. I do not know how part C will evolve but I applaud the merging of part A and B in a more complete article. I feel that most of my earlier comments were correctly addressed.

At this stage my only major comment is that the paper is lengthy (especially the supplement) so anything that can contribute to make it shorter and easier for the readers will be welcome.

We appreciate the reviewer taking the time to comment on this manuscript. We did everything we could to shorten the manuscript.

Following the suggestions of both reviewers we re-wrote the discussion section on 'in situ measurements.' We removed more extraneous discussion points related to backwasting rate with elevation. We removed the repeated mention of the role of measuring ice cliff backwasting at the top of the cliff from the discussion. We have removed all repeated sentences suggested by reviewer 1. We discuss the results in a more systematic fashion and ensure that there are topic sentences for each paragraph. We also removed the sentences from the conclusion that are related to the in situ measurements. We removed repeated text and figures in the supplemental.

General comments.

Maybe this is the result of the merging of two previous manuscripts (not a funny thing to do, I must admit) but I found the article rather long in particular the discussion. See example of repeated statements in my technical comments. Authors should aim at streamlining the article, especially the discussion.

And the conclusion also: A 12 bullet point conclusions is not a nice way to end up the paper. Cannot the authors find 4 or 5 main findings that would be the take home message for the readers, the reasons why they would then cite this study.

We converted the bullet point conclusion into paragraphs as requested and removed extraneous points.

The same applies to the supplement. Make this supplement more concise, better organized so that the reader can navigate through.

We agree with the reviewer and removed the supplementary photos to reduce the content. We now cite individual supplemental figure numbers in the main text to help the reader navigate the additional material. Each supplemental section has a clear heading to indicate what follows that also follows the order in the main text. Each supplemental figure plays a role in supporting the main text now. Much of the original Part A submission is now in the supplementary material. It is necessarily in the supplemental because it was suggested by Reviewer 2 in the first round of reviews.

We changed the section and figure labels following the TC guidelines as well.

Technical comments.

L23 "across our study area, the lower 8 km of the glacier". So that the "study area "is unambiguously defined".
L25. is it synonymous of the "study area" as defined above? Not clear again.

To resolve these two issues we add a definition of the study area to the abstract.

95  Original:

"We then use empirical relationships, to estimate melt area-averaged melt across the lower 8 kilometers of the glacier.

100  Ice cliffs cover 11.7% of the debris-covered tongue, the most of any glacier studied to date, which contribute 26% of melt in study area with a mean debris thickness of only 13.7 cm."

Changed text:

105

"We then use empirical relationships to estimate area-averaged melt across the debris-covered tongue (the study area), which constitutes the lower 8-km of the 42-km long glacier.

Ice cliffs cover 11.7% of the debris-covered tongue, the highest percentage from any glacier studied to date. Within the study area, debris cover is relatively thin with a mean thickness of 13.7 cm and the

110  abundant ice cliffs contribute 26% of total melt. "

L34. The recently published (maybe after the authors submitted?) debris inventory by Herreid & Pellicciotti should be cited I think:
Herreid, S. and Pellicciotti, F.: The state of rock debris covering Earth's glaciers, Nature Geoscience,

115  doi:10.1038/s41561-020-0615-0, 2020.

Thank you for the suggestion but we don't think this is absolutely necessary at this point. We have been asked to add a number of citations already to the text.

120  L48. "compelling" what?

The other other reviewer did not take issue with this word choice so we choose to leave the text as is.

L56 "Glacier"

125  Corrected 'glacier' to 'Glacier'

L85. "and"

130  Corrected 'an' to 'and.'

L93. I could count only two questions.

Changed 'three' to 'two'

135  L147 "the sites where we"

Corrected 'that' to 'where'

The text now:

"Debris measurement locations coincide with the sites where we also measured ice cliff backwasting and sub-debris melt (Fig. 4; Supplemental material)."

L153. delete "a"

Removed the 'a'

L159. "stakes"

changed 'states' to 'stakes'

Figure 6. Is it wise to have the bare ice point of the same color as the curve fitted to the sub-debris measurements? It confused me.

L165. Nonetheless

Corrected. To 'Nonetheless'

L318. "%"

Removed the '%'

L354. Remove first occurrence of "rates"

Corrected from:

"Ice cliffs backwasted at rates similar rates with and without ponds and streams at their base and there is no apparent aspect dependence on backwasting rates (Fig. 7)."

to:

"Ice cliffs backwasted at similar rates with and without ponds and streams at their base and there is no apparent aspect dependence on backwasting rates (Fig. 7) ."

L379. This should be a full sentence.

Original text:

"In total, 11.7 % of the debris-covered tongue of Kennicott glacier is occupied by ice cliffs. See Anderson (2014) for an estimate using an independent method on Kennicott Glacier that is consistent."

Changed to:

"In total, 11.7 % of the debris-covered tongue of Kennicott glacier is occupied by ice cliffs (see Anderson (2014) for an independent and consistent estimate of ice cliff coverage)."

L381. Reference needed here for Changri Nup. Maybe tell that Changri Nup is small (to re-emphasize the added value of your study of examining a large glacier, the largest debris-cover studied so far I think). This comparison to published work could be moved to the discussion by the way.

We added a reference and moved these few sentences into the discussion as requested. The new paragraph reads:

"The 11.7 % ice cliff coverage in the debris-covered tongue (24.2 km$^2$ in area) of Kennicott Glacier is the highest coverage from any glacier studied to date. The 11.7% coverage is 60% more coverage by percentage than the debris-covered portion of Changri Nup Glacier, the glacier with the second highest ice cliff coverage (Brun et al., 2018; Table 4). The debris-covered portion of Changri Nup Glacier is also substantially smaller in area (1.5 km$^2$) when compared to the debris-covered tongue of Kennicott Glacier (24.2 km$^2$). The Kennicott Glacier has the lowest mean debris thickness (13.7 cm) of glaciers with reported ice cliff coverage percentages and supports, by far the highest percentage of ice cliffs. This implies that ice cliff coverage could vary with debris thickness or a variable that co-varies with debris thickness (Table 4)."

L389. "In Figure 11, we"

Corrected as suggested to: "In Figure 11, we"

Also 8+26 = 34% What process is responsible for the remaining 66%? Authors are loosing their readers here (at least me...)

This is a typo and a good catch thank you!

The sentence now reads:

"When averaged across the entire study area, 74% of melt is derived from sub-debris melt and 26 (with extreme bounds of 20 , 40 )% from ice cliff melt."

L392. "26 (20 , 40 )%" Unclear notation. What 20 and 40% correspond to. Need to be explained/defined.

The sentence has been amended to explain the bounds. It now reads:

"When averaged across the entire study area, 74% of melt is derived from sub-debris melt and 26 (with extreme bounds of 20 , 40 )% from ice cliff melt. "

L397. Do authors mean "below" rather than "above"?

Yes the reviewer is correct. Corrected as suggested:

"The dominance of decreasing sub-debris melt downglacier, due to thickening debris, results in a deviation from the bare-ice melt rate below 700 m a.s.l. (relative to the 2013 glacier surface)."

L434. Help retaining? Help to retain?

This whole section was re-written following the suggestion of the other reviewer so this issue has been resolved.

L460. Before application "to" different glaciers

"to" was added as suggested.

L482. Seems like a repetition of the same statement 2 lines above. Try to be more concise when possible.

Original text:

"Ice cliffs tend to contribute a higher fraction of mass loss as debris thickness increases. This trend is visible on Kennicott Glacier as debris thickens toward the terminus (Fig. 12). This relationship also appears to hold when considering debris-covered glaciers from different regions (Table 4). As debris thickens the contribution of ice cliff melt also tends to increase. This appears to occur even though the fractional coverage of ice cliffs tends to decrease as mean debris thicknesses increase."

New text:

"Ice cliffs tend to contribute a higher fraction of mass loss as debris thickness increases. This trend is visible on Kennicott Glacier as debris thickens toward the terminus (Fig. 12). This relationship also appears to hold when considering debris-covered glaciers from different regions and even though the fractional coverage of ice cliffs also tends to decrease as mean debris thicknesses increase (Table 4)."

L488. Again exact repetition of a statement L484. This lack of concision is a pity, as it makes the readers (reviewer...) nervous.

We removed the repeated line as suggested by the reviewer.

The original paragraph:

"Ice cliffs do not counteract the insulating effects of debris on Kennicott Glacier (Fig. 12). The thin debris within the study area leads to melt rates closer to bare-ice melt rates than most other studied debris-covered glaciers. Measured ice cliff backwasting rates are comparable or higher than measurements from other studies (Table 3). Kennicott Glacier also has the highest fractional coverage of ice cliffs, relative to other studied glaciers, which also serves to increase melt rates (Table 4). Despite this, ice cliffs on Kennicott Glacier do not compensate for the insulating effects of debris. This suggests that the presence of ice cliffs is unlikely to counter the insulating effects of debris on glaciers with thicker debris and/or lower ice cliff coverage."

The new paragraph:

"Ice cliffs do not counteract the insulating effects of debris on Kennicott Glacier (Fig. 12). The thin
275 debris within the study area leads to melt rates closer to bare-ice melt rates than most other studied
debris-covered glaciers. Measured ice cliff backwasting rates are comparable or higher than
measurements from other studies (Table 3). Kennicott Glacier also has the highest fractional coverage
of ice cliffs, relative to other studied glaciers, which also serves to increase melt rates (Table 4).
Despite this, ice cliffs on Kennicott Glacier do not compensate for the insulating effects of debris. This
280 suggests that the presence of ice cliffs is unlikely to counter the insulating effects of debris on glaciers
with thicker debris and/or lower ice cliff coverage."

L501. I must say I am not found of this exercise. I do not feel we learn much from it.

285 We find this exercise to be revealing and vital to the conclusions of this manuscript! It shows how
dominant debris is in controlling the mass balance profile on Kennicott Glacier. This is essentially a
sensitivity test and we think an important contribution to the field . The other reviewer also had no
issue with this line of reasoning. We do not make a change here.

290 The other reviewer said this about this section: "Convincing hypothesis tests above."

L503. This is as reduction factor of 10 or a multiplicative factor of 0.1

Removed "(a 10-fold reduction)" to remove ambiguity.
295
L519. useful sentence???

This sentence and paragraph was removed and replaced with new text.

300 Conclusion. Again lack of concision. The conclusion should extract the main message and not a list of
12 bullet points… For me the main message is that for the largest debris-covered glaciers studied so far,
despite a very high density of ice cliffs compared to others studied glaciers, their contribution to the
tongue-wide ablation is moderate. To be a bit provocative, maybe our community should spend more
energy estimating spatial pattern of debris thickness than ice cliffs density.
305
We re-formated the conclusions following your suggestion. They are now in paragraph form. We also
removed the conclusions related to details of the in situ measurements. We also incorporated your
suggestion of the main message. Based on conversations with other researchers, they are interested in
other conclusions as well so we also include other items here.
310
L535. "near the terminus near the margin" strange formulation.

This is simply a typo which has been corrected.

315 L568. Is not "vital" a bit strong???

This word choice seems rather justified from the rest of the analysis above, as well as the extensive sensitivity analysis in the supplemental materials. By process of elimination ice dynamics are necessary here. The other reviewer had no issue with this word choice, either. The conclusion was re-written without this word.

We add a new paragraph above the conclusion that makes this more clear.

Supplement.
Authors should use the same font throughout the text of the supplement, organize the different sections clearly for easier navigation, e.g., increase a bit the line spacing also.

The text, not associated with section titles is now all the same font. We increase the spacing between sections though.

Not providing a line-numbered supplement seriously complicated the life of referee who was almost exhausted when he started to review this part of the article…

Sorry for this. Just didn't cross our minds. We will do this for future submissions, though.

Reducing the size of the supplement file (55 Mo) should be possible without loosing readibility.

The size was reduced and is now within TC guidelines (less than 50 MB).

Fig S3. Dead ice portion. Above or below 5 cm/day? For me dead ice is almost stagnant.

Good point. This is the error associated with the method from the citation in the caption. Because Armstrong et al. (2016) used this definition we choose not to deviate here from that definition already published.

Fig S4 "repeated measurements"

Good catch.

"Error bars in measured debris represent changes in debris thickness measurements upon repeated measurement."

changed to:

"Error bars for debris thickness represent changes upon repeated measurement."

Fig S5. Why not showing directly the difference instead of two set of points?

While this could be better there are already a ton of plots and we feel that this plot is just as good at showing how this correction does not effect the overall results of the study. Ultimately this seems like more work than the potential payoff for readers.

Fig S6. It is not clear how the others studies were selected. I think it would be very useful to add the data from a previous compilation made by Kraaijenbrink et al. in their Supplementary Figure S5. Your compilation would gain exhaustivity.

Kraaijenbrink, P. D. A., Bierkens, M. F. P., Lutz, A. F. and Immerzeel, W. W.: Impact of a global temperature rise of 1.5 degrees Celsius on Asia's glaciers, Nature, 549(7671), 257–260, 2017.

As in the caption: the point here is to only show data from debris-covered glaciers at a **similar latitude** to the Kennicott Glacier. The plot only includes data from glaciers are similar latitudes and these were all of the data we could find. Including more glaciers will detract from the point of the plot. That is not to take away from the Kraaijenbrink compilation at all just not our intent with this plot.

I do not see the point of having some figures repeated both in the main and in the supplement (Fig 16 and 17 at least).

We removed these figures.

Fig S17-S35. Cannot these sensitivity tests be summarized in one or two paragraphs? Do the authors really need to show all figures?

We do this on purpose so the reader can see the effect of each of these changes in the supplementary material. The main text has such a summary. We do not want to repeat that here. The first round of review seemed to also miss our description of uncertainty so we also wanted to make it clear the effect of each of these changes. For these reasons the figures are included. We also do not understand the issue with having a longer supplementary section. There is simply just more evidence to support the main conclusions in the text, so we leave the content in but work on the organization of the supplemental material.

**Report #2**

Thank you for taking the time to review this manuscript!

Review of 'Debris cover and the thinning of Kennicott Glacier, Alaska: in situ measurements, automated ice cliff delineation and distributed melt estimates' by Anderson et al, submitted to The Cryosphere

The revised manuscript by Anderson et al has combined parts A and B to focus on distributed melt estimates for most of the debris-covered area of Kennicott Glacier, Alaska. The scope is much clearer and the study is greatly improved in this formulation, which I appreciate has been a major undertaking for the authors. The presentation and analysis of observations from debris covered glaciers in Alaska are a welcome contribution to The Cryosphere, and this study uses empirical models to complement the remote sensing observations and inferences of Brun et al (2018)

regarding the importance of ice cliffs and sub-debris melt, and the resulting patterns of glacier-wide melt rates.

410 At this point I have few substantive comments, although I think the study would benefit aesthetically from additional careful editing and textual revision. In particular, some content seems misplaced between Methods/Results/Discussion,

We have addressed this issue by following all the comments of this reviewer. We moved one paragraph
415 into the introduction and melded another into the discussion. See below detailed comments for the specifics.

and the storyline is not entirely clear for the discussion,which comes up just short of stating explicitly that since melt rates are not the key factor driving the zone of maximum thinning, reduced ice fluxes
420 must play an important role.

We have streamlined the discussion by removing several more minor points, especially in the ice cliff backwasting section. We replaced Section 4.4 with one that walks the reader through our conclusion that the reduction of ice flow from upglacier is necessary for explaining the location of the ZMT.
425

My comments below are extensive, but primarily relate to the presentation of the study. There are a few more substantive comments, but none that require additional analysis.

430
Summary comments:

1. A need for careful final editing in terms of grammar and accuracy. At present the writing style is not particularly satisfying to read: many sentences start with 'it,' 'but,' 'and,'
435 'because,' etc that should be linked to the preceding thought, and the editing/proofreading has not been thorough.

We combined these sentences following the reviewer's advice.

440 2. When uncertainty values are given, it is not clear how they were derived or what they correspond to.

We have clarified this in each location requested by the reviewer. We also removed any repeated mention of the in situ uncertainty values.
445

In terms of representativeness, the authors seem to make some key
unstated assumptions: 1) that the peak ablation season is a good proxy for annual-average
ablation rates (possibly not true for ice cliffs), 2) small-scale debris thickness variability is not
important to assess glacier-scale melt rates despite the nonlinear relationship between
450 debris thickness and melt).

Although I disagree with both of these assumptions, I do not
think they affect the conclusions of the study. I do, however, think those assumptions should

be made clear; this study makes good use of empirical methods to ask an important
455   theoretical question, but the reader needs to be reminded of the fundamental assumptions
of the framework of analysis.

Here the two reviewers clash. One would like a simpler, more fluid read, while the other prefers more
details. This is one reason why supplemental is large. It allows readability of the main draft while also
460   providing the detailed support the reviewers desire.

There are valid arguments why the assumptions laid out here by this reviewer do not matter for the
conclusions of this paper. The reviewer even states that these assumptions are unlikely to affect our
conclusions. We mention these assumptions now in the lower part of the discussion but also point to a
465   new section in the supplemental that further discusses them to increase readability as per the desires of
reviewer 2.

For the small-scale variability we now include this sentence in Section 4.3.1:

470   "Our distributed melt-estimation approach assumes that small-scale debris thickness variability has a
negligible effect on area-averaged melt rates, despite the non-linear debris-melt rate relationship. The
sensitivity test in this paragraph reveals how improbable it is for small-scale debris variability to lead to
maximum melt rates in the *ZMT* (see Section S1.7 for further discussion)."

475   We now include this paragraph in the Supplemental material for the small-scale variability assumption
(note that we follow the logic of the reviewer here in the text we include):

**"**Our approach assumes that small-scale debris thickness variability has a negligible effect on area-
averaged melt rates, despite the non-linear debris-melt rate relationship. Including small-scale
480   variability in debris thickness cannot decrease the *effective* debris thickness (therefore increasing local
melt) below the minimum in situ measured debris thickness in any given elevation band. We see from
Figure 5 that the minimum measured debris thicknesses are much larger in the *ZMT* than immediately
above the *ZMT*. Small-scale variability in debris thickness therefore cannot to lead to a peak in melt
rates in the *ZMT*. It is also highly likely that the range of plausible effects from small-scale debris
485   variability lie within extreme sensitivity tests as described in this and Sections 4.4, and S3.1, further
suggesting that our assumption is unlikely to affect our conclusions."

For the ice cliff assumption we now include this paragraph in section 4.3.1:

"We implicitly assume that the peak melt season (the June to August study period) is a good proxy for
490   annual-average ablation rates, though it is exceedingly unlikely that this assumption affects our
conclusions. During shoulder seasons ice cliffs tend to produce high *relative* melt, even though
*absolute* melt must decrease due to the decline in available energy. In order for *absolute* annual area-
averaged ablation rates to be maximized in the *ZMT*, ice cliff backwasting rates in shoulder seasons
would need to be more than 6.5 times those measured in the summer of 201, due to reduced sub-debris
495   melt, and last for the same duration as the June to August study period."

3. Some content related to the continuity equation in the introduction and methods, but then this isn't really discussed in the results and discussion at the end. I guess these parts of the analysis have been retained for later publication, but it feels like an unfinished thought here; perhaps not entirely necessary (yet) or a bit of discussion would close the gap to simply indicate clearly that based on this analysis it should be tested whether a reduced ice flux can explain the ZMT.

This is a good point. We remedy this by adding a section at the end of the discussion to tie the continuity equation back in and explain our logic better.

Comments:
L18. 'melt reducing' is a compound modified and should normally be hyphenated

Corrected here and throughout the manuscript.

L19. This sentence is a bit complex due to the frequent commas. There should not be a comma after 'thinning'. The 'but' is counterintuitive here and is better replaced with 'and' as Kennicott is an example of the dynamics represented above. However, the 'under insulating debris cover' is only demonstrated by your manuscript later on (it is not yet background knowledge that the debris is thick), so perhaps best to remove 'insulating'.

The abstract was re-written.

L22. Although an excellent contribution, this is a semi-automated method as it requires training data.

We appreciate the reviewers opinion on the the definition of 'automated' but we disagree. This is the first application of this method such that the parameters need to be estimated and the results validated against digitized ice cliffs. Further application of the ABT method to other can use those same parameters and apply the method to other glaciers. So it is not clear how the reviewer can assert that '**training data is required.**' From our view that is a presumptuous statement about the method we develop here. For these reasons and the fact that the other reviewer does not have an issue with our use of the term 'automated' we choose to make no change.

L23. Why the comma after 'relationships'?

Corrected.

L25. 'which' should be 'and' for the sentence to make sense grammatically: Ice cliffs cover 11.7% of the debris-covered tongue [, ... ,] and contribute...' L26. 'with a' should be 'which has'. As written, the literal meaning is that the ice cliffs have a mean
debris thickness of 13.7cm, but I suppose that this should refer to the debris-covered area.

Corrected.

The sentence was split into three and now reads:

"Ice cliffs cover 11.7% of the debris-covered tongue, the most of any glacier studied to date. Within the study area, debris cover is relatively thin with a mean thickness of 13.7 cm. The abundant ice cliffs contribute 26% of melt in study area."

L29-30. This is the first time that the decline in ice discharge is mentioned in the abstract. Introducing at the end of current line 24 with a succinct methodological description would clarify to readers where this comes from. Otherwise, the abstract should also have a line along the lines of 'We find a decline in ice discharge from up-glacier...'

For us it doesn't make too much sense to put a description up higher. An easier fix for this issue is to add a bit more text to L29 which explains the logic.

"We therefore suggest that the decline in ice discharge from upglacier is the vital control defining the zone of maximum thinning."

changed to:

"By process of elimination from the continuity equation for ice we therefore suggest that the decline in ice discharge from upglacier is a necessary control on the location of the zone of maximum thinning."

L37. The debris is not itself expanding, but the debris-covered areas are.

Changed from:
"Adding to this insulating effect, debris is expanding for many glaciers even as they contract in response to climate warming (e.g., Tielidze et al., 2020). "

to:

"Adding to this insulating effect, debris covers are expanding for many glaciers even as they contract in response to climate warming (e.g., Tielidze et al., 2020). "

L38-39. I suggest joining these sentences as they are directly linked theses. It's rarely a good idea to start a sentence with 'But'. The first clause, however, is a hypothesis that should have a reference. In addition, the only study I know of that has assessed whether debris is actually thickening rather than simply expanding in coverage is (Gibson et al., 2017).

We correct all of these issues:

"Expanding and thickening debris cover should reduce glacier thinning relative to glaciers without debris. But the melt-suppressing effect of debris is not always apparent the observed thinning patterns of glaciers even when debris is thick and debris coverage is extensive (e.g., Kääb et al., 2012; Gardelle et al., 2013)."

changed to:

590

Expanding and thickening debris cover should reduce glacier thinning relative to glaciers without debris (Banerjee, 2017; Gibson et al., 2017), but the melt-suppressing effect of debris is not always apparent the observed thinning patterns of glaciers even when debris is thick and debris coverage is extensive (e.g., Kääb et al., 2012; Gardelle et al., 2013).

595

L41. (F. Brun et al., 2019) is a better reference for this phenomena, although the 2018 study is also appropriate.

We think the "e.g.," already in the text takes care of this comment. No change made.

600

L43. Again, I suggest linking this sentence to the one before: ', and has been documented...' Also, Asia has already been mentioned just two sentences before, so this could be 'and has also been documented in Europe'.

605 We follow the suggestion from the reviewer.

"This apparent paradox, in which rapid thinning is occurring under insulating debris cover is known as the 'debris-cover anomaly' (Pellicciotti et al., 2015). It has been documented in both Asia and in the European Alps (Nuimura et al., 2012; Agarwal et al., 2017; Lamsal et al., 2017; Wu et al., 2018; Mölg
610 et al., 2019)."

original changed to:

"This apparent paradox, in which rapid thinning is occurring under insulating debris cover is known as
615 the 'debris-cover anomaly' (Pellicciotti et al., 2015) and has also been documented in the European Alps (Mölg et al., 2019). "

620 L46. This phenomena being global or not is a bit out of scope for this paper. It's an interesting supposition but the important aspect for the background of your study is that it also occurs in Alaska.

A small change to the wording can address this comment, while also honoring our intentions here.

625 Changed from:

"The 'debris-cover anomaly' may in fact be a global phenomena."

to

630

"The 'debris-cover anomaly' is occurring in Alaska, suggesting that it may be a global phenomena."

L47. No need for 'from' in 'from within'

635

Corrected.

L48. 'Compelling' is out of place or a word is missing: 'This is a compelling ___ because ...'

Removed 'a' to fix this issue.

Edited sentence:

"This is compelling because the Wrangell Mountains are in Alaska and at a latitude (61 to 62 deg. N) where the effects of debris on glacier mass balance has received almost no attention."

L54-55. I suppose that this question is the overarching drive of this analysis, and should be highlighted as such rather than as a rhetorical question in the text.

We follow the suggestion of the reviewer:

"Why does the maximum thinning of Kennicott Glacier occur under debris at rates similar to nearby debris-free glaciers?"

changed to:

"This brings us to our overarching question: *Why does the maximum thinning of Kennicott Glacier occur under debris at rates similar to nearby debris-free glaciers?"*

Eq 1. The equation requires units for each of the quantities to be the same (m/a or m w.e./a) which should be noted somewhere, since b* usually differs in units from dH/dt and the flux divergence.

This equation is used for the introduction formally laying out that both mass balance and ice dynamics matter for glacier thinning. The inclusion of units here just complicates the introduction and ties in units where they are not relevant. Lower down we highlight that all melt is in ice equivalent units.

Also, the flux divergence terms are not strictly correct, they should be dQ_x/dx and dQ_y/dy and are partial derivatives.

We have edited the equation to reflect this suggestion:

$$\frac{dH}{dt} = \dot{b} - \frac{\partial Q_x}{\partial x} - \frac{\partial Q_y}{\partial y}$$

L61-75. It would be good to include a reference for the continuity equation and its simplified version here; this has been done before and deserves a reference. Cuffey and Paterson is the obvious choice because it contains all these concepts, but the continuity equation has been applied to mountain glaciers extensively in Peru and the Alps since the 1990's, and since the 1970's at least for the ice sheets. I'm not suggesting a comprehensive list, but a reference or two would be value-added.

685 From our view, the continuity equation is the result of applying mass conservation to the transport of a 'fluid.' The equation has been used since the development of fluid mechanics and by the first contributors to quantitative glaciology. For this reason we find the the Cuffey and Paterson reference is enough here because it is the state-of-the-art glaciology text that already honors that historical perspective. For this reason we do not make any change here.

L72. Benn et al (2017) is one of the few studies that have tried to estimate $b^*\_e$, and is highly
690 relevant.

Add the reference at the end of the sentence.

L76. This is a nice way to present the various hypotheses.
695

Thank you, we appreciate it!

L80. I believe you intended to italicize the 'M' in 'Melt' as well.

700 Corrected.

L84. The comma should be removed from this statement.

Comma removed.
705

L85. This is not necessarily upglacier of the debris, but upglacier of the thick debris.

The suggestion from the reviewer over complicates what we intend to stay here. By adding too many details this sentence will become too complex and take away from the point we are making. Whether
710 thin debris enhances melt in a fashion actually important for this process is up for debate (see Fyffe, et al., 2020). The first reviewer also did not take issue with this wording.  We leave this sentence as it is for these reasons.

Typo: 'thinning and reduced ice flow'
715

Corrected by adding a 'd.'

L87. (Fanny Brun et al., 2018) summarized this concept very well in their Figure 11.

720 L89. Typo: 'revealing'

Corrected from 'reveling' to 'revealing.'

L90. It's definitely marginally acceptable to reference your own thesis for a paper on the same topic.
725

My thesis is actually pretty different than the work presented here. Considering the history of activity on Kennicott Glacier we would like to point to it so readers will find an additional resource. Because

the other reviewer did not have an issue here, and we do not share the intuition of the review, we prefer to leave the reference in. This reference also cited again below.

730

L92. 'from across Kennicott Glacier' is already clear from this sentence

Removed.  The sentence now reads:

735  "If melt hotspots are the sole control on the *location and magnitude* of the zone of maximum thinning or *ZMT* for Kennicott Glacier then we should expect melt rates (averaged across the glacier width) to be maximized there."

L93-95. The mention of 2011 appears from nowhere since it hasn't been introduced that this is the
740  period you have data for. Perhaps it is tidier to leave the questions unconstrained temporally, but then in the next paragraph to introduce the summer of 2011 as the period of observations.

Good catch. We removed 2011 from these questions. The following paragraph now reads:

745  "To address these questions, we quantify the role of ice cliffs and sub-debris melt across the debris-covered tongue of Kennicott Glacier during the summer of 2011. We limit our scope to ice cliffs and sub-debris melt, leaving an examination of surface ponds and streams as later contributions. Our analysis is rooted in abundant in situ data collected from the glacier surface in the summer of 2011. We measured debris thickness, sub-debris melt rates, and ice cliff backwasting rates. In addition to helping
750  address the questions raised above, these in situ measurements, from this latitude and Alaska, are vital for developing a global perspective on glacier response to climate change as well as the next generation of global glacier models incorporating the effects of debris cover."

You then still need to be careful to indicate that the same patterns are expected to be apparent in the
755  summer of 2011 as for long term/annual mass balance and thinning, for example based on Das et al (2014) and possibly the ITS_LIVE velocities?

Indeed, we addressed this in the last set of revisions, so there is no need for a correction here.

760  Lower in section 4.3.1 you will find:

"For this discussion we make the assumption that the *ZMT* – which was stable between the 1957 to 2004 and 2000 to 2007 time periods-- remained in the same location during summer of 2011."

765  L98-99. I think it's fine to relegate the analysis of streams and ponds to another paper, but if these are also potential contributors to surface mass balance, doesn't this cut your thesis short? I.e. you are no longer able to make a statement for all melt hotspots combined, but only for cliffs. Now, we certainly know that there are few ponds on Kennicott, and their location is different to the TMZ; why not simply pretend they melt at the same rate as cliffs (somewhat as in (Kraaijenbrink, Bierkens,
770  Lutz, & Immerzeel, 2017)) so that your ablation budget is complete? Streams are even more problematic, since they are prevalent and unconstrained. I make this point not to criticize your work; I am convinced that the hotspots are only part of the explanation and that ice dynamics are vital to explain the debris cover anomaly. I just this your huge amount of work here will be much stronger by

representing these somehow (even as a hypothesis test).

Thank you for this suggestion. It is helpful. We include a hypothesis test at the end of the discussion that follows:

"While we do not explicitly document the melt rate of ponds and streams (i.e., melt hotspots) we follow Kraaijenbrink et al. (2017)'s approach and assume they melt at the same rate as ice cliffs. Using this logic, in order for ice cliffs (melt hotspots) in the *ZMT* to compensate for the insulating effects of debris, ice cliff (melt hotspot) area would need to increase from 11.7% to 90% of the glacier surface. Ice cliffs, ponds, and streams assuredly do not occupy 90% of the *ZMT*. This again suggests that ice cliff and other melt hotspots do not control the location of the *ZMT*."

L105. I believe this should be WorldView-1.

Corrected:

"We therefore present and apply a new method for remotely delineating ice cliffs using high-resolution WorldView-1 orthoimages."

L117. Is this 20% figure from your own digitization (if so, of what imagery and when?), or from (Scherler, Wulf, & Gorelick, 2018)? Just best to indicate the source (or 'approximately'), as these numbers often get recycled.

We follow this suggestion.

"As of 2015, 20% of Kennicott Glacier was debris-covered."

Changed to

"As of 2015, approximately 20% of Kennicott Glacier was debris-covered."

L130. There is an extra space before 'duration'.

Space removed.

Understanding that the scope of this study is the bulk of the ablation season for this glacier, have you considered the representativeness of this period to the full-year ablation budget? This is not likely to change your outcomes, but it is worth thinking whether ablation hotspots such as cliffs and ponds 'turn off' their melt contributions at the same time as the general debris-covered tongue. This probably depends based on the glacier- specific site and characteristics, but cliffs and ponds could have positive surface energy balanceduring periods of the year when conduction through debris is already negligible. The assumption here (with regards to the research question) is that the peak ablation season corresponds well to the average annual ablation across the glacier. I don't think this assumption is particularly bad, but it should be made explicit.

Absolutely this is certainly true but we agree with the reviewer that this effect plays a minor role. As described above we chose to add a few sentences about this in the discussion section. Because it is minor we do not mention it until the discussion section.

L136-142. I completely agree with all of this content, but it belongs in the Introduction.

These lines were moved up to the introduction as suggested.

L147. Perhaps combine this sentence with L148-149 'by digging through...'?

Corrected to:

"We measured debris thicknesses at 109 sites by digging through the debris to the ice surface (after Zhang et al., 2011). Debris measurement locations coincide with the sites where we also measured ice cliff backwasting and sub-debris melt (Fig. 4; Supplemental material). "

L152. It is not clear what these values correspond to – the uncertainty and variability of the measurements themselves, or (as it appears in Fig 5) of the altitudinal curve-fit. For each of these, a bit of additional information is needed. How did you get an thickness uncertainty of 0.3cm? This is incredibly precise considering the challenge of the measurement – excavation to the ice surface without disturbing the ice surface, finding the reference height, ensuring a vertical measurement, etc. For a 30cm pit, this implies a measurement within 6 degrees of vertical, ignoring all other uncertainties, or for 60cm, 4 degrees. What does the standard deviation correspond to – the variation between measurements at the same elevation? I guess that the 'maximum error' is the maximum standard error of the elevation's mean debris thickness?

In order to clear up any confusion we expand the explanation of errors. We just simply re-measured the debris thickness at 52 sites. We do not want to over complicate the text as well. The original text was:

"The mean uncertainty of the debris thickness measurements is ±0.3 cm, with a standard deviation of ±1.8 cm, and a maximum error of ±6.7 cm (Fig. 5). Error estimates were based on repeated measurements, but measurement error is a negligible compared to the changes in debris thickness down and across the glacier."

and was changed to:

"Based on repeated measurement of debris thickness at 52 sites (with a mean debris thickness of 7.9 cm), the mean uncertainty of the debris thickness measurements is ±1.3cm, the standard deviation is ±1.4 cm, and the maximum error measured within the population of re-measured sites is ±6.7 cm (Fig. 5; Fig. S4). We consider measurement error to be negligible compared to the changes in debris thickness down and across the glacier."

L159. Again, the 'how' of the uncertainty is as important as the value. In this case I would suspect

that this uncertainty represents the uncertainty in the stake height measurements, spread over time?

We have changed the original text:

"The mean uncertainty in the sub-debris melt rates was ± 0.1 cm d$^{-1}$, the standard deviation was 0.05 cm d$^{-1}$, and the maximum error was 0.25 cm d$^{-1}$ for the three ablation states with the shortest measurement period of 8 days. These measurement uncertainties are small compared to the changes in melt rate with debris thickness (Fig. 6)."

to:

"We estimated uncertainty by assuming a combined ±2 cm uncertainty in the marking of the ice surface on each ablation stake and the measurement between marks. We include all stakes as they vary in space and time. We measured the tilt of each ablation stake at the end of the measurement period which averaged 5° from vertical. The mean uncertainty in the sub-debris melt rates is ±0.06 cm d$^{-1}$, the standard deviation is 0.03 cm d$^{-1}$, and the maximum error is 0.125 cm d$^{-1}$ for the three ablation stakes with the shortest measurement period (8 days). These measurement uncertainties are negligible compared to the changes in melt rate with debris thickness (Fig. 6)."

Similarly for L176.

This information was in the supplemental figure referenced in the sentence but it should be brought into the main text. We bring it into the updated sentence now:

"The mean error of the ice cliff backwasting rates is ±0.5 cm d$^{-1}$ (Fig. 7; Fig. S9). Maximum error is ±1 cm d$^{-1}$ for 10 cliffs that were measured over the shortest interval (21 days). The standard deviation of errors is ±0.2 cm d$^{-1}$."

Changed to:

"The mean error of the ice cliff backwasting rates is ±0.5 cm d$^{-1}$ based on an estimated measurement uncertainty of ±20 cm applied to both the initial and final distance measurement (Fig. 7; Fig. S9). Maximum estimated error is ±1 cm d$^{-1}$ for 10 cliffs that were measured over the shortest interval (21 days). The standard deviation of errors is ±0.2 cm d$^{-1}$ based on error estimates from all 60 ice cliffs."

L165. Typo in 'nonetheless'

corrected to 'nonetheless'

L166-171. This is nice content and a very tidy synthesis of key points from past work on cliffs, but shouldn't this be in the Introduction? No methods appear until L171-2

We could not find a natural way to incorporate this into the introduction so we moved the content of these sentences to the discussion where they directly relate to the arguments we make. So we effectively follow the reviewer's suggestion.

910 L196-197. The content of the stereoimagery is out of place here, and interrupts the description of the ice cliff mapping method, for which it seems irrelevant. Perhaps it could be part of 2.3, since you use the elevation to prescribe the melt rates across the glacier?

Good catch. We moved this content into the introduction of the methods. Which now reads as:

915

"Our methods fit into three broad categories: 1) in situ measurements; 2) automatic ice cliff delineation; and 3) distributed melt rate estimates. In situ measurements were made within the broad study area shown in Figure 1C, which is within 8 kilometers of glacier terminus. Distributed melt estimates on the other hand are made across the delineated medial moraines shown in Figure 4A. In

920 total the distributed melt estimates were made over 24.2 km$^2$ which we consider here to be the 'debris-covered tongue' of the glacier. In situ measurements were all made within the full field campaign duration and study period from 18 June to 16 August 2011. We correct each measured melt rate to represent the full duration of the study period, as described below. We used WV stereoimagery from 2013 to produce glacier surface DEMs at 5 m spatial resolution using the Ames Stereo Pipeline (Shean

925 et al., 2016), which we use to represent the glacier surface during the study period."

L222. As written, I'm not sure which problems are for 'ABT and SED' or 'SED only'. This becomes clearer later in the paragraph, so maybe simplify and remove the indication of which method suffers from which problems here?

930

We clarified this by making sure it is clear which problems are linked to which method:

"The last step in our processing process is morphological filtering to remove spurious data.  Both delineation methods (*ABT* and *SED*) produce false positives from shaded, over-exposed, or textureless

935 debris cover (*SED* only). The *SED* approach produces many false positives, which generally have a characteristic speckled appearance, and often occur in small, isolated groups. We apply morphological opening (Dougherty, 1992) to remove these isolated false positives in both the *ABT* and *SED* approaches (skimage.morphology.opening; Fig. 8). In addition, the *SED* approach creates false positives in regions that have been over-exposed by the saturation stretch and therefore lack texture.

940 For the *SED* method only, we remove these false positives by masking pixels with the maximum brightness."

We felt that removing the labels for the different methods would actually do the opposite of what the reviewer intends so we clarified the issue in another fashion.

945

L231-2. Makes sense to reference (Steiner, Buri, Miles, Ragettli, & Pellicciotti, 2019) here. This difference is 3% of total cliff area or 3% area mismatch (non-overlapping areas)? Please make it clear in the text.

950 We note now that the total ice cliff area differs by 3 %. Changed from:

"As these independent delineations agreed within 3% in their ice cliff area, we consider operator misidentification to be a negligible source of error."

955 to

"As these independent delineations agreed within 3% in their total ice cliff area, we consider operator misidentification to be a negligible source of error."

960

L259-270. I am impressed by the effort to represent the debris thickness variability based on flow units, which is a key aspect of spatial heterogeneity in deriving glacier-wide melt estimates based on (necessarily) spatially-biased measurements.

965    Thank you for recognizing that!

L274. It's nice to see h* explained in very clear terms (the half-melt thickness) and related to physical meaning. Please do add units to the definitions of each variable.

970    The units for terms not explicitly mention are obvious based on the other terms and units provided. We feel that adding the units here would clutter the description so we leave it as is.

L295. Indicate the unit of degrees in the equation (for 90).

975    changed to:

$$\dot{b}_{icecliff} = \dot{b}_{backwasting} \cos(90° - \theta)$$

980

L312. Verb tense issue: '... the curve fits ... are calculated ...'

Corrected to:

985    "For the best case the curve fits through debris thickness are calculated using the median of data from the 50-m elevation bins (Fig. 5)."

L317. 'ice cliffs' should be singular (ice cliff slopes) or possessive (ice cliffs' slopes)

990    Changed to: "We also apply ±1σ range for sub-debris melt and ice cliff backwasting rates, and a ±1σ range for ice cliff slopes."

L318. Doubled %%

995    removed one %

L340. Please do not start this sentence with 'And'.

The sentence now reads:
1000

"Debris consistently 1 m thick was observed just out of the study area but still in moraine 9 at 730 m a.s.l.."

L346. A suitable reference for this is (Mertes, Thompson, Booth, Gulley, & Benn, 2016).

1005

Reference added.

L383. Nice simple inference worth testing further. Most likely this relates to the difficulty of mobilizing debris, see e.g. (Moore, 2017).

1010

Thank you.

"This implies that ice cliff coverage could vary with debris thickness or a variable that co-varies with debris thickness (Table 4)."

1015

Changed to:

"This implies that ice cliff coverage could vary with debris thickness or a variable that co-varies with debris thickness, like the mobility of debris (Moore, 2018; Table 4)."

1020

L392. It's not immediately clear what the values in parentheses mean. Are these the lower/higher estimates? Just make it clear in the first instance.

Good catch. The sentence now reads:

1025

"When averaged across the entire study area, 8% of melt is derived from sub-debris melt and 26 (with extreme bounds of 20 , 40 )% from ice cliff melt. "

L421. Is this in cm? This content is a bit unrelated; the consistent decline in sub-debris melt is not
1030 unexpected for many other reasons as well, which h* just indicates in aggregate.

Yes, corrected 'm' to 'cm.' We are simply drawing attention to the consistency of the h* from Kennicott and similarity of Ostrem's curves from glaciers at similar latitudes with the global compilation and the fact that sub-debris melt is reduced as debris thickens. We do understand the need
1035 for a change here so we leave the text as is.

L428. Awkward comma placement. Best to get rid of 'Becuase' here, and connect to the next sentence (which begins with 'But'). Again, see (Moore, 2017) for a review of these processes.

1040 This section was removed to streamline the discussion follow both reviewer's comments.

L423-446. This is interesting content but largely conjecture and lacking a synthesis – so there are many possible reasons for heterogeneity in cliff backwasting rates as you measured, but what does it mean for science? What is the point? That we need large N? Or high-quality measurements and
1045 models? Or...?

We re-wrote this section following the reviewer comments. Ultimately we need to discuss the trends or lack thereof in our ice cliff measurements. We chose to present that discussion in a form that presents a few hypotheses and support those hypotheses with references. They are inferences that could be further explored by the community.

L457. This sentence undermines itself: 'Our method works if you correct it for bias.' I think the point here is that your method presents the spatial variability of ice cliff areas very well, even if it underestimates the total cliff area somewhat; this spatial variability is key for you to derive the subdebris melt estimates!

We adjust the wording following the reviewer's suggestion:

"Our automated methods provide an accurate estimate of ice cliff area as it varies across a large debris-covered tongue. Both the *ABT* and *SED* ice cliff delineation methods underpredict ice cliff area somewhat."

It would be nice to see how consistent those biases are between the distinct evaluation patches, possibly as a panel in the SuppMat. A lot of content, time and effort has gone into the cliff mapping method and it would be nice to see the high quality results in more detail there.

We appreciate the reviewer's desire for more analyses but frankly we do not see how this is needed. Perhaps a comparison between ice cliff delineation methods should be performed with independent operators?

Considering that the reviewer in their introduction noted about their review:

"There are a few more substantive comments, but none that require additional analysis."

We chose not to provide more analyses to a supplemental with 25 figures, one that the other reviewer finds too long already.

L489. Great synthesis. My only suggestion would be to also clearly acknowledge that ice cliffs greatly enhance the overall melt rates for the debris-covered area; as presently written, the implication is that one only has to account for the increasing debris cover to get a reasonable melt estimate; this would underestimate melt by 26% in your results. Clearly the debris dominates the glacier-wide melt rate and wrt elevation, but also it is clear that you need to take cliffs into account somehow.

As this paragraph is written in no way does it imply that ice cliffs should be neglected. Rather it simply states that area-averaged melt rates are dominated by debris that reduced melt relative to bare ice. Perhaps this is written in a fashion that is not typical for ice cliff studies but it is still accurate. There is a running misconception in the community and the literature where % melt contribution is confused with the absolute melt contribution.

To assuage the reviewer we add this sentence in the new conclusion:

"While ice cliffs should not be neglected, our analysis suggests that the community should focus more energy on quantifying debris thickness and sub-debris melt rates as they vary across individual glaciers and regions."

L515. Convincing hypothesis tests above. For completeness, you may as well perform a back-of-envelope for cliffs and streams to emphasize that, even poorly-constrained, these features cannot source sufficient energy for melt to be the only driver of the ZMT.

We follow this advice and include this new paragraph:

"While we do not explicitly document the melt rate of ponds and streams we follow Kraaijenbrink et al. (2017)'s approach and assume all melt hotspots melt at the same rate as ice cliffs. Using this approach, in order for melt hotspots in the *ZMT* to compensate for the insulating effects of debris, melt hotspots would need to cover 90% of the glacier surface, which they assuredly do not. This again suggests that ice cliffs (and other melt hotspots) do not control the location of the *ZMT*."

L515-517. Agreeing that melt is not maximized in the ZMT, isn't it also clear that cliff and debris melt is maximized (although still less than for clean ice) leading into the ZMT? I.e. aren't cliffs and high subdebris melt rates important drivers of the reduced ice flux into the ZMT?

Interesting idea but not what we consider to be the key process here. We address this idea in a new section at the end of the discussion that ties the continuity equation back into the manuscript.

L519. 'debris-covered glacier termini' appears to have been accidentally used twice.

Corrected to: "On debris-covered glacier termini, debris tends to thicken downglacier (e.g., Anderson and Anderson, 2018) as is the case for Kennicott Glacier.**"**

L522. (Bisset et al., 2020) examined selected glaciers across HMA, not only in the Everest region

Corrected to: "This leads to the expectation that sub-debris melt rates will decline towards the terminus reversing the mass balance gradient, similar to the results and conclusions for selected glaciers across High Mountain Asia (Bisset et al., 2020)."

L524. The 42% figure comes from nowhere – do you mean Ngozumpa via Thompson et al (2016)? Anyways I think the suggestion is a good simplified debris thickness representation – if you can figure out how to estimate the apportioning of melt to cliffs!

We agree and removed that sentence, though it was referring to Kennicott Glacier and the elevation binned percentages in Fig. 12. The new paragraph is here:

"On debris-covered glacier termini, debris tends to thicken downglacier (e.g., Anderson and Anderson, 2018) as is the case for Kennicott Glacier. This leads to the expectation that sub-debris melt rates will

decline towards the terminus reversing the mass balance gradient, similar to the results and conclusions for selected glaciers across High Mountain Asia (Bisset et al., 2020). The overall mass balance profile for the summer of 2011 (Fig. 12) shows this Østrem's curve like pattern, suggesting that it is more strongly influenced by debris thickness than melt hotspots. Future efforts to represent the effects of ice cliffs on glacier mass balance at the regional scale should consider using a modified debris thickness-melt relationship with a percentage enhancement based on empirical relationships between debris thickness and ice cliff melt contribution. "

L530. To my knowledge it is the first time that melt has been quantified for a debris-covered tongue in Alaska, and one of the few times it has been done convincingly, with in situ measurements, for a debris-covered area globally. Nice work.

Thank you.

L531-569. Please rework this long list (12 items!) of bullet points into cohesive paragraphs.

The conclusion was re-worked into paragraphs.

L572. Please just archive the data in a repository. You can still force people to request them or let you know what they intend to use the data for.

We are making the in situ data available openly. The DOI is in the manuscript now.
Table 1. The values for ice cliff backwasting in the 'min' and 'max' cases do not seem to correspond to Table 2 or Figure 7 (these would be 4.6 and 9.6). Which values did you use?

Good catch thank you. For Table 1 it is the values in your parentheses. We changed the labels in Figure 1 to clarify that these are for uncertainty estimates. We also made it clear in Table 2 that those min and max values are for the in situ measurements.

Table 4. Note that (Thompson, Benn, Mertes, & Luckman, 2016) lumped all cliff-associated influence by necessity, whether due to cliffs/ponds/streams etc.

Added a note to the table:

"***Combined contribution from ice cliffs, ponds, and streams"

Figure 1. On panel (c) the legend need some work to show the variation (dH/dt) and units clearly.

We have now included this text in the caption: "The units for the legend are above the labeled colors."

Please also consider using a proper color ramp – I know it is a pain in QGIS.

We do not see how this change will actually improve the legibility of the figure. With a continuous colormap it will be nearly impossible to see the specific labels for the colors (a common problem using continuous colormaps). As it is we find that the legend is quite legible. Additionally reviewer 1 did not have any issue with these legends so we choose not to follow this suggestion.

I don't think there is a need for the statistics from the Das et al (2014) study in the caption, but if you want to include them, it's not clear what you mean by the mean error and 1 std – is the 'mean' the glacier-wide error of the mean, and 1 std (can be sigma) the standard deviation across the glacier? This is easily misunderstood.

We remove the statistics as requested.

"Map of the general study area with dH (dt)$^{-1}$ from 1957 to 2004 see Das et al. (2014) (mean error 0.04 m yr$^{-1}$ and 1 std 0.15 m yr$^{-1}$ based on 3 km$^2$ area within 4 km of the modern terminus)."

Changed to

"Map of the general study area with dH (dt)$^{-1}$ from 1957 to 2004 see Das et al. (2014)."

We include the statistics here because of the extra emphasis on the need for uncertainty to be defined in previous reviews.

1190

Figure 3. Very nice synthesis.

1195    Thank you!

Figure 4. As for figure 1, the legend in panel (a) needs to be tidied.

We think the non-continuous colormap is much more legible than a continuous colormap as the reviewer suggests. Because ourselves and the other reviewer did not find that this was a problem we choose not to make any changes here.

1200

Figure 5. Correction to caption text: 'The red bars are the median...'.

1205    Corrected.

For completeness of the manuscript itself, and also as a nice comparison to this relationship, I would prefer to see the relationships of debris thickness with elevation for the other moraines here.

We appreciate and understand the desire for this but do not this change will improve the manuscript as it will overwhelm the reader as per the comments of Reviewer 1. We now reference the supplemental figure where readers can see the data from each of the individual medial moraines

1210

Figure 6 caption. '... is smaller than the marker ...'. The RMSE units are presumably cm w.e. d^-1?

1215

Corrected.

Figure 7. I suggest to indicate 'ZMT' next to the arrow for clarity.

1220    ZMT was added to this figure as well as figures 2, 5 and 6.

In (b) the axis labels are not terribly clear: 'aspect' is not labelled anywhere (nor the units) and the ice cliff backwasting axis values are a bit confusing due to their location.

This is a good catch. We added 'degrees' to the 'aspect' labels. We rotated the backwasting rate labels so it is clear that they are different from the aspect labels. The caption does mention aspect and we think that adding the words 'Aspect' to panel 'b)' will confuse the reader.

Figure 8. In some respects the results without morphological opening look even better, because the opening has also removed thin slivers of true ice cliff area. Did you try a connected-pixels morphological clean? This would eliminate (for example) 'cliffs' with less than 10 pixels, and would generally leave the thin cliffs unmodified. Perhaps you are happy with the current performance, but it's just a suggestion.

Interesting suggestion which we would love to explore further but we do not see the reason now for an additional sensitivity test for this manuscript which is already full of analyses. I am sure there is a good argument for using other approaches as well. Thank you for bringing it to our attention.

Figure 9. In the caption, there is mention of 'thin red lines' but I don't see these anywhere. Perhaps use a different colour for these?

We adjusted the caption to better explain the thick and thinner red lines.

Figure 10. Please indicate the elevation bin size for the hypsometry.

Added to the caption: "All panels use 20 m elevation bins."

Figure 11. Why not delineate the ZMT here using elevation contours (thus corresponding directly to the rest of the paper)? Otherwise I find myself wanting to flip back several pages to see where the contours are, etc.

The ZMT is defined early on is defined by the area with thinning less than -1.2 m/yr. Because of the complexity of the topography on the glacier surface following the reviewer's comment will make the figure much less legible and actually detract from the legibility of the figure. Reviewer 1 had no issue with this and the ZMT is labeled clearly in the figures. For these reasons we choose not to follow this suggestion.

Figure 12. Small typo: '84.1% of estimates are within the grey shaded band.'

Thank you, corrected.

[revised manuscript text omitted]

---

## Author Response (AR3)

Thank you for your decision! We appreciate your comments and suggestions. Please see our changes and replies below.

I have few technical details for you to address, as I feel the English of the text could still be more polished.

We have re-read the manuscript again and smoothed the language.

I have suggested a restructured version of the abstract below, which maintains all your content of course. I feel the order of the paragraphs was not entirely logical, and some of the wording could be improved. Feel free to use that or a modified, more fluent and tidier version of yours.

We followed most of the suggested changes to the abstract, see below. We changed some wording as well.

I would encourage the authors to go through their text once more, and try to polish it some more and correct style/grammar, based on some of my comments below (in few instances the text contains still grammar errors). This is something that the co-authors could help with.

We have re-read the manuscript again and smoothed the language.

REVISED SUGGESTED ABSTRACT:
Many glaciers are thinning rapidly beneath melt-reducing debris cover, including the Kennicott Glacier in Alaska. The zone of maximum thinning at Kennicott Glacier is located under debris. Scattered within the debris cover, melt hotspots, such as ice cliffs, locally increase melt rates. We explore the roles of debris and ice cliffs in controlling rapid thinning under thick debris at Kennicott Glacier.

We collected abundant in situ measurements of debris thickness, sub-debris melt, and ice cliff backwasting allowing for extrapolation across the debris-covered tongue (the study area and the lower 24.2 km 2 of the 387 km 2 glacier). A newly developed automatic method, the Adaptive Binary Threshold, uses for the first time only optical satellite imagery to delineate ice cliffs. We show that the method accurately estimates ice cliff coverage even where ice cliffs are small and debris color varies.

Kennicott Glacier exhibits the highest fractional area of ice cliffs (11.7 %) documented to date. Ice cliffs contribute 26% of total melt across the glacier tongue. Although the relative importance of ice cliffs to area-average melt is significant, the absolute area-averaged melt is dominated by debris.

At Kennicott Glacier, glacier-wide melt rates are not maximized in the zone of maximum thinning, and we show that this rapid thinning is due to a decline in ice discharge through time. There is more debris-covered ice in Alaska than any other region on Earth. Through our efforts, Kennicott Glacier is now the first in glacier in Alaska and the largest globally where melt across its debris-covered tongue has been rigorously quantified.

We follow these suggested changes to the abstract (minus some minor changes to the wording) and move the two sentences to the end. Here is the new abstract:

"Many glaciers are thinning rapidly beneath melt-reducing debris cover, including Kennicott Glacier in Alaska where glacier-wide maximum thinning also occurs under debris. This contradiction has been explained by melt hotspots scattered

within the debris cover. However, melt hotspots cannot account for the rapid thinning at Kennicott Glacier. We consider the significance of ice cliffs, debris, and ice dynamics in addressing this outstanding problem.

We collected abundant in situ measurements of debris thickness, sub-debris melt, and ice cliff backwasting allowing for extrapolation across the debris-covered tongue (the study area and the lower 24.2 km$^2$ of the 387 km$^2$ glacier). A newly-developed automatic ice cliff delineation method is the first to use only optical satellite imagery. The Adaptive Binary Threshold method accurately estimates ice cliff coverage even where ice cliffs are small and debris color varies.

Kennicott Glacier exhibits the highest fractional area of ice cliffs (11.7 %) documented to date. Ice cliffs contribute 26% of total melt across the glacier tongue. Although the *relative* importance of ice cliffs to area-average melt is significant, the *absolute* area-averaged melt is dominated by debris.

At Kennicott Glacier, glacier-wide melt rates are not maximized in the zone of maximum thinning. Declining ice discharge through time therefore explains the rapid thinning. There is more debris-covered ice in Alaska than in any other region on Earth. Through this study, Kennicott Glacier is the first glacier in Alaska, and the largest glacier globally where melt across its debris-covered tongue has been rigorously quantified."

COMMENTS
COMMENTS RELATED TO THE REBUTTAL TEXT'

_On line 265 of the document (in the authors' reply to reviewer 1): you show as original and new paragraph exactly the same text! Please clarify here if and how you made changes to that text.

It looks like we mistakenly put the same text in for the rebuttal text: our apologies. The entire paragraph was re-written as was the discussion for the last iteration following Reviewer 1's comments to reduce repeated text.

_Text on line 590: there is a "in" missing I assume?

Original text:

"Expanding and thickening debris cover should reduce glacier thinning relative to glaciers without debris (Banerjee, 2017; Gibson et al., 2017), but the melt-suppressing effect of debris is not always apparent the observed thinning patterns of glaciers even when debris is thick and debris coverage is extensive (e.g., Kaab et al., 2012; Gardelle et al., 2013)".

It should be:

Expanding and thickening debris cover should reduce glacier thinning relative to glaciers without debris (Banerjee, 2017; Gibson et al., 2017), but the melt-suppressing effect of debris is not always apparent IN? the observed thinning patterns of glaciers even when debris is thick and debris coverage is extensive (e.g., Kaab et al., 2012; Gardelle et al., 2013)".

The 'in' was added as suggested.

_Lines 765 and following: In response to a comment by Reviewer 2: Krajeenbrink et al (2017) only consider ponds, not ice cliffs, in their melt estimates for HMA. I would also say that the evidence available until now suggests that the two do not melt at the same rate. So I suggest you rephrase the sentence you have added:

"While we do not explicitly document the melt rate of ponds and streams (i.e., melt hotspots) we follow Kraaijenbrink et al. (2017)'s approach and assume they melt at the same rate as ice cliffs. Using this

logic, in order for ice cliffs (melt hotspots) in the ZMT to compensate for the insulating effects of debris, ice cliff (melt hotspot) area would need to increase from 11.7% to 90% of the glacier surface. Ice cliffs, ponds, and streams assuredly do not occupy 90% of the ZMT. This again suggests that ice cliff and other melt hotspots do not control the location of the ZMT."

I would recommend that in this paragraph you simply state that you do not address ponds and streams here, that you will do it in a follow up publication, but that, given current estimates of their contributions to ablation, their inclusion will not change your main conclusions.

Reviewer 2 from the last round of reviews provided a clear way to include the plausible effects of these melt hot spots which we followed. It makes our analysis much more complete and therefore makes sense that we include it.

Through the three iterations of the review we have already changed the inclusion of ponds and streams in this manuscript multiple times. Reviewer 2 requested that ponds and cliffs be removed in the first round of review, then asked me to include them as we have for the minor revisions iteration. Now we are being asked to remove them again.

Everything written in the paragraph in question is factually accurate and the assumptions are clearly stated following Krajeenbrink et al (2017)'s approach and Reviewer 2's suggestion.

Here is Reviewer 2's full comment from the last iteration, which we followed for the latest revisions:

"L98-99. I think it's fine to relegate the analysis of streams and ponds to another paper, but if these are also potential contributors to surface mass balance, doesn't this cut your thesis short? I.e. you are no longer able to make a statement for all melt hotspots combined, but only for cliffs. Now, we certainly know that there are few ponds on Kennicott, and their location is different to the TMZ; why not simply pretend they melt at the same rate as cliffs (somewhat as in (Kraaijenbrink, Bierkens, Lutz, & Immerzeel, 2017)) so that your ablation budget is complete? Streams are even more problematic, since they are prevalent and unconstrained. I make this point not to criticize your work; I am convinced that the hotspots are only part of the explanation and that ice dynamics are vital to explain the debris cover anomaly. I just this your huge amount of work here will be much stronger by representing these somehow (even as a hypothesis test)."

We clearly state in the text that we 'assume that all melt hotspots melt at the same rate as ice cliffs.' Our methodology is very clear and the assumptions have been stated. We do not want to cite a future paper of ours to make this point that may or may not actually make the exact calculations that are included in the text now. This sensitivity analysis regarding ice cliffs and streams also naturally follows from the other sensitivity analyses in this section of the discussion. For these reasons and following the suggestion of Reviewer 2 we include the text below:

"While we do not explicitly document the melt rate of ponds and streams we follow Kraaijenbrink et al. (2017)'s approach and assume that all melt hotspots melt at the same rate as ice cliffs. Using this assumption, in order for melt hotspots to compensate for the melt-reducing effects of debris in the *ZMT*, melt hotspots would need to cover 90% of the glacier surface, specifically in the *ZMT*. This assuredly is not the case."

_Line 802: provide the source of that 20% value (your own digitisation).

Ok, the sentence now reads: "As of 2015, 20% of Kennicott Glacier was debris-covered (based on manual digitization of a Landsat image)."

_Line 876 and following: on uncertainty of ablation measurements: how do you combine the uncertainty in the marking and the uncertainty due to the tilt of the stake? Can you please explain?

Here is the updated text:

"We estimated uncertainty using data from all ablation stakes based on the uncertainty in marking and measurement as well as the tilt of the stake. We assume a ±2 cm error in the distance measurement along ablation stakes. The average-measured tilt of the ablation stakes was 5° from vertical. Bare-ice melt rates were also measured at several locations in the northeastern portion of the study area on the Root Glacier."

Depending on the ± of measurement error along the stake different tilt will change the uncertainty of the final measurement.

_Line 946: the reviewer suggested added reference: "L231-2. Makes sense to reference (Steiner, Buri, Miles, Ragettli, & Pellicciotti, 2019) here". And I would agree with that, even if this is one of our studies.

Ok, reference added.

_Line 970: I agree with the reviewer that you should provide units.

The units were already stated at the end of the sentence so there is now need to add them in again. Furthermore for the equations below: we use the to inform the reader about the model but do not make any calculations using those equations. We could put in units but it limits our aim, which is to teach the reader about the model.

COMMENTS RELATED TO THE REVISED TEXT
The following comments and lines refer to lines of the revised paper (in track-changes):

_Line 1340:
«Greater surface elevation changes»: say if these are positive of negatine, I assume you refer to thinning patterns here?

This was fixed by referring to 'thinning' instead of 'surface elevation changes.'

_Line 1434: I would remove this sentence
"leaving a detailed examination of other melt hotspots for another contribution.", or rephrase it. The sentence has it is grants an explanation of why they are not included here, and does not add much to the paper. There will most likely be a second contribution, and readers will be aware of it.

The inclusion of this text was suggested by Reviewer 2 in the last round and it reconciles issues related to mentioning hotspots but then only addressing ice cliffs in detail in this manuscript.

As this is a very minor issue and we prefer to keep the text as is and to keep the sensitivity analysis in as well. This text keeps the reader from wondering if we address the other hotspots in this manuscript and points them to the sensitivity analysis that was also suggested by Reviewer 2 in the discussion. See comments above regarding this additional sensitivity analysis.

_Line 1442
I would add, after (DEMs), "but showed that the method is sensitive to the DEMs resolution and its predictive skills diminish for coarser resolution".

We do not see the need to go into the details of performance for the respective ice cliff delineation methods. Adding this sentence distracts from the main direction of the paragraph so we prefer to keep the text more simple as it is.

_Line 1448
Consider removing: "thereby addressing the questions outlined above»

removed as recommended.

_Line 1496: I would remove "better"

We clarify this so our intent is more clear. The sentence now reads:

"For debris to be incorporated into large-scale models, debris thermal properties and onglacier meteorology must also be documented as they vary across glacier surfaces."

_Line 1566: I would rephrase as: We DEVELOP an automated algorithm to delineate ice cliffs from optical satellite imagery.

Changed to 'develop' as suggested.

_Line 1567 and previous occurrences: WorldView (WV) satellite- check where you use WorldWiew as a word for the first time, provide the acronym there and be consistent in its use. You have used both acronym and extended word in the text before this occurrence.

We only use WorldView in the text now. Thank you for catching that.